# Acetylation of TIR domains in the TLR4-Mal-MyD88 complex regulates immune responses in sepsis

Xue Li [1,2,3,8]✉, Xiangrong Li [4,8], Pengpeng Huang [1], Facai Zhang[1], Juanjuan K Du[1], Ying Kong[5], Ziqiang Shao[1], Xinxing Wu[1], Weijiao Fan[1], Houquan Tao[1], Chuanzan Zhou[1], Yan Shao[1], Yanling Jin[1], Meihua Ye[1], Yan Chen[1], Jong Deng [2], Jimin Shao [6], Jicheng Yue [2], Xiaju Cheng [7]✉ & Y Eugene Chinn [1,2]✉

## Abstract

**Activation of the Toll-like receptor 4 (TLR4) by bacterial endotoxins in macrophages plays a crucial role in the pathogenesis of sepsis. However, the mechanism underlying TLR4 activation in macrophages is still not fully understood. Here, we reveal that upon lipopolysaccharide (LPS) stimulation, lysine acetyltransferase CBP is recruited to the TLR4 signalosome complex leading to increased acetylation of the TIR domains of the TLR4 signalosome. Acetylation of the TLR4 signalosome TIR domains significantly enhances signaling activation via NF-κB rather than IRF3 pathways. Induction of NF-κB signaling is responsible for gene expression changes leading to M1 macrophage polarization. In sepsis patients, significantly elevated TLR4-TIR acetylation is observed in CD16+ monocytes combined with elevated expression of M1 macrophage markers. Pharmacological inhibition of HDAC1, which deacetylates the TIR domains, or CBP play opposite roles in sepsis. Our findings highlight the important role of TLR4-TIR domain acetylation in the regulation of the immune responses in sepsis, and we propose this reversible acetylation of TLR4 signalosomes as a potential therapeutic target for M1 macrophages during the progression of sepsis.**

**Keywords** Sepsis; Macrophage; TLR4-TIR Acetylation; NF-κB Signaling Pathway; HDAC1 Inhibitor
**Subject Categories** Immunology; Microbiology, Virology & Host Pathogen Interaction; Signal Transduction

## Introduction

The Toll-like receptor (TLR) family plays a crucial role in pattern recognition and signaling transduction in mammalian host defense (Fitzgerald and Kagan, 2020). Specifically, toll-like receptor 4 (TLR4) is responsible for sensing the key structural component of the gram-negative bacteria, lipopolysaccharide (LPS), which induces the assembly of a signalosome around the cytoplasmic domain of TLR4 in cells (Rathinam et al, 2019; Zanoni et al, 2011). The cytoplasmic domain of TLR4, known as the Toll-IL-1-receptor (TIR) domain, is pivotal in these processes (Luo et al, 2017). The TIR domain of TLR4 can recruit two different sets of adaptors; Myeloid differentiation factor 88 (MyD88) and MyD88 adapter-like (MAL), and TIR domain-containing adaptor-inducing IFN-β (TRIF) and TRIF-related adaptor molecule (TRAM), which couple two distinct signaling pathways respectively. MyD88, a canonical adaptor for inflammatory signaling pathways downstream of members of the TLR receptor families, tends to activate the nuclear transcription factor NF-κB (Deguine and Barton, 2014). For TLR4, MAL, also recognized as TIR domain-containing adaptor protein (TIRAP), acts as a bridge between the receptor and MyD88. The recruitment of MAL and MyD88 to the TIR domain of TLR4 facilitates the interaction of IL-1R-associated kinase (IRAK) family members and subsequent activation of TNF receptor-associated factor 6 (TRAF6) (Bovijn et al, 2012; Kim et al, 2007; Pereira et al, 2022; Ve et al, 2017). These TIR-TIR interactions and associated post-translational modifications within the complex lead to NF-κB activation eventually (Guven-Maiorov et al, 2015; New et al, 2016). MyD88-deficient mice peritoneal macrophages are defective in the activation of NF-κB in response to LPS, whereas they respond to LPS by activating IFN-regulatory factor 3 (IRF3) through a MyD88-independent pathway (Kawai et al, 2001). In the MyD88-independent pathway, the TLR4-mediated signaling pathway requires two adaptor molecules, TRAM and TRIF. TRAM is thought to bridge the activated TLR4 complex and TRIF (Carty et al, 2006). Unlike the MyD88-dependent pathway that is initiated

[1]Institute of Clinical Medicine Research, Zhejiang Provincial People's Hospital of Hangzhou Medical College, Hangzhou, China. [2]Yantai Peninsular Cancer Center, Binzhou Medical University, Yantai, China. [3]Life Science Research Institute, Zhejiang University, Hangzhou, China. [4]Institutes of Biomedical Sciences, Fudan University, Shanghai, China. [5]Department of Urology, the First Affiliated Hospital of Soochow University, Suzhou, China. [6]Department of Pathology and Pathophysiology, Key Laboratory of Disease Proteomics of Zhejiang Province, Zhejiang University School of Medicine, Hangzhou, China. [7]State Key Laboratory of Radiation Medicine and Protection, School of Radiation Medicine and Protection, and Collaborative Innovation Center of Radiological Medicine of Jiangsu Higher Education Institutions, Soochow University, Suzhou, China. [8]These authors contributed equally: Xue Li, Xiangrong Li. ✉E-mail: snowlee@zju.edu.cn; xjcheng@suda.edu.cn; chinyue@suda.edu.cn

at the plasma membranes, the MyD88-independent pathway is operational from early endosomes following endocytosis of TLR4 (Kagan et al, 2008). However, both NF-κB and IRF3 activation are essential for innate immunity through distinct sets of transcriptional activation and production of pro-inflammatory cytokine (Luo et al, 2019).

The dynamic recruitment of adaptors by TLR4 during signaling in the induction of innate immunity remains a central and largely unresolved issue. Myeloid differentiation factor 2 (MD2) is required for the activation of TLR4 in the form of heterodimer complex (Park et al, 2009). MD2 is also essential for correct intracellular distribution and LPS-recognition of TLR4. In MD2 knockout mice embryonic fibroblasts, TLR4 can't reach the plasma membrane and do not respond to LPS (Nagai et al, 2002). In response to LPS stimulation, LPS binding protein (LBP) serves for LPS capture and delivery to the pattern recognition receptor CD14 (Tobias et al, 1989). The CD14-LPS complex then binds to the TLR4-MD2 dimerization complex and initiates the cytosolic signalosome assembly (Park and Lee, 2013; Ryu et al, 2017). While CD14 serves as the primary docking site for LPS during TLR4 activation, it is reported that FcγRIII (CD16) could also associate with TLR4 in a IgG immune complexes dependent manner, however, the specific role of CD16 in this activation is still unclear (Rittirsch et al, 2009).

Post-translational modification (PTM) of protein is a powerful strategy in regulating the physicochemical properties of proteins, which provides a crucial mechanism for the dynamic regulation of TLR4 signalosome assembly and signaling transduction. However, how the recruitment of TLR4 adaptors is affected by various PTMs remains poorly investigated. Even serine/threonine kinases are implicated in TLR4 signalosome assembly (Qian et al, 2001), whether serine/threonine phosphorylation plays a role in the interaction of the TIR domains remains elusive. In contrast, tyrosine phosphorylation in the TIR domain of the TLR4–MAL–MyD88 complex and the impact on downstream signaling are relatively extensive studied (Curson et al, 2023; Gray et al, 2006; Gurung et al, 2017; Medvedev et al, 2007; Piao et al, 2008). Although blocking histone deacetylase (HDAC) activity with either HDAC inhibitors or siRNA affects TLR4 protein stability on human myeloid dendritic cells and the transcriptional regulation of downstream genes, the accurate acetylation targets in TLR4 signaling remains unknown (Halili et al, 2010; Song et al, 2011).

In this study, we found that in the stimulation of LPS, the CREB-binding protein (CBP) was recruited to the TLR4 signalosome complex, where the TIR domains of TLR4, MAL, and MyD88 were acetylated. The acetylation in the TIR domains complex emerged as a crucial factor for the TLR4/MAL/MyD88 signal pathway activation. Our investigations revealed that CBP-mediated TIR domain complex acetylation enhanced the LPS-induced polarization of M1 macrophages, resulting in pronounced production of pro-inflammatory cytokines such as IL-6 and TNF-α. Notably, in the sepsis patients, TLR4-TIR acetylation was predominantly observed in CD16+ monocytes combined with elevated expression of M1 macrophage markers. In addition, the TIR domain complex was deacetylated by Histone deacetylase 1 (HDAC1). Inhibition of HDAC1 exacerbated the M1 macrophage polarization and pro-inflammatory cytokine production leading to the progression of LPS-induced sepsis, whereas CBP inhibition alleviated these symptoms. In a word, our findings

underscored the critical role of acetylation in the TIR domain complex in regulating the inflammatory immune response, highlighting the potential of reversible acetylation as a promising therapeutic target in the context of sepsis.

## Results

### Cytoplasmic CBP induces TLR4-TIR acetylation in M1 macrophages upon LPS stimulation

In our investigation into the assembly of the TLR4 signaling complex in macrophages, we observed an elevation in TLR4 acetylation levels upon treatment with LPS, concomitant with the degradation of the inhibitor of NF-κB alpha (IκBα) (Figs. 1A and d EV1A). Notably, the LPS-induced TLR4 acetylation was further intensified by pretreatment using the histone deacetylase (HDAC) inhibitor trichostatin A (TSA) (Fig. 1B), underscoring the involvement of HDAC family members in TLR4 signal transduction. Mass spectrometry analysis revealed two lysine acetylation residues at K730 (Fig. 1C) and K810 (Fig. 1D) within the TIR domain of TLR4 in mouse M1 macrophages. Importantly, these residues demonstrated high conservation in both mouse (K730, K810) and human (K732, K813) counterparts (Fig. 1E). To facilitate the examination of the two residues acetylation, we generated specific polyclonal antibodies. It was found that acetylated antibodies at two sites only reacted to acetylated peptides at their respective sites, while not to acetylated peptides at other sites (Fig. EV1B). When the two acetylation site were mutated, acetylation antibodies cannot detect the acetylation at their respective mutant sites (Fig. EV1C). In addition, we have evaluated the specificity of TLR4-acK antibody with the TLR4 mutant mice. When the lysine residue of TLR4 was mutated to arginine, the acetylation antibody at the corresponding site could not be detected, confirming the great specificity of these antibodies (Fig. EV1D). Western blotting detection of these antibodies revealed that the peaks of TLR4 acetylation in macrophages appeared at 30 min of LPS treatment (Fig. EV1E). Moreover, the application of these acetylated antibodies can be also applied to detect the acetylation at two sites of TLR4-TIR in macrophages treated with LPS by flow cytometry (Fig. 1F,G).

To identify the acetyltransferase responsible for TLR4 acetylation, proteins co-immunoprecipitated with TLR4 after LPS treatment were subjected to mass spectrometry analysis, the result indicated that the transcription coactivators CREB-binding protein (CBP), a histone acetyltransferase, is a highly binding protein of TLR4 (Fig. 1H). Upon LPS treatment, a rapid increase in cytoplasmic CBP in macrophages was observed (Fig. 1I), indicating a dynamic shuttling between the nuclei and cytoplasm in response to LPS—a crucial prerequisite for TLR4-CBP binding. Co-transfection experiments with TLR4 and lysine acetyltransferase (KAT) members CBP, p300, or TIP60, respectively, demonstrated specific acetylation of TLR4 by CBP (Fig. EV1F). and our antibodies validated both K732 and K813 of TLR4 as specific substrate residues of CBP (Fig. 1J). And surface plasmon resonance (SPR) revealed TLR4 and CBP with an equilibrium dissociation constant (KD) of 3.51E-5 (Fig. 1K). These results suggested a close correlation between TLR4 acetylation and CBP. Conversely, depletion of CBP in macrophages dramatically reduced the induction of TLR4 acetylation by LPS (Fig. 1L), confirming the

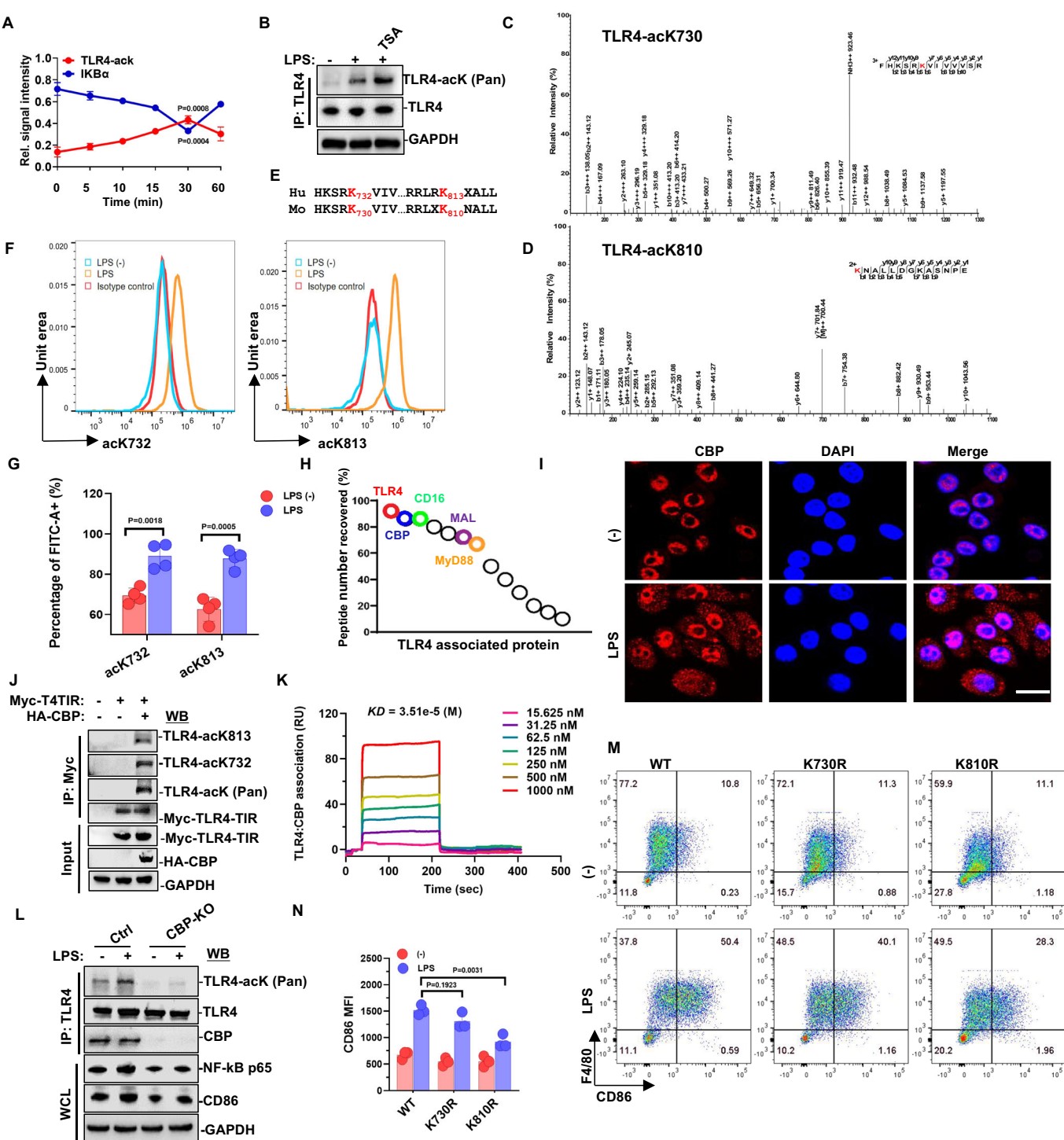

pivotal role of CBP in TLR4 acetylation induction. Furthermore, mutants TLR4-K732R or K813R in the TIR domain significantly diminished CBP-induced TLR4-TIR acetylation level (Fig. EV1G). Sequence alignment of TLR family members revealed that the TLR4-K732 site was conserved among TLR family members, while TLR4-K813 was not conserved (Fig. EV1H). Collectively, these findings suggest that CBP is a component of the LPS-induced TLR4-TIR domains.

Macrophages play a pivotal role in maintaining homeostasis, orchestrating immune responses, regulating inflammation, participating in tissue regeneration, and contributing to the resolution of inflammation (Watanabe et al, 2019). The established classification of macrophages distinguishes between classically activated M1 macrophages and alternatively activated M2 macrophages. Previous research has highlighted that the polarization of M1 macrophages is principally induced by lipopolysaccharide (LPS) and/or other

**Figure 1.   Cytosolic CBP mediates TLR4-TIR acetylation induction in M1 macrophages.**

(A) Macrophages derived from mouse bone marrow were stimulated with 100 ng/mL LPS for 0, 5, 10, 15, 30, and 60 min. Immunoprecipitation with a TLR4 antibody was followed by western blotting with pan-acetyl-K and TLR4 antibodies. Whole cell lysate (WCL) was probed with IκBα antibody. Data were obtained from three independent experiments. (B) Macrophages derived from mouse bone marrow were pretreated with 1 μM TSA for 1 h and then exposed to 100 ng/mL LPS for 30 min. Immunoprecipitation with a TLR4 antibody was followed by western blotting with pan-acetyl-K and TLR4 antibodies. (C, D) Macrophages derived from mouse bone marrow, treated with 100 ng/mL LPS for 30 min, underwent immunoprecipitation with a TLR4 antibody. Mass spectrometry analysis of TLR4 was conducted to identify the acetylation residues of TLR4-K730 (C) and TLR4-K810 (D). (E) The "HKSRKV" (K732/730) and "RRLRKA" (K813/810) motifs in the human and mouse TLR4-TIR domain were found to be highly conserved. (F, G) Macrophages derived from mouse bone marrow treated with 100 ng/mL LPS for 30 min were analyzed for the acetylation of TLR4-K813 and TLR4-K732 using flow cytometry (n = 4). (H) THP-1 cells were induced into macrophages by administering 100 ng/ml PMA, followed by treatment with 100 ng/mL LPS for 30 min. Immunoprecipitation with a TLR4 antibody was followed by mass spectrometry analysis of TLR4-associated proteins. (I) Treatment of macrophages derived from mouse bone marrow with 100 ng/mL LPS for 30 min resulted in an increase of CBP in the cytoplasm. Immunostaining of CBP (red) and the nucleus (blue) was observed by microscopy. Scale bar = 5 μm. (J) HEK293T cells were transfected with Myc-TLR4-TIR and HA-CBP. Immunoprecipitation with a Myc antibody was followed by western blotting with TLR4-acK813, TLR4-acK732, pan-acetyl-K, and Myc antibodies. (K) Surface plasmon resonance (SPR) technology was employed to detect the interaction and affinity between CBP and TLR4 proteins. (L) Macrophages with CBP knockout were treated with 100 ng/mL LPS for 30 min. CBP knockout resulted in the abolishment of TLR4 acetylation induction by LPS in macrophages. (M, N) Macrophages from TLR4-WT or TLR4-KR mutant mice bone marrow were treated with or without 100 ng/mL LPS for 12 h. Flow cytometry analysis of cells was conducted to detect the polarization of M1 macrophages by staining with anti-F4/80 (macrophage marker) and anti-CD86 (M1 marker) fluorescence antibodies (n = 3). One-way ANOVA and Student's t test. Error bars, s.e.m. Source data are available online for this figure.

stimulations, triggering pro-inflammatory responses through the activation of the NF-κB signaling pathway (Leopold Wager and Wormley, 2014). To delve into the impact of TLR4-TIR acetylation on M1 macrophage polarization, we examined the LPS-induced expression of the M1 macrophage marker CD86 in TLR4-KR mutant mice in the TIR domain compared with TLR4-wild-type (WT) mice. Intriguingly, the results revealed a significant reduction in CD86 expression in TLR4-K810R mutant mice (Fig. 1M,N). Furthermore, the knockout of CBP diminished the expression of CD86 and inhibited NF-κB activation induced by LPS (Fig. 1L). These findings underscore the regulatory role of TLR4-TIR acetylation, mediated by CBP, in modulating M1 macrophages polarization and NF-κB activation in response to LPS.

## CBP-mediated TIR domain complex acetylation promotes TLR4/MAL/MyD88 signaling pathway activation

As mentioned above, CBP promoted LPS-induced TLR4-TIR acetylation. In addition, acetylation of residues K231 and K238 in the TIR domain of MyD88 were detected in LPS-treated mouse macrophages (Fig. EV1I), and acetylation of residues K210 in the TIR domain of MAL were detected (Fig. EV1J). To elucidate the role of TIR domain complex acetylation in the formation of the TLR4 signaling pathway, we investigated the interaction between TLR4 and its adaptor proteins. The TLR4-TIR domain typically recruits four TIR domain-containing adaptors: MAL, MyD88, TRIF, and TRAM. TLR4 was originally discovered to recruit MyD88 for signaling (Wesche et al, 1997). Subsequently, MAL was reported to be recruited by MyD88 for the so-called MyD88-dependent signaling (Bernard and O'Neill, 2013). The published evidence indicates that MAL also interacts directly with TLR4 and such association between TLR4 and MAL is stronger than TLR4 and MyD88 (Ohnishi et al, 2009). Our findings revealed that treatment with TSA promoted the interaction between the TLR4-TIR and MAL (Fig. 2A) or MyD88 (Fig. 2B). As well as the endogenous interaction between TLR4 and MyD88 induced by LPS and TSA (Fig. 2C). When mouse macrophages were treated without TSA, the overlapping color showed a yellow dominance, indicating the co-localization of MAL (green) and MyD88 (red). In contrast, after TSA treatment, the overlapping color displayed a fuchsia dominance, signifying the co-localization of TLR4 (purple) and

MyD88 (red) (Fig. 2D). However, the impact on the interaction between the TLR4-TIR domain and TRIF or TRAM was relatively slight (Fig. EV2I,J).

We proceeded to investigate the impact of acetylation at the K732 and K813 residues on TLR4's interaction with adaptor proteins. Utilizing molecular docking and dynamics simulations, we examined the binding of MAL or MyD88 with TLR4. The analysis indicated that the proposed residues K732 and K813 were situated at the interaction surface of both TLR4–MAL and TLR4-MyD88 interaction models (Fig. 2E). Subsequently, molecular dynamics simulations were conducted for MAL or MyD88 with non-acetylated TLR4 and TLR4 acetylated at K732 and K813. The analysis of root mean square deviation (RMSD) revealed that the skeleton atoms of TLR4 remained in a constrained status. While TLR4-MyD88 interaction was not affected by TLR4 acetylation, the thermal dynamics of MAL increased sharply when acetylation of TLR4 was removed, implying that a stable TLR4–MAL interaction relied on the TLR4 acetylation level (Fig. EV2A,B). Accordingly, the TLR4 acetylation significantly enhanced TLR4–MAL interaction, as evidenced by increased hydrophobic interactions and hydrogen bonds (Fig. EV2C,D). Further analysis of the binding patterns of MAL or MyD88 with non-acetylated TLR4 versus TLR4 acetylated at K732 and K813 revealed that acetylated K813 formed hydrogen bonds with R207 and E211 of MAL, whereas acetylated K732 formed hydrophobic interactions with surrounding hydrophobic residues Y216 and T219 (Fig. 2F). Analogously, acetylated K813 forms hydrogen bonds with K261 of MyD88, whereas acetylated K732 forms hydrophobic interactions with surrounding hydrophobic residues T272 and V273 (Fig. 2G). In addition, we assessed the root mean square fluctuation (RMSF) and protein hydrophilic and hydrophobic surfaces, which revealed that the structure of amino acid residues at the binding interface was less flexible after TLR4 acetylation, and acetylated K732/K813 exhibited better binding in the groove of MAL and MyD88 (Fig. EV2E–H). As hydrogen bonds are stronger than hydrophobic interaction, it was suspected that the K813 acetylation may contribute more than the K732 acetylation in TLR4 adaptor binding.

To validate these findings, residues (R207 and E211 of MAL-TIR, K261 of MyD88-TIR) critical for forming hydrogen bonds with acetylated TLR4-TIR were mutated to alanine, resulting in reduced interactions between TLR4 and the adaptors (Fig. 2H,I).

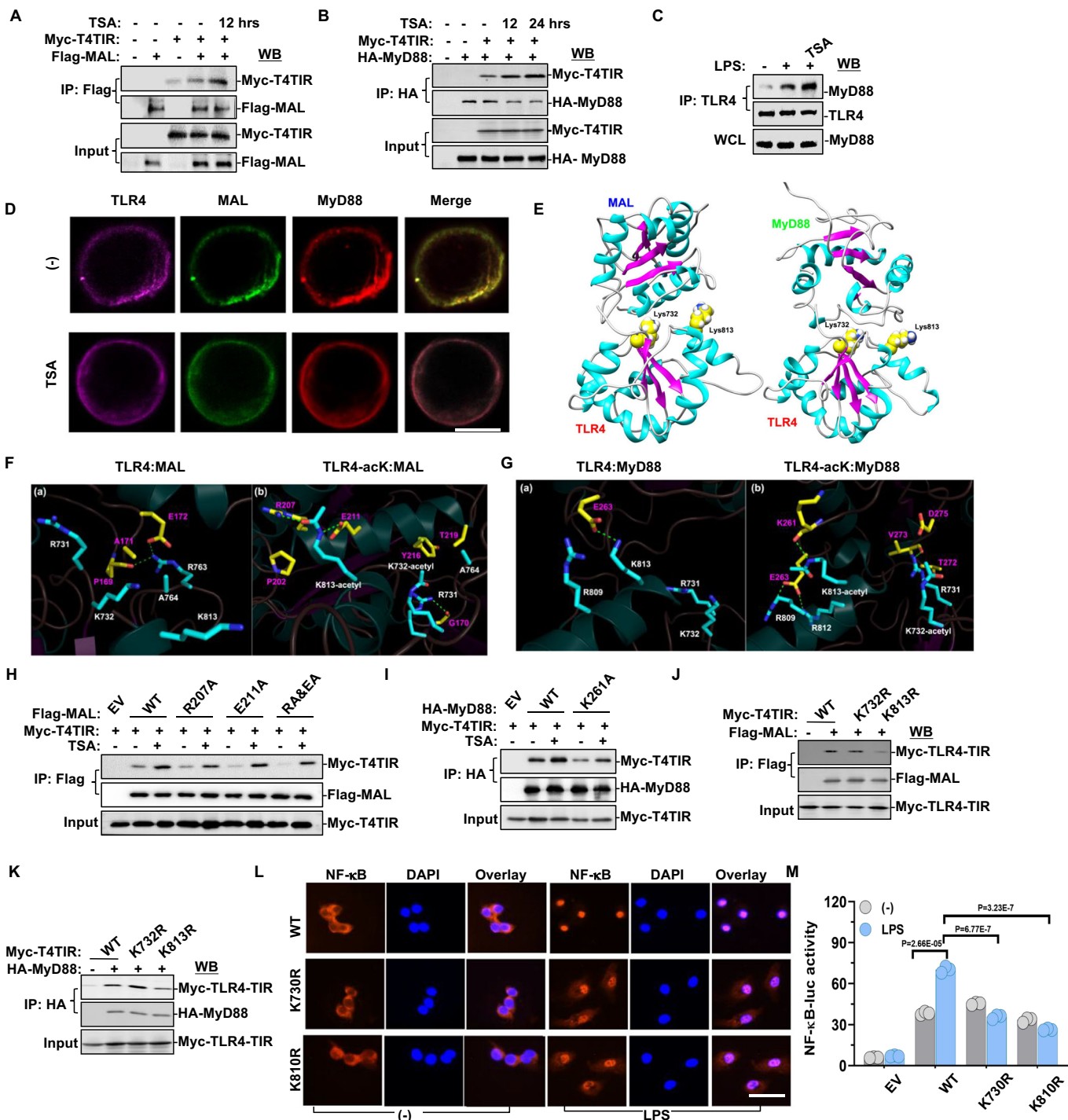

To assess the impact of K732 and K813 acetylation in the TIR domain on the TLR4 signaling complex, K732R, and K813R-TLR4 mutants were constructed to disrupt acetylation modification. The TLR4-K813R mutant significantly lost its interaction with MAL and MyD88, emphasizing the major contribution of K813 acetylation in this process, while the impact of the K732R mutation was less pronounced (Fig. 2J,K). However, it was not sure whether the mutations induced TLR4 complex dissociation resulted from improper protein folding or reduced hydrogen bonds.

In addition, a similar analysis was conducted on TLR4, TRIF, and TRAM interactions. The TIR domains of TRIF and TRAM, although similar, were less conserved compared to those of MAL and MyD88. TSA treatment slightly increased the interaction between TLR4-TIR and TRAM, but had no effect on TRIF (Fig. EV2I,J). Co-immunoprecipitation assays indicated that neither the K732R nor the K813R mutations abolished TLR4 interaction with TRIF and TRAM (Fig. EV2K,L). In summary, acetylation in the TIR domain complex enhanced the assembly of

Figure 2.  CBP-mediated TIR domain complex acetylation promotes TLR4/MAL/MyD88 signaling pathway and NF-κB activation.

(A, B) HEK293T cells were transfected with the indicated plasmids and treated with 1 μM TSA for 12 h or 24 h. Immunoprecipitation with Flag or HA antibody followed by western blot with Myc and Flag or HA antibodies revealed that TSA treatment significantly induced the interaction between TLR4 and MAL and/or MyD88. (C) Macrophages derived from mouse bone marrow treated with 1 μM TSA for 12 h followed by stimulation with 100 ng/ml LPS for 30 min displayed a significant induction of TLR4 and MyD88 interaction. (D) Immunostaining of TLR4, MAL, and MyD88 was observed by microscopy after mouse macrophages were treated with 1 μM TSA for 12 h. Scale bar = 5 μm. (E) Interaction models of MAL-TIR (PDB: 3UB2) and MyD88-TIR (PDB: 4EO7) with TLR4-TIR (aa: 670–815). Molecular docking revealed that the proposed residues of TLR4 (K732 and K813) are at the interaction surface of both MAL-TLR4 and MyD88-TLR4 interaction models. (F, G) The binding pattern of MAL or MyD88 with non-acetylated or K732/K813-acetylated TLR4 after simulation. The associated amino acids are shown as stick models and dashed green lines represent hydrogen bonds. Amino acids of MAL or MyD88 and TLR4 are shown in yellow and blue, respectively. (H) HEK293T cells, transfected with indicated plasmids and treated with 1 μM TSA for 12 h, underwent immunoprecipitation with Flag antibody followed by western blot with Flag and Myc antibodies. R207A and E211A mutation of MAL-TIR attenuated the TIR-TIR interaction of TLR4 and MAL. (I) HEK293T cells, transfected with indicated plasmids and treated with 1 μM TSA for 12 h, underwent immunoprecipitation with HA antibody followed by western blot with HA and Myc antibodies. K261A mutation of MyD88-TIR attenuated the TIR-TIR interaction of TLR4 and MyD88. (J, K) TLR4-WT or TLR4-K732R or K813R mutation was co-transfected with MAL or MyD88 in HEK293T cells. TLR4-K813R mutant significantly abolished its interaction with MAL and MyD88, while TLR4-K732R mutant had a lesser influence on their interaction. (L) Macrophages obtained from mouse bone marrow displayed immunostaining of NF-κB p65 (Red) and the nucleus (Blue) after treatment with 100 ng/mL LPS for 30 min. Scale bar = 5 μm. (M) The effect of TLR4 acetylation on NF-κB transcriptional activity regulation was tested in HeLa cells. Cells were transfected with the indicated plasmids and NF-κB luciferase reporter for 42 h, followed by treatment with 1000 ng/mL LPS for another 6 h ($n = 3$). One-way ANOVA and Student's $t$ test. Error bars, s.e.m. Source data are available online for this figure.

the TLR4/MAL/MyD88 signaling pathway, with acetylation of TLR4-K813 playing a major role in this process.

## TLR4-TIR acetylation enhances NF-κB dependent pro-inflammatory gene expression in M1 macrophages

The formation of the TLR4 signalosome complex led to the activation of several serine/threonine kinases, resulting in nuclear localization of NF-κB and IRF3 (O'Neill and Bowie, 2007). Macrophages were obtained from TLR4-KR mutant mice and TLR4-WT mice bone marrow and pretreated with 10 ng/ml m-CSF for 7 days. Compared to the readily nuclear translocation of NF-κB (p65) in TLR4-WT macrophages, TLR4-KR mutant macrophages exhibited lower nuclear translocation in response to LPS stimulation (Fig. 2L). We noticed that in the HEK293T NF-κB luciferase assay, LPS-induced NF-κB activation was significantly enhanced by CBP co-transfection (Fig. EV2M), likely due to the upregulation of TLR4-TIR acetylation by exogenous CBP. The effects of TLR4-TIR acetylation on NF-κB activation were then examined in the NF-κB luciferase assay. HeLa cells transfected with TLR4-WT exhibited pronounced NF-κB activation upon LPS treatment, while the cells transfected with TLR4-KR mutant failed to respond to LPS stimulation (Fig. 2M). However, when the IRF3 luciferase reporter assay was tested in the HeLa cells, TLR4-K730R or K810R mutants did not abolish LPS-mediated IRF3 activation (Fig. EV2N) and did not influence IFN-β1 expression (Fig. EV2O). These results indicated that TLR4-TIR acetylation induced by LPS was required for nuclear translocation and activation of NF-κB but not for IRF3.

To further investigate the biological function of TLR4-TIR acetylation, we conducted RNA-seq analysis before and after LPS treatment in peritoneal macrophages from TLR4-WT and TLR4-KR mutant mice. In the TLR4-WT macrophages, upregulation of pro-inflammatory cytokines and inflammation-related genes, such as IL-6, TNF, IL-1β, and STAT4 was observed in response to LPS treatment. In contrast, the response was attenuated in TLR4-KR macrophages, particularly in the K810R cells (Fig. 3A–F). Venn analysis showed significant gene differences in WT vs K730R or WT vs K810R with or without LPS treatment. When LPS treatment, 27 genes (16 + 11) are differentially regulated. These genes represent those influenced by LPS through the K730 site. Specifically, 16 genes are dependent on both the K730 site and

LPS stimulation, while 11 genes are differentially regulated independently of LPS stimulation. Similarly, for the K810 site, 299 genes are dependent on both the K810 site and LPS stimulation for expression, while 104 genes are differentially regulated independently of LPS stimulation. (Fig. 3G,H). Subsequently, we confirmed the data by qRT-PCR (Fig. 3I). In addition, we examined the mRNA levels of the M1 macrophages marker iNOS. The expression of iNOS was significantly upregulated by LPS in TLR4-WT but not TLR4-KR macrophages (Fig. 3I). To confirm the polarization of M1 macrophages, we further analyzed the secretion of M1 pro-inflammatory cytokines IL-6, TNF-α, IFN-γ, and IL-1β. These cytokines were significantly upregulated by LPS in TLR4-WT macrophages, showing higher levels compared to those in TLR4-KR macrophages (Fig. 3J). These results indicated that increased TLR4-TIR acetylation promoted NF-κB activation induced by LPS and thus the polarization of M1 macrophages, while the loss of TIR acetylation function in TLR4-KR mutants attenuated the polarization of M1 macrophages induced by LPS.

## TLR4-TIR acetylation promotes M1 macrophages polarization in the mouse model of sepsis

To delve into the impact of TLR4-TIR acetylation on M1 macrophages polarization, we isolated and examined the LPS-induced polarization of the M1 macrophages. Compared to the TLR4-WT group, the reduction in CD86 expression in the TLR4-KR groups was confirmed (Fig. 1M,N). Sepsis is a severe systemic inflammatory response disease usually triggered by severe pathogenic infections, the transformation of M0 macrophages into M1 macrophages played a significant role in the progression of sepsis (Wang et al, 2023). Building on previous exciting results, we further investigated the effects of TLR4-TIR acetylation on septic inflammatory immune responses in vivo. Relative to the TLR4-WT, the TLR4-KR mutants significantly improved the survival rates of LPS-induced septic mice (Fig. 4A). Histopathological analysis of the lungs and spleens of the TLR4-WT and TLR4-KR mutant mice without LPS treatment revealed clean alveolar cavity and intact spleen structure (Fig. EV3C). In addition, we found no differences in the amount of white blood cells, lymphocytes, granulocytes, and NK cells (Fig. EV3A,B), and in the expression of TLR4 in macrophages between TLR4-WT and TLR4-KR mutant

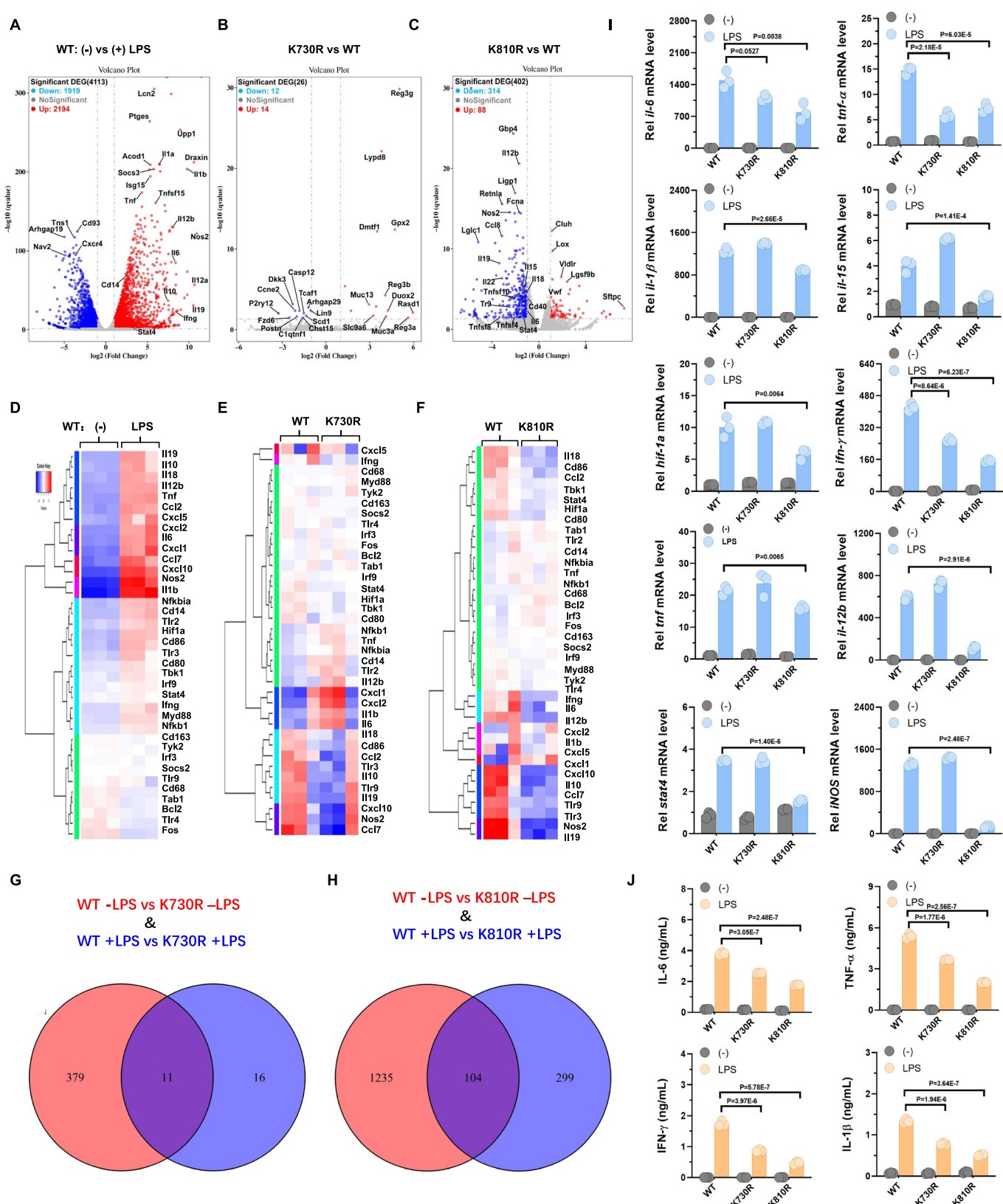

**Figure 3.   TLR4-TIR acetylation in NF-κB dependent pro-inflammatory gene expression induction.**

(A) Volcano plot illustrating upregulated and downregulated genes in LPS-treated peritoneal macrophages from TLR4-WT mice compared to mock-treated cells. (B) Volcano plot showing upregulated and downregulated genes in LPS-treated peritoneal macrophages from TLR4-K730R mutant mice relative to LPS-treated peritoneal macrophages from TLR4-WT mice. (C) Volcano plot displaying upregulated and downregulated genes in LPS-treated peritoneal macrophages from TLR4-K810R mutant mice relative to LPS-treated peritoneal macrophages from TLR4-WT mice. (D) Heatmap depicting gene expression in LPS-treated peritoneal macrophages versus mock-treated cells from TLR4-WT mice, indicating upregulation of multiple inflammatory genes after LPS treatment. (E) Heatmap illustrating gene expression in LPS-treated peritoneal macrophages from TLR4-K730R mutant mice relative to LPS-treated peritoneal macrophages from TLR4-WT mice. (F) Heatmap presenting gene expression in LPS-treated peritoneal macrophages from TLR4-K810R mutant mice relative to LPS-treated peritoneal macrophages from TLR4-WT mice. (G) Venn diagram presenting gene expression with or without LPS-treated peritoneal macrophages from TLR4-K730R mutant mice relative to LPS-treated peritoneal macrophages from TLR4-WT mice. (H) Venn diagram presenting gene expression with or without LPS-treated peritoneal macrophages from TLR4-K810R mutant mice relative to LPS-treated peritoneal macrophages from TLR4-WT mice. (I) Quantitative qRT-PCR validation of indicated pro-inflammatory genes in Macrophages from TLR4-WT or TLR4-KR mutant mice bone marrow treated with or without LPS ($n = 3$). (J) Mouse peritoneal macrophages from TLR-WT or TLR4-KR mutant mice were treated with or without 100 ng/mL LPS for 12 h. Cell supernatant samples were collected for ELISA analysis to detect the secretion factors of M1 macrophages ($n = 3$). One-way ANOVA. Error bars, s.e.m. Source data are available online for this figure.

mice (Fig. EV3D,E). However, TLR4-WT mice with LPS treatment revealed a large amount of effusion and neutrophil infiltration in the alveolar cavity, as well as unclear splenic medullary structure in the spleen, whereas, the lungs and spleens in TLR4-KR mutant mice, especially in the TLR4-K810R group, reduced (Fig. 4B). Flow cytometry analysis of peritoneal macrophages shown decreased polarization of M1 macrophages in TLR4-KR mutant mice (Fig. 4C,D), consistent with previous in vitro results. Furthermore, analysis of pro-inflammatory cytokine mRNA expression in abdominal macrophages revealed significant upregulation of IL-6 and TNF-α mRNA levels by LPS in TLR4-WT macrophages. However, a weaker mRNA upregulation trend of them was observed by LPS in TLR4-KR macrophages compared to TLR4-WT macrophages (Fig. 4E), which was further confirmed by their serum concentrations (Fig. 4F). In summary, TLR4-TIR acetylation promoted M1 macrophages polarization induced by LPS and pro-inflammatory cytokines secretion in the mouse model of sepsis.

## TLR4-TIR acetylation presented in CD16+ monocytes exhibit a pro-inflammatory M1 macrophage phenotype in sepsis patients

To further explore the role of TLR4-TIR acetylation in sepsis, we initially isolated TLR4-TIR-acetylated cells from the PBMCs enriched monocytes of sepsis patients using the specific TLR4 acetylation antibody. Subsequently, the TLR4-TIR-acK marked cells underwent single-cell sequencing analysis (Fig. 5A). Generally, human monocytes were sub-divided based on CD16 and CD14 expression into three major populations; classical (CD14+), non-classical (CD16+), and intermediate (CD14 + CD16+) (Kapellos et al, 2019; Ong et al, 2019). UMAP analysis revealed that TLR4-TIR acetylation was present in all three monocyte subsets (Fig. 5B). The CD16+ monocytes were labeled pro-inflammatory based on higher expression of pro-inflammatory cytokines and potency in antigen presentation (Ziegler-Heitbrock, 2007). Our single-cell sequencing data demonstrated that sepsis marker genes were highly elevated in CD16+ monocytes, including *NAP1L1* (Freitag and Schwertz, 2022), *CFD* (Sommerfeld et al, 2021), *BID* (Weber et al, 2008), *BCL2A1* (Li et al, 2022a), *MALAT1* (Chen et al, 2022), *RNH1* (Zechendorf et al, 2020), and *LILRA5* (Ning et al, 2023) genes (Fig. 5C,D). In addition, an increase in classical M1 macrophages markers CD68 and CD86 was detected in CD16+ monocytes (Fig. 5E). Furthermore, *IL16*, *CXCL6*, *IL15*, and *IL21R* genes were also found to be greatly elevated in CD16+ monocytes (Fig. 5F).

Monocytes were obtained by sorting with TLR4-TIR-acK and CD16 antibodies from both normal individuals and sepsis patients. The markers of M1 macrophage activation, TNF-α and IL-6, are significantly elevated in sepsis (Fig. EV4G). These data imply that these monocytes tend to differentiate into M1 macrophages during sepsis. Moreover, we found that serum levels of IL-6 and TNF-α were significantly higher in patients with septic shock compared to those with sepsis (Fig. EV4H), suggesting that IL-6 and TNF-α are closely associated with the progression of sepsis. In consistency with previous report (Kapellos et al, 2019), the KEGG enrichment scatter plot indicated that CD16+ monocytes played a significant role in pathways related to COVID-19, virus infection, influenza A, and metabolic pathways (Fig. 5G). These results shown that the TLR4-TIR acetylation was present at a high level in CD16+ monocytes, which exhibited the potency to differentiate into classical phenotype of M1 macrophages and may serve as one of the important pro-inflammatory cells in the progression of human sepsis.

In addition, TLR4-TIR acetylation of PBMCs enriched monocytes from normal individuals and sepsis patients was analyzed by flow cytometry using specific TLR4-acK732 and TLR4-acK813 antibodies. The levels of both TLR4-acK732 and TLR4-acK813 were significantly increased in sepsis patients (Figs. 6A and EV4A), as well as TLR4 expression (Fig. EV4B). Meanwhile, we found that TLR4-acK increased in all monocyte populations, however, the relative acetylation level (TLR4-acK/TLR4) was only significantly elevated in CD16+ monocytes of the sepsis patients compared to the normal individuals (Figs. 6B–E and EV4C–F). The basic information of normal individuals and sepsis patients is presented in Table 1.

## HDAC1 serves as the key deacetylase of TLR4-TIR in M1 macrophages

As mentioned above, HDAC inhibitor TSA accelerated TLR4-TIR acetylation and CBP promoted NF-κB activation and M1 macrophages polarization induced by LPS. To determine if HDAC inhibitors affect M1 macrophages marker-CD86 expression, macrophages from TLR4-WT mice bone marrow were pretreated with various HDAC family inhibitors. After pretreatment, the macrophages were exposed to LPS (Fig. EV5A) or not (Fig. EV5B). Cells were then collected to perform M1 macrophages analysis. HDAC family inhibitors JNJ-26481585, RG2833, CUDC-907, and MGCD0103 significantly promoted the expression of M1

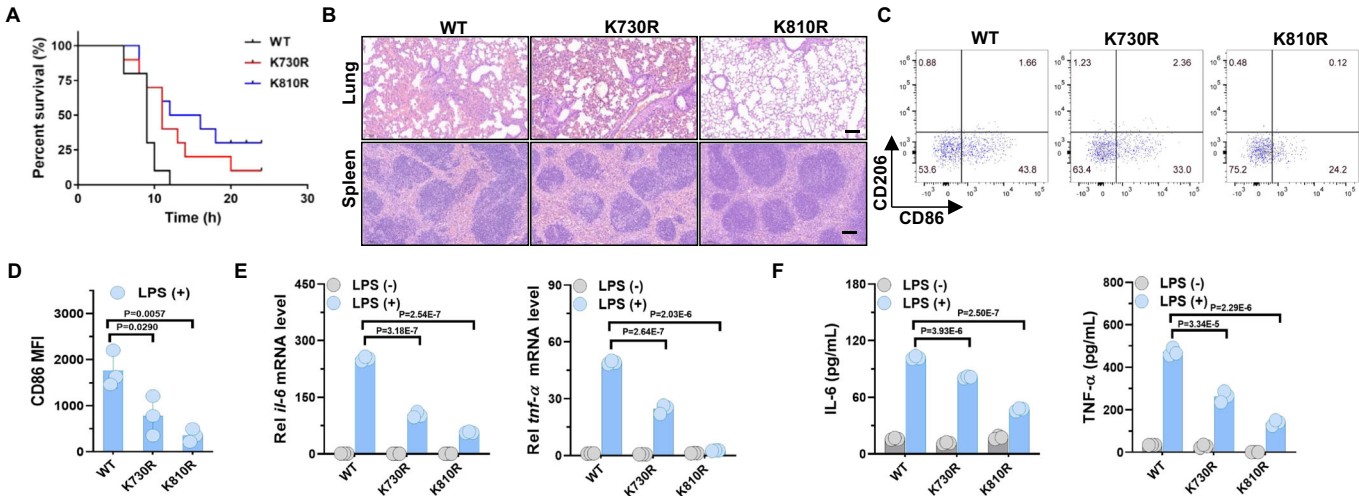

**Figure 4. TLR4-TIR acetylation promoted M1 macrophages polarization in the mouse model of sepsis.**

(A) Survival curves of TLR4-WT or TLR4-KR mutant mice after intraperitoneal injection with 25 mg/kg LPS. $n = 10$. (B) Representative H&E staining images of lung and spleen tissues isolated from TLR4-WT or TLR4-KR mutant mice after intraperitoneal injection with 25 mg/kg LPS for 4 h. Scale bar = 100 μm. (C, D) Mouse peritoneal macrophages obtained from TLR4-WT or TLR4-KR mutant mice after intraperitoneal injection with 25 mg/kg LPS for 4 h. Cells were collected for flow cytometer analysis to detect the polarization of M1 macrophages using anti-F4/80 (macrophage marker), anti-CD86 (M1 marker), and anti-CD206 (M2 marker) fluorescence antibodies ($n = 3$). (E) Quantitative qRT-PCR validation for IL-6 and TNF-α in mouse peritoneal macrophages obtained from TLR4-WT or TLR4-KR mutant mice after intraperitoneal injection with 25 mg/kg LPS for 4 h ($n = 3$). (F) The secretion of IL-6 and TNF-α in the serum collected from LPS-treated TLR4-WT or TLR4-KR mutant mice for 4 h, detected by ELISA assay ($n = 3$). Log-rank test. One-way ANOVA. Error bars, s.e.m. Source data are available online for this figure.

macrophages marker CD86 in response to LPS treatment (Fig. EV5C). For all these inhibitors, HDAC1 was the common target, suggesting HDAC1 could be the deacetylase of TLR4-TIR (Fig. EV5D). The dosages and targets of the HDAC family inhibitors were shown in "small molecule inhibitor" method. In addition, when HEK293T cells were transfected with TLR4-TIR, CBP, and a series of HDAC family plasmids, it was found that the TLR4-TIR domain was efficiently deacetylated by the expression of HDAC1 (Fig. EV5E). Knockdown of HDAC1 by macrophages significantly promoted acetylation of two TLR4-TIR domain sites (Fig. EV5F).

As mentioned above, CD16+ monocytes tended to differentiate to M1 macrophages in sepsis, therefore, we referred to this group of cells as CD16 + M1 macrophages. Then, we simply verified the functionality of CBP/HDAC1 system in human CD16 + M1 macrophages. It was observed that CBP was significantly enriched in the cytoplasm of macrophages isolated from sepsis patients (Fig. 6F), suggesting its potential role in promoting high levels of TLR4-TIR acetylation in human CD16 + M1 macrophages. In addition, the expression of HDAC1 and HDAC3 was significantly reduced in CD16 + M1 macrophages from sepsis patients compared to that of normal individuals, particularly the HDAC1 (Fig. 6G,H). Overall, these findings indicated that CBP can promote TLR4-TIR acetylation, while HDAC1 may also serve as the deacetylase of TLR4-TIR in human CD16+ monocytes during sepsis.

## Targeting TLR4-TIR acetylation shows promise for the treatment of sepsis in the mouse model

Given the role of TLR4-TIR acetylation by CBP and HDAC1 in LPS-induced NF-κB activation, we conducted further investigations

into the impact of CBP inhibitor and HDAC1 inhibitor on the polarization of M1 macrophages and the progression of sepsis. Initially, we treated macrophages from TLR4-WT mice with CBP inhibitors SGC-CBP30 and HDAC inhibitors TSA prior to LPS treatment and analyzed the efficiency of M1 macrophage polarization using flow cytometry. The results revealed a decrease in the polarization of M1 macrophages in the SGC-CBP30 group and an increase in the TSA group (Fig. 7A,B). In addition, the mRNA level of the M1 marker iNOS was confirmed using qRT-PCR, which showed a decrease in the SGC-CBP30 group and an increase in the TSA group (Fig. 7C). Furthermore, LPS-induced TLR4 acetylation in macrophages was significantly reduced after treatment with SGC-CBP (Fig. 7D).

Subsequently, we investigated the impact of CBP inhibitor and HDAC1 inhibitor on the progression of LPS-induced sepsis by intraperitoneally injecting TLR4-WT mice with SGC-CBP30, TSA, and HDAC1 inhibitor RG2833. The survival of mice was monitored every 2 h for 48 h. It was found that under LPS treatment, SGC-CBP30 significantly improved the survival rate of mice, while RG2833 decreased the survival rate of mice (Fig. 7E). Histopathological analysis of the lungs and spleens of mice without LPS treatment revealed clean alveolar cavities and intact spleen structures (Fig. EV3F). In contrast, the analysis of lung and spleen tissues from the TSA and RG2833 groups of TLR4-WT mice showed that LPS-induced injuries were worsening. However, the injuries were ameliorated in the SGC-CBP30 group (Fig. 7F). In addition, the flow cytometry analysis revealed a decrease in the polarization of M1 macrophages in the SGC-CBP30 group and an increase in the RG2833 and TSA groups compared to the LPS alone group (Fig. 7G,H). TSA and RG2833 also showed similar results in TLR4-K810R mice (Fig. EV3G–K). We suspect that it

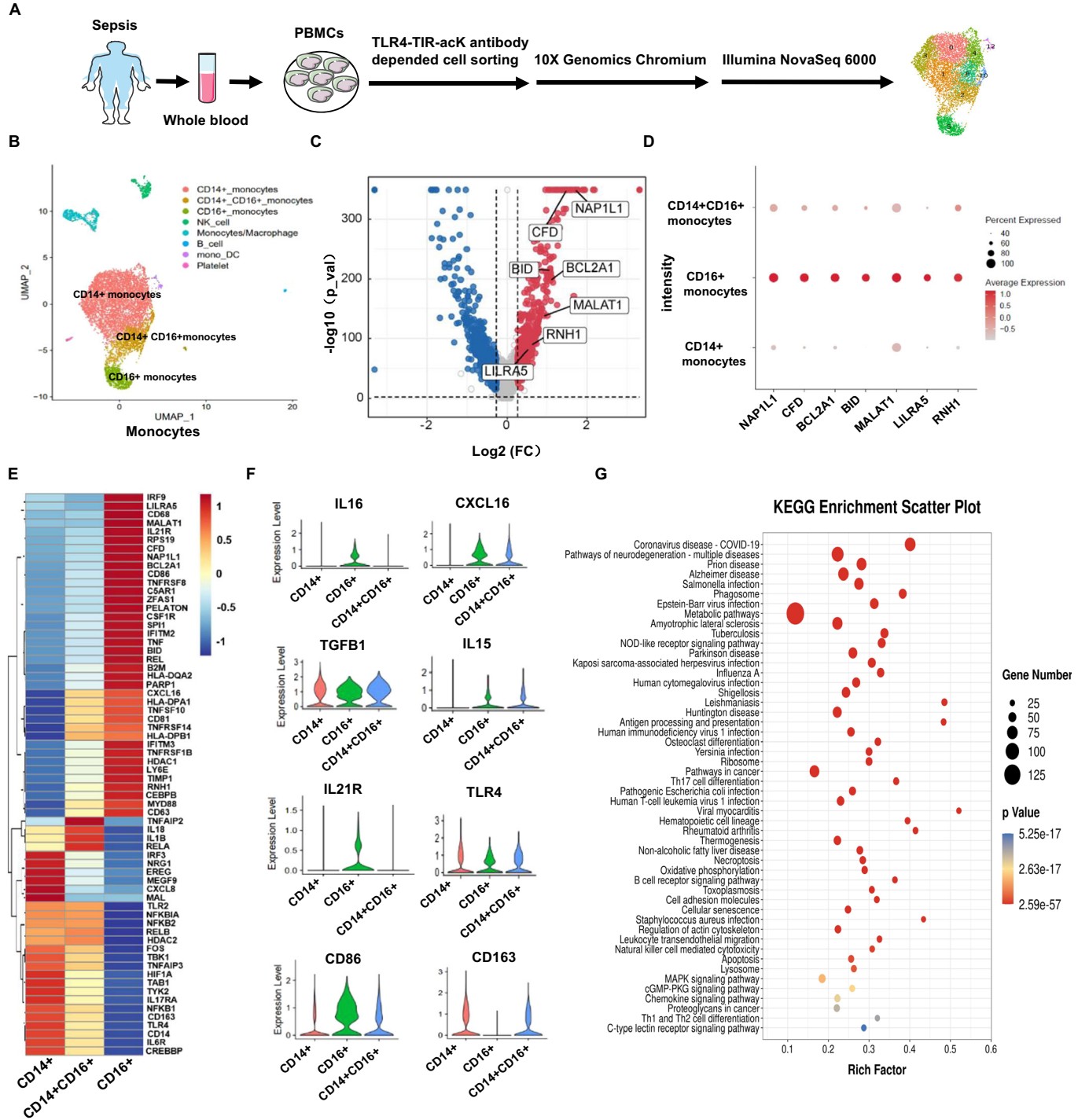

**Figure 5. CD16+ monocytes exhibit a pro-inflammatory M1 macrophages phenotype in sepsis.**

(A) Schematic diagram outlining the experimental workflow for the entire study. Peripheral blood mononuclear cells (PBMCs) were isolated from human blood, then TLR4-acK antibody positively stained monocytes were sorted and used for the construction of single-cell RNA-Seq cDNA libraries. At last, the libraries were sequenced on an Illumina NovaSeq 6000 sequencing system and analyzed as described in the methods section. (B) UMAP visualization depicting cell types based on the expression of known marker genes. (C) Volcano plot illustrating gene expression across cell types, highlighting genes upregulated in CD16+ cells (red) or upregulated (green) in CD14+ types cells. Bimod test. (D) Sepsis-specific genes identified in three types of cells, with CD16+ monocytes being the major producer of these factors. (E) Heatmap displaying differentially expressed genes in three types of cells. (F) Gene expression levels measured in three types of cells, emphasizing CD16+ monocytes exhibit a pro-inflammatory M1 macrophages phenotype in sepsis (n = 1). (G) KEGG enrichment scatter plot for pathway names in CD16+ monocytes. Fisher's Exact test.

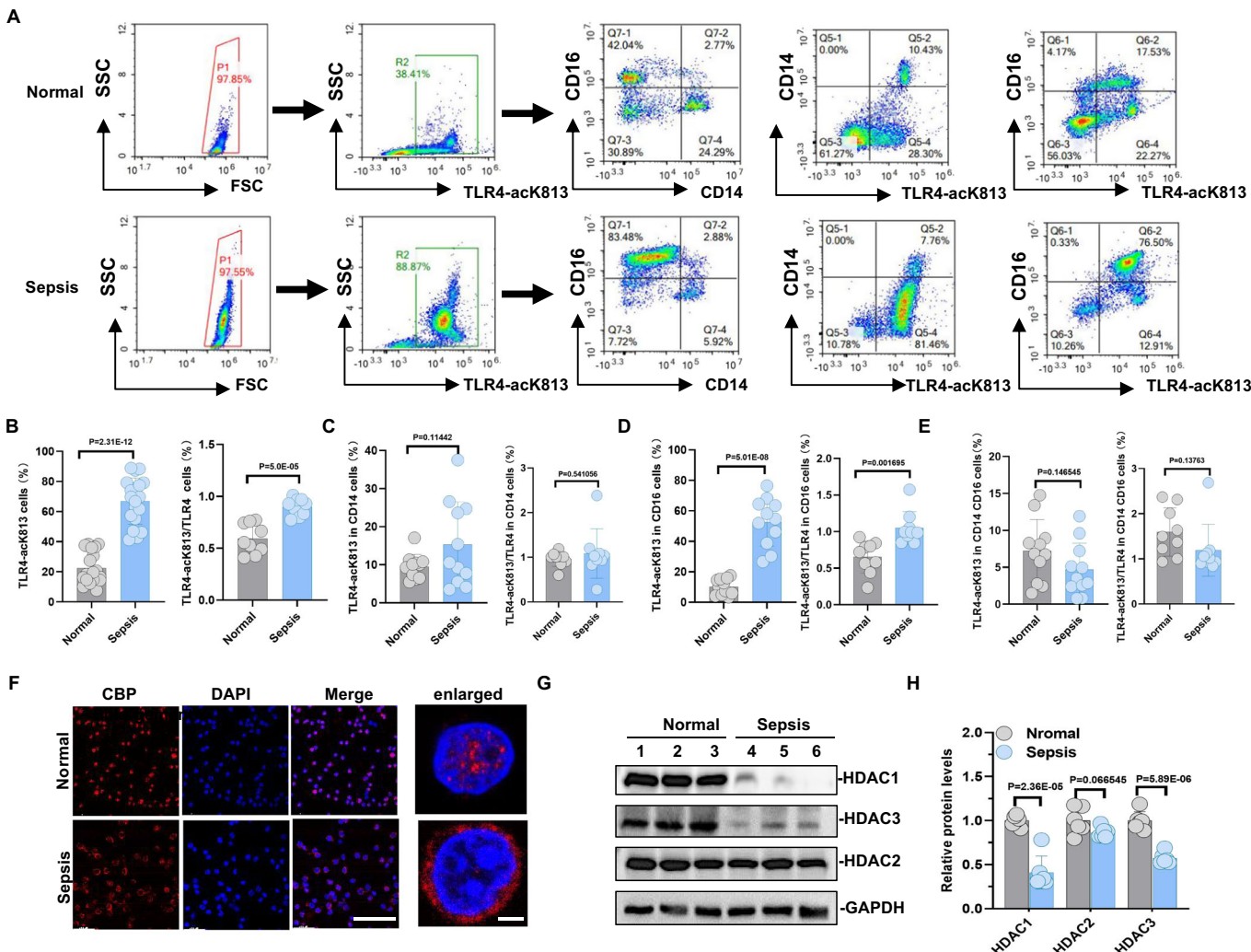

**Figure 6. TLR4 acetylation predominantly occurs in CD16+ monocyte during sepsis.**

(A) Monocytes were collected for flow cytometric analysis to detect the polarization of TLR4-acK813 in CD14 + , CD16 + , and CD14 + CD16+ types cells. (B) Monocytes were obtained for flow cytometric analysis to determine the ratio of TLR4-acK813 (n = 19) and TLR4-acK813/TLR4 (n = 9) in both normal and sepsis patients. (C) Flow cytometric analysis was performed on monocyte to assess the ratio of TLR4-acK813 (n = 11) and TLR4-acK813/TLR4 (n = 9) in normal and sepsis patient CD14+ type cells. (D). Flow cytometric analysis was conducted on monocytes to evaluate the ratio of TLR4-acK813 (n = 11) and TLR4-acK813/TLR4 (n = 9) in normal and sepsis patient CD16+ type cells. (E) Monocytes were obtained for flow cytometric analysis to determine the ratio of TLR4-acK813 (n = 11) and TLR4-acK813/TLR4 (n = 9) in normal and sepsis patient CD14 + CD16+ type cells. (F). Sepsis patients exhibited an increase in cytoplasmic CBP (n = 5 patients and 5 normal controls). Immunostaining of CBP (red) and nucleus (blue) was observed by microscopy. Scale bar = 30 μm and 1 μm. (G, H) CD16+ types cells were obtained from normal and sepsis patients and subjected to western blot analysis with HDAC1 antibody (G). Densitometry measurements were performed using ImageJ (H) (n = 6). Student's t test. Error bars, s.e.m. Source data are available online for this figure.

may be linked to the acetylation of the TLR4-K732 site or to factors unrelated to TLR4 acetylation in TLR4-K810R mutant mice. Besides TLR4, several genes, including IFIH1 and STAT1, can affect M1 macrophage polarization and contribute to the progression of sepsis (Wang et al, 2023). The activation of these genes might also be influenced by acetylation. However, the CD86 expression of peritoneal macrophage (Fig. EV3J–K) and tissues injury (Fig. EV3I) have no significant difference between the control and SGC-CBP30 groups. It may be due to that TLR4-K810R mice naturally exhibit the reduced sepsis symptoms and M1 macrophage activity. Therefore, the therapeutic effect of SGC-CBP30 on TLR4-K810R mice is unconspicuous. Moreover, the

analysis of pro-inflammatory cytokine mRNA expression in macrophages shown that both IL-6 and TNF-α mRNA levels were significantly upregulated by LPS combined with RG2833 or TSA but were downregulated in combination with SGC-CBP30 (Fig. 7I). This was further confirmed by measuring serum pro-inflammatory cytokine concentration using ELISA, which demonstrated substantial suppression of IL-6, TNF-α, and IL-1β levels in the LPS + SGC-CBP30 group compared to LPS alone and LPS combined with RG2833 or TSA group (Fig. 7J). Taken together, CBP activates the TLR4/MAL/MyD88 pathway by promoting acetylation of TIR domain complex in M1 macrophages. This process leads to the release of pro-inflammatory cytokines and

**Table 1. Baseline and disease characteristics in normal and patients.**

| | Normal (n = 43) | Sepsis (n = 35) | Septic shock (n = 8) |
|---|---|---|---|
| **Characteristics** | | | |
| Age, years | 55.76 ± 6.71 | 57.71 ± 11.11 | 67.5 ± 12.66 |
| Sex, female, n % | 21 (48.8) | 20 (57.1) | 3 (37.5) |
| Body mass index (kg/m²) | 22.65 ± 1.41 | 22.51 ± 2.18 | 22.28 ± 1.38 |
| Vasoactive drug therapy, n % | – | 10 (28.6) | 8 (100) |
| Mechanical ventilation, n % | – | 15 (42.9) | 8 (100) |
| SOFA, median (IQR) | – | 7.21 (5–10) | 9.75 (8–11) |
| C-reactive protein, mg/L, median (IQR) | – | 101 (15–270) | 171 (61–313) |
| Procalcitonin, ng/ml, median (IQR) | – | 9.0 (0.15–30.83) | 38.2 (5–95.4) |
| **Baseline comorbidities** | | | |
| Hypertension, n % | 16 (37.2) | 16 (45.7) | 4 (50) |
| Diabetes mellitus, n % | 7 (16.2) | 5 (14.3) | 3 (37.5) |
| Coronary artery disease, n % | 4 (9.3) | 2 (5.7) | 2 (25) |
| Liver cirrhosis | 0 | 0 | 0 |
| Malignant tumors | 0 | 0 | 0 |
| Recent surgery | 0 | 0 | 0 |

worsens sepsis progression, whereas HDAC1 exerts the opposite effect. Therefore, CBP inhibitor SGC-CBP30 can alleviate inflammatory injury in lung and spleen tissues during sepsis by reducing the pro-inflammatory cytokine production and polarization of M1 macrophages.

## Discussion

TLR4 plays a pivotal role in host defense by detecting the LPS component within the cell wall of gram-negative bacteria, thereby initiating the innate immune response (Geng et al, 2021; Krishnan et al, 2022). Upon LPS stimulation, TLR4-mediated cytoplasmic signaling pathways can be distinguished into MyD88-dependent and independent pathways based on distinct TIR-TIR domain interactions (Fitzpatrick et al, 2020; Jiang et al, 2020). In the MyD88-dependent pathway, TLR4 recruits MAL and MyD88 adaptors (Hu et al, 2020). whereas in the MyD88-independent pathway, TLR4 recruits TRIF and TRAM adaptors (Deveci Ozkan et al, 2020; Guney Eskiler et al, 2019). However, the precise mechanism by which TLR4 dynamically recruits these adaptors during innate immunity induction remains largely unresolved. Recent research has highlighted post-translational modifications as pivotal players in TLR4 cytoplasmic signaling cascades during

innate immune responses. These modifications include phosphorylation, ubiquitination, glycosylation, and acetylation (Hu et al, 2013; Medvedev et al, 2007; New et al, 2016; Qian et al, 2001). Among them, lysine acetylation, a reversible modification, is increasingly acknowledged as a key regulator of inflammatory gene expression (Gil et al, 2017).

In our study, we have demonstrated that the cytoplasmic TLR4-TIR domain in macrophages undergoes acetylation by CBP upon stimulation with LPS. This acetylation is likely attributed to the translocation of CBP from the nucleus to the cytoplasm in response to LPS. We have identified two acetylation sites at residues K732 and K813 of the human TLR4-TIR, corresponding to K730 and K810 in mouse TLR4-TIR. Besides, acetylation sites in the MyD88 and MAL-TIR domains were also detected, which was not concretely discussed here. Interestingly, the TIR domain complex acetylation significantly enhanced TLR4's ability to recruit TIR connector proteins such as MAL and MyD88, but not TRIF and TRAM. However, we are unsure which factors are related to this situation. Is it determined by the specificity of the two acetylation sites of TLR4-TIR, or by CD14, which transfers LPS to the TLR4-MD2 dimerization complex? (Park and Lee, 2013; Ryu et al, 2017). We observed significant acetylation of TLR4-TIR in monocytes, particularly in CD16+ monocytes, in patients with sepsis. Genes related to the progression of sepsis are highly expressed in CD16+ monocytes, indicating a close correlation between these cells and the development of sepsis. As CD16 associated with TLR4 in response to IgG immune complexes (Rittirsch et al, 2009), it was supposed that CD16 association has a potential impact on LPS-induced TLR4 signalosome assembly and downstream signaling cascade or IgG immune complexes could also promote polarization of M1 macrophages. In addition, M1 macrophages markers such as CD86, TNF-α and IL-6 were also highly present in CD16+ monocytes, suggesting that CD16+ monocytes in sepsis patients were the cells that differentiated into M1 macrophages eventually, which was in concert with previous report that in human monocytes differentiation, CD14+ classical monocytes leave bone marrow and differentiate into CD14 + CD16+ intermediate monocytes and sequentially to CD16+ non-classical monocytes in peripheral blood circulation, then CD16+ monocytes invade in specific tissue locations and develop into mature tissue macrophages (Zawada et al, 2012; Ziegler-Heitbrock et al, 1993), in another word, the expression of CD14 decreases with macrophages differentiation. In our analysis, increased TLR4 expression was noted in sepsis patients, it was supposed that increase in CD14 expression should occur in LPS stimulated macrophages.

In summary, we have identified that TIR domain complex in macrophages undergoes acetylation by CBP in response to LPS stimulation, which preferentially promoted the TLR4/MAL/ MyD88 signal pathway. In monocytes from sepsis patients, acetylation of TLR4-TIR was significantly enhanced specially in CD16+ monocytes that had the potency to differentiate into M1 macrophages and acted as the major pro-inflammatory cells in the progression of sepsis. Overall, our findings highlight the crucial role of TIR domain complex acetylation activating TLR4/MAL/MyD88 signal pathway in the regulation of inflammatory immune response, and suggest that the inhibition of acetylation could serve as a novel therapeutic target for patients with infectious and autoimmune diseases.

# Methods

### Reagents and tools table

| Reagent/resource | Reference or source | Identifier or catalog number |
|---|---|---|
| **Experimental models** | | |
| HEK293T | ATCC | CRL-3216 |
| HeLa | ATCC | CCL-2 |
| THP-1 | ATCC | TIB-202 |
| C57BL/6 (Mus musculus) | Shanghai Bangyao Biotechnology | A20230401001 |
| TLR4-K730R (Mus musculus) | Shanghai Bangyao Biotechnology | A20230401001 |
| TLR4-K810R (Mus musculus) | Shanghai Bangyao Biotechnology | A20230401001 |
| sepsis patients and healthy individuals | Zhejiang Provincial People's Hospital | QT2023206 |
| **Recombinant DNA** | | |
| Myc-TLR4-TIR (homo sapiens) | This study | |
| HA-CBP (homo sapiens) | This study | |
| Flag-MAL (homo sapiens) | This study | |
| HA-MyD88 (homo sapiens) | This study | |
| Myc-TLR4-TIR-K732R (homo sapiens) | This study | |
| Myc-TLR4-TIR-K813R (homo sapiens) | This study | |
| Flag-MAL-R207A (homo sapiens) | This study | |
| Flag-MAL-E211A (homo sapiens) | This study | |
| Flag-MAL-RA&EA (homo sapiens) | This study | |
| HA-MyD88-K261A (homo sapiens) | This study | |
| **Antibodies** | | |
| Rabbit-anti-CBP | Cell Signaling Technology | #7389 |
| Rabbit-anti- Pan-acetyl-K | Cell Signaling Technology | #9441 |
| Mouse-anti- Myc | Cell Signaling Technology | #2276 |
| Rabbit-anti-HA | Cell Signaling Technology | #5017 |
| Rabbit-anti-MyD88 | Cell Signaling Technology | #4283 |
| Rabbit-anti- NF-κB p65 | Cell Signaling Technology | #8242 |
| Rabbit-anti- IRF3 | Cell Signaling Technology | #4302 |
| Mouse-anti- IκBα | Cell Signaling Technology | #4814 |
| Rabbit-anti- CD86 | Cell Signaling Technology | #19586 |
| Rabbit-anti- GAPDH | Cell Signaling Technology | #5174 |
| Rabbit-anti- TLR4 | Thermo | 48-2300 |
| APC- anti-mouse F4/80 | Biolegend | #123116 |
| PE- anti-mouse CD86 | Biolegend | #105007 |
| PE/Cy7- anti-mouse CD206 | Biolegend | #141719 |
| FITC anti-rabbit IgG | Biolegend | #406403 |
| APC anti-human CD14 | Biolegend | #399206 |
| PE anti-human CD16 | Biolegend | #302008 |

| Reagent/resource | Reference or source | Identifier or catalog number |
|---|---|---|
| **Oligonucleotides and other sequence-based reagents** | | |
| PCR primers | | |
| GAPDH (Mus musculus) F: GGTGAAGGTCGGTGTGAACG R: CTCGCTCCTGGAAGATGGTG | Accurate Biology | |
| Il6 (Mus musculus) F: TAGTCCTTCCTACCCCAATTTCC R: TTGGTCCTTAGCCACTCCTTC | Accurate Biology | |
| tnfα (Mus musculus) F: CCCTCACACTCAGATCATCTTCT R: GCTACGACGTGGGCTACAG | Accurate Biology | |
| iNOS (Mus musculus) F: CCAAGCCCTCACCTACTTCC R: CTCTGAGGGCTGACACAAGG | Accurate Biology | |
| il15 (Mus musculus) F: ATTCTCTGCGCCCAAAAGAC R: GTGGATTCTTTCCTGACCTCT | Accurate Biology | |
| il1β (Mus musculus) F: CTGCAGCTGGAGAGTGTGGAT R: CTCCACTTTGCTCTTGACTTCTATCTT | Accurate Biology | |
| il12b (Mus musculus) F: GCCACGGTCATCTGCCGCAA R: GGGCACAGATGCCCATTCGCT | Accurate Biology | |
| ifnγ (Mus musculus) F: ACAGCAAGGCGAAAAAGGATG R: TGGTGGACCACTCGGATGA | Accurate Biology | |
| hif1α (Mus musculus) F: ACCTTCATCGGAAACTCCAAAG R: ACTGTTAGGCTCAGGTGAACT | Accurate Biology | |
| stat4 (Mus musculus) F: CCTGGGTGGACCAATCTGAA R: CTCGCAGGATGTCAGCGAA | Accurate Biology | |
| ifnβ1 (Mus musculus) F: CAGCTCCAAGAAAGGACGAAC R: GGCAGTGTAACTCTTCTGCAT | Accurate Biology | |
| **Chemicals, enzymes, and other reagents** | | |
| human recombinant TLR4 protein | Abcam | ab233665 |
| human recombinant CBP protein | activemotif | 31590 |
| JNJ-26481585 | MedChemExpress | HY-15433 |
| CAY-10683 | MedChemExpress | HY-N0931 |
| BG45 | MedChemExpress | HY-18712 |
| RG2833 | MedChemExpress | HY-16425 |
| ABR-215050 | MedChemExpress | HY-10528 |
| LMK-235 | MedChemExpress | HY-18998 |
| HPOB | MedChemExpress | HY-19747 |
| TMP195 | MedChemExpress | HY-18361 |
| PCI34051 | MedChemExpress | HY-15224 |
| TMP269 | MedChemExpress | HY-18360 |
| CUDC-907 | MedChemExpress | HY-13522 |
| MGCD0103 | MedChemExpress | HY-12164 |
| SGC-CBP30 | MedChemExpress | HY-15826 |
| TSA | MedChemExpress | HY-15144 |
| **Software** | | |
| flowjo | | https://www.flowjo.com/solutions/flowjo/downloads |
| snapgene | | https://www.snapgene.cn/ |
| **Other** | | |
| Pan Monocyte Isolation Kit | Miltenyi Biotech | 130-096-537 |

## Methods and protocols

### Mice

The TLR4-K730R mutant and TLR4-K810R mutant mice were generated at C57BL/6 background by Shanghai Bangyao Biotechnology Co. Ltd, employing the CRISPR/Cas9 technique. In general, paired-gRNA was designed around the targeted exon 3 of TLR4. Both the dCas9 mRNA and the gRNAs were prepared by in vitro transcription using the mMESSAGE mMACHINE T7 ULTRA kit (Life Technologies), purified using the MEGA clear kit (Life Technologies) and eluted in RNase-free water. The dCas9 mRNA, gRNAs and synthesized DNA oligoes containing the mutation codon (K730R: AAG—CGC; K810R: AAA--CGA), respectively, were mixed and injected into mouse ES cells using a FemtoJet microinjector (Eppendorf). The injected ES cells were cultured in KSOM medium to the 2-cell stage by 1.5 days. Thereafter, ten to fifteen ES cells were injected into the 4- and 8-cell blastocyst embryos, and injected blastocysts were cultured in KSOM for 1 to 2 h until the blastocyst cavity was recovered. Healthy blastocysts were surgically transferred to the uterine horns of 2.5 dpc pseudopregnant feMALes. Finally, the chimeric mice (F0) were crossed with C57BL/6 mice, and the genotypes of the mutant mice were determined by PCR of genomic DNA extracted from toes and confirmed by Sanger sequencing. The mice were housed in a Specific Pathogen-Free (SPF) experimental aniMAL room under standard conditions ($25 \pm 2°C$, $60 \pm 10\%$ relative humidity) with a 12-h light/12-h dark cycle. All animal experiments were conducted in accordance with the approved protocol by Zhejiang Provincial People's Hospital (Number: A20230401001).

### Cell lines

The HEK293T(Number:CRL-3216) and HeLa (Number:CCL-2)cell lines were cultured in high-glucose DMEM medium supplemented with 10% fetal bovine serum (FBS) from Gemini, Germany. THP-1 cells were cultured in RPMI-1640, also supplemented with 10% FBS. THP-1 cells (Number:TIB-202) were treated with phorbol 12-myristate 13-acetate (PMA, 100 ng/ml, Sigma-Aldrich, Cat#: P8139) for 48 h to induce macrophage differentiation. The cell lines were purchased from the ATCC.

### Primary macrophages collection

Primary mouse peritoneal macrophages were obtained according to a previously described method (Davies and Gordon, 2005). Briefly, mice were intraperitoneally injected with 2 mL of 4% thioglycolate medium for 72 h. Peritoneal lavage was performed using 5 mL of ice-chilled PBS, and the collected cells were isolated. Bone marrow-derived macrophages (BMDMs) were obtained by flushing the tibia and femur of mice, followed by culturing the cells in RPMI-1640 medium containing 10% FBS and supplemented with 30% L929 supernatant containing macrophage colony-stimulating factor or 10 ng/ml m-CSF (MCE, Cat#: P7085) for 7 days. All complete cell culture media were supplemented with 100 U/mL penicillin and 100 ng/ml streptomycin.

### Plasmids

The Human TLR4 cytoplasmic TIR domain (TLR4-TIR), CBP, and MyD88, obtained through RT-PCR subcloning, were incorporated into the pcDNA3.1 eukaryotic expression vector with a Myc or HA tag at the C-terminal end. Full-length TLR4, TRIF, TRAM, and MAL, also obtained through RT-PCR subcloning, were integrated into the p3×Flag-CMV-10 vector. Site-directed mutagenesis of Flag-TLR4 and Myc-TLR4-TIR was conducted using the QuickChange XL site-directed mutagenesis kit (Stratagene, USA). The authenticity of all cDNA constructs employed in this project was confirmed through DNA sequencing. The transfection of all plasmids into the cell lines was carried out using Lipofectamine 2000 reagent, following the manufacturer's instructions (Invitrogen, USA).

### Antibodies

Polyclonal or monoclonal antibodies against CBP (#7389), Pan-acetyl-K (#9441), Myc (#2276), HA (#5017), MyD88 (#4283), NF-κB p65 (#8242), IRF3 (#4302), IκBα (#4814), CD86 (#19589), and GAPDH (#5174) were procured from Cell Signaling Technology. The Anti-Flag M2 monoclonal antibody (#F3165) was acquired from Sigma. The Anti-TLR4 antibody (ab13556) was obtained from Abcam, and the Anti-TLR4 antibody (48-2300) was sourced from Thermo. APC-conjugated anti-mouse F4/80 (#123116), PE-conjugated anti-mouse CD86 (#105007), PE/Cy7-conjugated anti-mouse CD206 (#141719), FIT-conjugated anti-rabbit IgG (#406403), APC anti-human CD14 (#399206), and PE anti-human CD16 (#302008) were purchased from Biolegend.

Synthesize acetylated TLR4-K732 and TLR4-K813 peptides. The TLR4-K732 peptide sequence was EGFHKSRK$_{ac}$VIV, and the TLR4-K813 peptide sequence was GRHIFWRRLRK$_{ac}$AL. After immunizing with conjugated antigens to obtain antiserum. After immunizing with conjugated antigens to obtain antiserum, pour the serum into a non-modified peptide-coupled chromatography column and collect the flow-through. Next, pour the collected serum into a modified peptide-coupled chromatography column. Wash the modified peptide-coupled chromatography column with PBS. Add 2 ml of glycine (200 mM, pH 2.5) to the column, mix the solution in the collection tube, and add 5 ml of glycerol. Wash the column with 20% ethanol, then add 20% ethanol to preserve the antibodies. The specificity of polyclonal antibodies targeting the acetylation residues TLR4-K732 and TLR4-K813 was assessed using synthesized peptides as antigens in a dot blot analysis. These two antibodies were mainly used for Immunoprecipitation and flow cytometry experiments.

### Immunoprecipitation and western blot

Cells were lysed using freshly prepared RIPA buffer containing 50 mM pH 7.4 Tris, 150 mM NaCl, 1% Triton X-100, and protease inhibitors. Subsequently, the cell lysates were incubated with the specified antibody and Protein A/G agarose beads overnight. The immunocomplexes were washed, boiled, fractionated by SDS-PAGE, and then transferred to a nitrocellulose membrane. Following this, the membranes were blocked with 5% BSA for 1 h, followed by incubation with the indicated primary antibody for 2 h and the secondary antibody for 1 h at room temperature. Finally, western blot results were scanned and analyzed using the LI-COR Odyssey system (Biosciences, USA).

### Histology and immunostaining

Fresh lungs and spleens were harvested and fixed overnight at room temperature in 4% formaldehyde. Subsequently, the tissues were processed, embedded, and sectioned into 4-μm slices. Histological examination was performed using hematoxylin and eosin. For cell

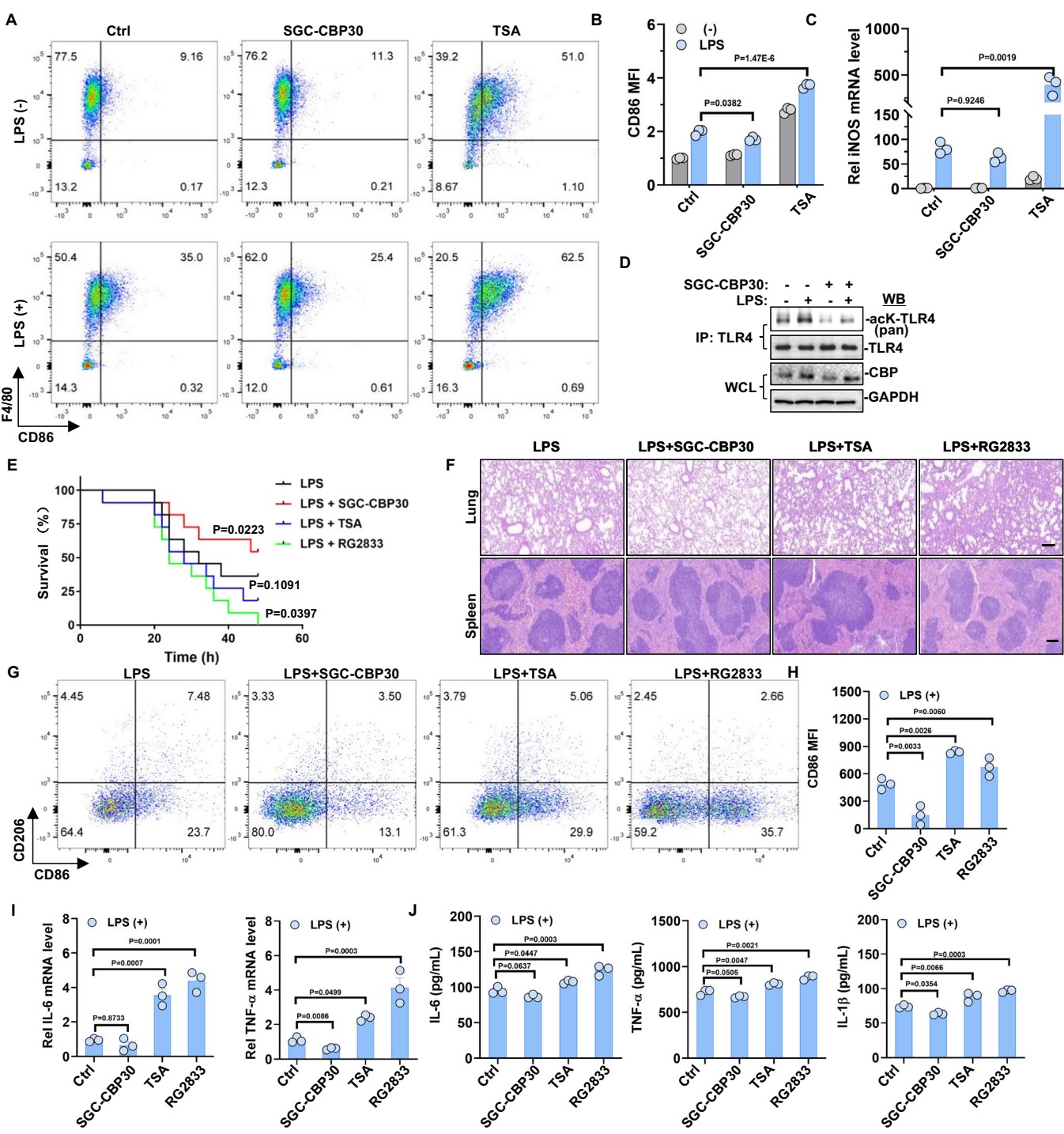

**Figure 7. Divergent effects of HDAC1 and CBP inhibitors in sepsis therapy.**

(A, B) Macrophages from TLR4-WT mice bone marrow, as previously described, were pretreated with or without 1 μM TSA or SGC-CBP30 for 6 h, followed by stimulation with 100 ng/mL LPS for 12 h. Flow cytometric analysis was conducted to assess M1 macrophage polarization using anti-F4/80 (Macrophage marker) and anti-CD86 (M1 marker) fluorescence antibodies (n = 3). (C) Macrophages from TLR4-WT mice bone marrow, as previously described, were pretreated with or without 1 μM TSA or SGC-CBP30 for 6 h, followed by stimulation with 100 ng/mL LPS for 12 h. qRT-PCR analysis was performed to measure iNOS mRNA levels (n = 3). (D) In macrophages derived from mouse bone marrow, inhibition of CBP with SGC-CBP30 resulted in the abrogation of TLR4 acetylation induction by LPS. (E) Survival curves of TLR4-WT mice subjected to different treatments after intraperitoneal injection with 20 mg/kg LPS (n = 11). (F) Representative H&E staining images of lung and spleen tissues isolated from TLR4-WT mice subjected to various treatments after intraperitoneal injection with 20 mg/kg LPS for 12 h. The scale bar = 100 μm. (G, H) TLR4-WT mice peritoneal macrophages were obtained from mice subjected to different treatments after intraperitoneal injection with 20 mg/kg LPS for 12 h. Flow cytometric analysis was conducted to detect M1 macrophage polarization using anti-F4/80 (Macrophage marker), anti-CD86 (M1 marker), and anti-CD206 (M2 marker) fluorescence antibodies (n = 3). (I) Quantitative qRT-PCR validation for IL-6 and TNF-α in TLR4-WT mice peritoneal macrophages subjected to different treatments after intraperitoneal injection with 20 mg/kg LPS for 12 h (n = 3). (J) The levels of IL-6, TNF-α, and IL-1β in the serum collected from TLR4-WT mice subjected to different treatments for 12 h were detected by ELISA assay (n = 3). One-way ANOVA. Log-rank test. Error bars, s.e.m. Source data are available online for this figure.

treatments, after various procedures, cells were fixed with 4% (v/v) paraformaldehyde and permeabilized with 1% Triton X-100 for 30 min. The cells were then blocked with 3% BSA for 1 h and incubated with the specified primary antibody overnight at 4 °C. The following day, cells were rinsed three times with PBS and incubated with a secondary fluorescence antibody for 60 min at room temperature. DAPI was used to stain the cell nucleus. Immunostaining was visualized under a confocal microscope (Nikon Eclipse Ti, Japan).

### Mass spectrometry

The cells were lysed, and immunoprecipitation was performed using the cell lysate. Subsequently, proteins were separated by SDS-PAGE, and the gel was stained with Coomassie brilliant blue. The specific gel band was excised for mass spectrometry analysis, conducted as previously described, using the LC-MS/MS system (Thermo Fisher ISQ, USA).In brief, peptides, digested through a modified in-gel chymotrypsin procedure, were extracted by eliminating the ammonium bicarbonate solution, followed by washes with a solution containing 50% (v/v) acetonitrile and 5% (v/v) formic acid. The samples were then dried and reconstituted in 5 µL of HPLC solvent A [5% (v/v) acetonitrile, 0.005% heptafluorobutyric acid, 0.4% acetic acid]. A nanoscale reverse-phase HPLC capillary column was prepared by pacKing 5-µm C18 spherical silica beads into a fused silica capillary (75-µm inner diameter × 12-cm length). After column equilibration, each sample was pressure-loaded offline onto the column, which was subsequently reattached to the HPLC system. A gradient was applied, and peptides were eluted with increasing concentrations of solvent B [95% (v/v) acetonitrile, 0.005% heptafluorobutyric acid, 0.4% acetic acid]. The mass values of all potential trypsin-digested peaks of candidate proteins were compared with the mass values of the peptide sequences containing putative acetylation site(s). A Liquid Chromatography electrospray 3D Quadrupole ion trap (LCQ) mass spectrometer was configured to perform directed MS/MS events continually for the chymotryptic peptides and the acetylated version of the same peptide. We identified peptide segments using an enzyme-free method by matching the obtained fragment patterns with the protein database.

### Human samples

Peripheral blood samples from both sepsis patients and healthy individuals were sourced from Zhejiang Provincial People's Hospital, with the approval of the Ethical Committee on Human Research (Number:QT2023206). All individuals provided informed consent. In some experiments, we divided these sepsis patients into two categories: sepsis and septic shock. The inclusion criteria for patients with sepsis and septic shock are based on "The Third International Consensus Definitions for Sepsis and Septic Shock (Sepsis-3.0)" (Singer et al, 2016). The exclusion criteria for sepsis were: the presence of chronic inflammatory diseases or malignancy, treatment with immunosuppressive drugs or steroids in one month, HIV/AIDS, hepatic failure, hepatitis B or C, and pregnancy. The sex and ages between Sepsis and healthy individuals were matched. Detailed information is displayed in Table 1. Sample selection was random and not blind.

### Blood monocytes isolation

The blood samples were diluted with PBS in a 1:1 ratio and layered onto a Ficoll-Paque density gradient cell separation (GE Healthcare). After centrifugation for 20 min at 600×*g*, the layer containing

mononuclear cells was carefully collected. The collected cells were re-dissolved in 10 ml of cold PBS and centrifuged for an additional 10 min at 350×*g*. Pan Monocyte Isolation Kit (Miltenyi Biotech) was applied to further purify monocytes from blood mononuclear cells. The purified monocytes obtained from this process were utilized for subsequent experiments.

### Single-cell RNA sequencing

The purified monocytes were labeled with TLR4-acK and FITC-conjugated secondary antibody, then positively stained cells were enriched through the BD FACSAria™ III Cell Sorter. The single-cell suspension was added to the 10x Chromium chip according to the instructions for the 10X Genomics Chromium Single-Cell 3' kit (V3), cDNA amplification and library construction were performed according to standard protocols. Libraries were sequenced by LC-Bio Technology (Hangzhou, China) on an Illumina NovaSeq 6000 sequencing system at a minimum depth of 20,000 reads per cell. Results from Illumina sequencing offline were converted to FASTQ format using bcl2fastq software (version 5.0.1). The scRNA-seq sequencing data were compared to the reference genome using CellRanger software. The output CellRanger expression profile matrix was loaded into Seurat (version 4.1.0) for filtering of low-quality cells from scRNA-seq data, and the filtered data was downscaled and clustered, and cells were projected into 2D space using UMAP.

### Molecular docking

To predict the docking of acetylated TLR4 with its adapter protein, we used the Gromacs 2019 program under constant temperature, constant pressure, and periodic boundary conditions. The Amber14SB full atomic force field and SPC/E water model were applied. During the molecular dynamics (MD) simulation process, all hydrogen bonds were constrained using the LINCS algorithm with an integration step size of 2 fs. Electrostatic interactions were calculated using the particle mesh Ewald (PME) method, with a cutoff value for non-bonding interactions set to 10 Å, updated every ten steps. The V-rescale temperature coupling method was employed to maintain the simulated temperature at 310 K, and the Parrinello-Rahman method was used to control the pressure at 1 bar. Initially, the steepest descent method was used to minimize the energy of the four systems, eliminating overly tight atomic contacts. Subsequently, a 100 ps NPT equilibrium simulation was conducted at 310 K. Finally, a 100 ns MD simulation was performed on the four different systems, with conformations saved every 20 ps, resulting in a total of 5000 conformations saved.The visualization of simulation results was achieved using the embedded Gromacs program and VMD.

### Surface plasmon resonance

To prepare the TLR4-immobilized sensor chip, human recombinant TLR4 protein (ab233665, Abcam, purity >95%) was reconstituted in distilled water. It was then diluted in 10 mM sodium acetate (pH 4.5). The TLR4 protein solution was allowed to contact the CM5 sensor chip for 7 min and then immobilized using an amine coupling kit containing 1-ethyl-3-(3-dimethylaminopropyl) carbodiimide hydrochloride (EDC), N-hydroxysuccinimide (NHS), and 1 M ethanolamine hydrochloride-NaOH (pH 8.5). Chip regeneration was carried out using 50 mM NaOH. The analytes were diluted in CBP (31590, activemotif, purity >90%)

with 0.5% (v/v) DMSO to concentrations of 15.625, 31.25, 62.5, 125, 250, 500, and 1000 nM. The analyte solution was injected onto the TLR4 ECD-immobilized sensor chip in a Biacore X100 system (Cytiva) at a flow rate of 30 μl/min (contact time: 120 sec, dissociation time: 200 sec). Manual run and single-cycle kinetics modes were employed to determine the response unit (RU) values and dissociation rate (Kd) values, respectively. Data analysis was conducted using Biacore X100 evaluation software version 1.0+ (Cytiva).

### Small-molecule inhibitor

In some experiments, macrophages derived from bone marrow were pretreated with 1 μM CBP inhibitor (SGC-CBP30) for 6 h. In addition, HDAC family inhibitors were used to treat macrophages for 6 h, and JNJ-26481585 was administered at a concentration of 0.1 μM, targeting HDAC1-11. CAY-10683 was used at 1 μM, specifically targeting HDAC2 and HDAC6. BG45 was given at 1 μM, focusing on HDAC3. RG2833 was also administered at 1 μM, targeting both HDAC1 and HDAC3. ABR-215050 was provided at 1 μM, specifically targeting HDAC4. LMK-235 was used at a concentration of 0.1 μM, targeting HDAC1, HDAC2, HDAC4, HDAC5, HDAC6, HDAC8, and HDAC11. HPOB was administered at 1 μM, specifically targeting HDAC6. TMP195 was used at a dosage of 1 μM, targeting HDAC4, HDAC5, HDAC7, and HDAC9. PCI34051 was given at 1 μM, specifically targeting HDAC8. TMP269 was administered at 1 μM, focusing on HDAC9, HDAC7, HDAC4, and HDAC5. CUDC-907 was provided at a concentration of 0.1 μM, targeting HDAC1, HDAC3, and HDAC10. Lastly, MGCD0103 was used at 1 μM, targeting HDAC1, HDAC2, HDAC3, and HDAC11.

### Knockout cell preparation

Gene targeting using CRISPR-Cas9 was achieved by inserting gene-specific sgRNAs into the pLentiCRISPR v2 vector. The CBP-sgRNAs were used as follows: ATGGCTGAGAACTTGCTGGA. The construction system was prepared as lentiviral particles by transfecting HEK293T cells with packaging plasmids psPAX2 and pMD2G. Macrophages were screened with puromycin for at least 2 weeks, and individual colonies were amplified and validated through western blot analysis.

### siRNA transfection

SiRNA-HDAC1 sequence: siRNA-HDAC1-1: sense strand: GAGU-CAAAACAGAGGAUGAUU; antisense strand: UCAUCCUCUG UUUUGACUCUU. siRNA-HDAC1-2: sense strand: GCUCCA UCCGUCCAGAUAAUU; antisense strand: UUAUCUGGACG-GAUGGAGCUU. siRNA-HDAC1-3: sense strand: GCCGGUCAU-GUCCAAAGUAUU antisense strand: UACUUUGGACAUGA CCGGCUU. Dilute RNAiMAX reagent and siRNA in Opti-MEM medium, then add diluted siRNA to diluted RNAiMAX reagent (1:1 ratio) and transfect them into cell.

### Flow cytometry

Macrophages originated from the bone marrow were treated with 100 ng/ml S-LPS (055:B5) for 30 min. Add 100 μL of IC fixation buffer (00-8222-49, Invitrogen) to fix macrophages at room temperature for 30 min. Then, add 2 mL of 1X permeabilization buffer (00-8333-56, Invitrogen) and centrifuge at 500× g for 5 min at room temperature. Resuspend the cell pellet in 100 μL of 1×

permeabilization buffer. Add the TLR4-acK732 or TLR4-acK813 antibodies and incubate at 4 °C for 1.5 h. Subsequently, incubate with a secondary fluorescent antibody for an additional 1 h. In some experiments, macrophages were collected and incubated with APC-conjugated anti-mouse F4/80, PE-conjugated anti-mouse CD86, and PE/Cy7-conjugated anti-mouse CD206 for 1 h. Purified monocytes obtained from both normal and sepsis conditions were incubated with antibodies against TLR4-acK732 or TLR4-acK813, APC-conjugated anti-human CD14, and PE-conjugated anti-human CD16 for 1 h. This was followed by incubation with a FITC-conjugated secondary fluorescence antibody for an additional 1 h. Finally, the cells were analyzed using a FACS Canto II system (BD, USA).

### Luciferase reporter assay

HeLa cells were transfected with NF-κB or IRF3 luciferase reporter plasmids, along with the specified plasmids, using Lipofectamine 2000, as previously described. Subsequently, the cells were treated with 100 ng/mL S-LPS for 6 h. After an additional 30 h, cell lysates were collected for luciferase analysis. Triplicate samples containing the luciferase substrate were transferred to a 96-well plate, and the bioluminescence of the reaction mix was measured using a Synergy 2 system (BioTek, USA).

### RNA-seq

Mouse peritoneal macrophages were obtained as previously described and subjected to treatment with 100 ng/ml S-LPS for 6 h. Subsequently, total RNA from the cells was isolated using the RNeasy Mini Kit (QIAGEN, Valencia, CA, USA), following the manufacturer's instructions. The RNA-seq analysis was performed by GENEWIZ, Inc., utilizing an Illumina sequencing platform. Genes were considered significantly differentially expressed if they had an adjusted $P$ value < 0.01.

### qRT-PCR

Initially, total RNA was extracted from peritoneal macrophages following the manufacturer's protocol. Subsequently, the RNA underwent reverse transcription using the high-capacity cDNA Synthesis kit, following the manufacturer's guidelines. Real-time PCR was conducted with SYBR Green Master Mix (YEASEN, China) on an ABI QuantStudio (Applied Biosystems). The expression levels of the target genes were normalized to GAPDH, and "fold differences" were calculated using the ΔΔCt method. The primer sequences are provided in "Reagents and Tools Table".

### ELISA analysis

Mouse peritoneal macrophages from TLR-WT or TLR4-KR mutant mice were treated with or without 100 ng/mL S-LPS for 12 h. Cell supernatant samples were collected for ELISA analysis to detect IL-6, TNF-α, IFN-γ and IL-1β.The concentrations of pro-inflammatory cytokines (IL-6, TNF-α) in the serum of treated mice with 25 mg/kg S-LPS for 4 h were determined using ELISA kits, following standard protocols. Data collection was performed using Gen5 (version 3.10).

### LPS-induced sepsis model

For the survival curve test, TLR4 WT, K730R, and K810R mutant mice were challenged with 25 mg/kg S-LPS ($n$ = 10) (Buras et al, 2005; Li et al, 2022b), and the survival of mice was monitored every

hour for 24 h. Six feMALe mice per group were subjected to intraperitoneal injection with 25 mg/kg S-LPS or saline. After 4 h, mice were euthanized, and eyeball blood was collected to obtain serum for ELISA analysis. The spleen or lungs were fixed with 4% polyformaldehyde for subsequent H&E staining.

### Statistical analysis

Statistical analysis of the data was conducted using the Statistical analysis of data was performed by Student's *t* test or One-way ANOVA test with GraphPad Prism 8.0 software. Key experiments were performed in triplicates. All data are presented as mean ± SEM, and *P* values less than 0.05 were considered statistically significant.

## Data availability

The RNA-seq datasets from this study have been deposited to the Gene Expression Omnibus (https://www.ncbi.nlm.nih.gov/geo/query/acc.cgi?acc=GSE268405) with the accession # GSE26840. The Single-cell RNA-seq datasets have been deposited to the Gene Expression Omnibus with the accession # GSE268406. The mass spectrometry proteomics data have been deposited to the ProteomeXchange Consortium via the PRIDE partner repository with the dataset identifier PXD052433.

The source data of this paper are collected in the following database record: biostudies:S-SCDT-10_1038-S44318-024-00237-8.

## Peer review information

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

## Acknowledgements

The authors gratefully acknowledge the financial support received from the Natural Science Foundation of China (Grant Numbers: 81820108023, 82030077, 8192913, 2018YFC1705505), the Open Project Program of the State Key Laboratory of Radiation Medicine and Protection (Grant Number: GZK12023025), the Interdisciplinary Basic Frontier Innovation Program of Suzhou Medical College, Soochow University (Grant Number: YXY2304036), a Project Funded by the Medical Science and Technology Project of Zhejiang Province (Grant Number: 2024KY670), and another project supported by the Priority Academic Program Development of Jiangsu Higher Education Institutions.

## Author contributions

**Xue Li**: Data curation; Formal analysis; Supervision; Investigation; Methodology; Writing—original draft; Writing—review and editing. **Xiangrong Li**: Data curation; Supervision; Methodology; Writing—original draft. **Pengpeng Huang**: Data curation; Methodology. **Facai Zhang**: Software; Methodology. **Juanjuan K Du**: Software; Methodology. **Ying Kong**: Methodology. **Ziqiang Shao**: Resources; Methodology. **Xinxing Wu**: Resources; Methodology. **Weijiao Fan**: Supervision; Methodology. **Houquan Tao**: Resources; Methodology. **Chuanzan Zhou**: Methodology. **Yan Shao**: Methodology. **Yanling Jin**: Resources. **Meihua Ye**: Supervision; Methodology. **Yan Chen**: Conceptualization; Resources; Methodology. **Jong Deng**: Supervision. **Jimin Shao**: Conceptualization. **Jicheng Yue**: Writing—review and editing. **Xiaju Cheng**: Data curation; Formal analysis; Supervision; Methodology; Writing—original draft; Writing—review and editing. **Y Eugene Chin**: Conceptualization; Resources; Supervision; Funding acquisition; Methodology; Project administration; Writing—review and editing.

Source data underlying figure panels in this paper may have individual authorship assigned. Where available, figure panel/source data authorship is listed in the following database record: biostudies:S-SCDT-10_1038-S44318-024-00237-8.

## Disclosure and competing interests statement

The authors declare no competing interests.

# Expanded View Figures

**Figure EV1.  Cytosolic CBP induced by LPS promotes acetylation of the TIR domain complex.**

(**A**) Macrophages derived from mouse bone marrow were exposed to 100 ng/mL LPS for 0, 5, 10, 15, 30, and 60 min. Subsequently, cells were immunoprecipitated with a TLR4 antibody, followed by western blot analysis with pan-acetyl-K and TLR4 antibodies. Whole cell lysate (WCL) was probed with an IκBα antibody. (**B**) The specificity of polyclonal antibodies targeting residues of TLR4-acK732 and TLR4-acK813 acetylation, using synthesized peptides as antigens, was assessed through dot blot analysis. (**C**) HEK293T cells were transfected with the indicated plasmids. Immunoprecipitation with Flag antibody was followed by western blot analysis with acK732 and acK813-TLR4 and Flag antibodies. (**D**) Macrophages derived from the bone marrow of TLR4-WT and TLR4-KR mice were treated with 100 ng/mL LPS for 30 min. Subsequently, the cells were immunoprecipitated with a TLR4 antibody, followed by western blot analysis using TLR4-acK732 and TLR4-acK813 antibodies. (**E**) THP-1 cells were induced into macrophages as described previously. Immunoblotting was performed with TLR4-acK813, TLR4-acK732, and TLR4 antibodies. Whole cell lysate (WCL) was probed with TLR4 and GAPDH antibody. (**F**) HEK293T cells were transfected with the indicated plasmids. Immunoprecipitation with Flag antibody was followed by western blot analysis with pan-acetyl-K and Flag antibodies. (**G**) HEK293T cells were transfected with Myc-TLR4-TIR and HA-CBP. Immunoprecipitation with Myc antibody was followed by western blot analysis with Myc-TLR4-TIR pan-acetyl-K and Myc antibodies. (**H**) Conservation analysis of the K732 and K813 sites in TLR family members. (**I**) Macrophages derived from mouse bone marrow were treated with 100 ng/mL LPS for 30 min. Immunoprecipitation with MyD88 antibody was followed by mass spectrometry analysis of MyD88 to identify acetylation residues, specifically acK231 and acK238, in the MyD88-TIR domain. (**J**) HEK293T cells were transfected with Flag-MAL and HA-CBP. Immunoprecipitation with Flag antibody was followed by mass spectrometry analysis of MAL to identify acetylation residues, specifically acK210 in the MAL-TIR domain.

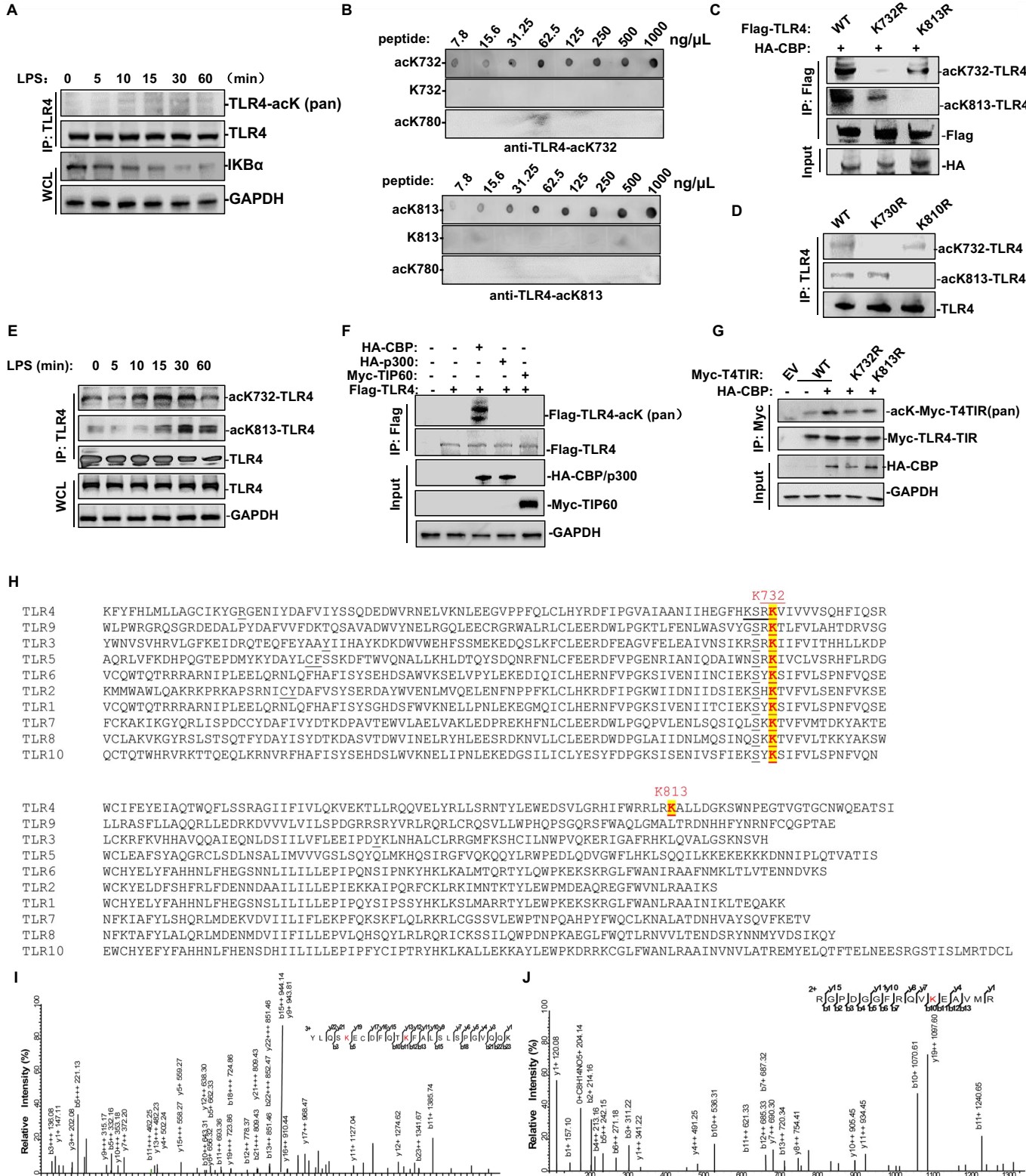

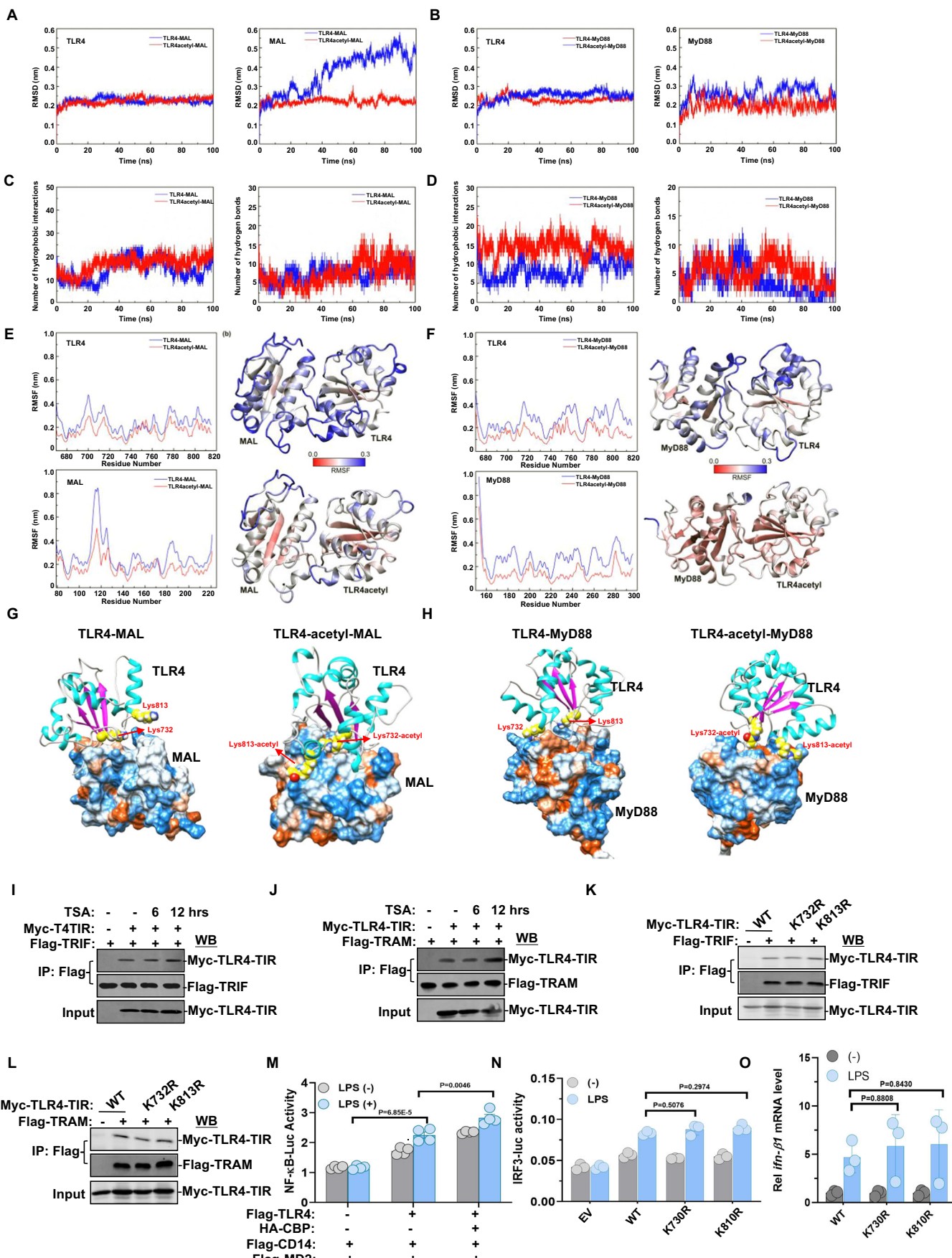

◀ **Figure EV2. Molecular simulation reveals that acetylated TLR4 enhances its binding affinity with adaptors MAL and MyD88.**

(A) The root mean square deviation (RMSD) of skeleton atoms in the TLR4–MAL and TLR4-acetyl-MAL systems during MD simulation (B) The root mean square deviation (RMSD) of skeleton atoms in the TLR4-MyD88 and TLR4-acetyl-MyD88 systems during MD simulation. (C) The number of hydrophobic interactions and hydrogen bonds formed between MAL and non-acetylated or K732/K813-acetylated TLR4 during molecular simulation. (D) The number of hydrophobic interactions and hydrogen bonds formed between MyD88 and non-acetylated or K732/K813-acetylated TLR4 during molecular simulation. (E) The root mean square fluctuation (RMSF) distribution of residues in the TLR4–MAL and TLR4-acetyl-MAL systems and the distribution of RMSF on protein secondary structure. (F) The root mean square fluctuation (RMSF) distribution of residues in the TLR4-MyD88 and TLR4-acetyl-MyD88 systems and the distribution of RMSF on protein secondary structure. (G) The interaction pattern between MAL and non-acetylated or K732/K813-acetylated TLR4 on the hydrophilic and hydrophobic surfaces after MD simulation (blue and orange regions on the protein surface represent hydrophilic and hydrophobic regions, respectively). (H) The interaction pattern between MyD88 and non-acetylated or K732/K813-acetylated TLR4 on the hydrophilic and hydrophobic surfaces after MD simulation (blue and orange regions on the protein surface represent hydrophilic and hydrophobic regions, respectively). (I, J) HEK293T cells were transfected with the indicated plasmids and then treated with 1 μM TSA for 6 h and 12 h. Cells were subjected to immunoprecipitation with Flag antibody followed by western blot with Myc and Flag antibodies. TSA treatment slightly enhances the interaction of TLR4-TIR with TRIF or TRAM. (K, L) TLR4-WT or TLR4-K732R or K813R mutation was co-transfected with TRIF or TRAM in HEK293T cells. Neither the K732R nor K813R mutant could abolish the interaction of TLR4-TIR with TRIF or TRAM. (M) The effect of TLR4 acetylation on NF-κB transcriptional activity regulation was tested in HEK293T cells. Cells were transfected with the indicated plasmids and NF-κB luciferase reporter for 42 h, and then treated with 1000 ng/ml LPS for another 6 h ($n = 4$). (N) The effect of TLR4 acetylation on IRF3 transcriptional activity regulation was tested in HeLa cells. Cells were transfected with the indicated plasmids and IRF3 luciferase reporter for 42 h, and then treated with 1000 ng/mL LPS for another 6 h ($n = 3$). (O) Quantitative qRT-PCR validation of IFN-β1 in M1 macrophages from TLR4-WT or TLR4-KR mutant mice bone marrow treated with or without LPS ($n = 3$). One-way ANOVA. Error bars, s.e.m.

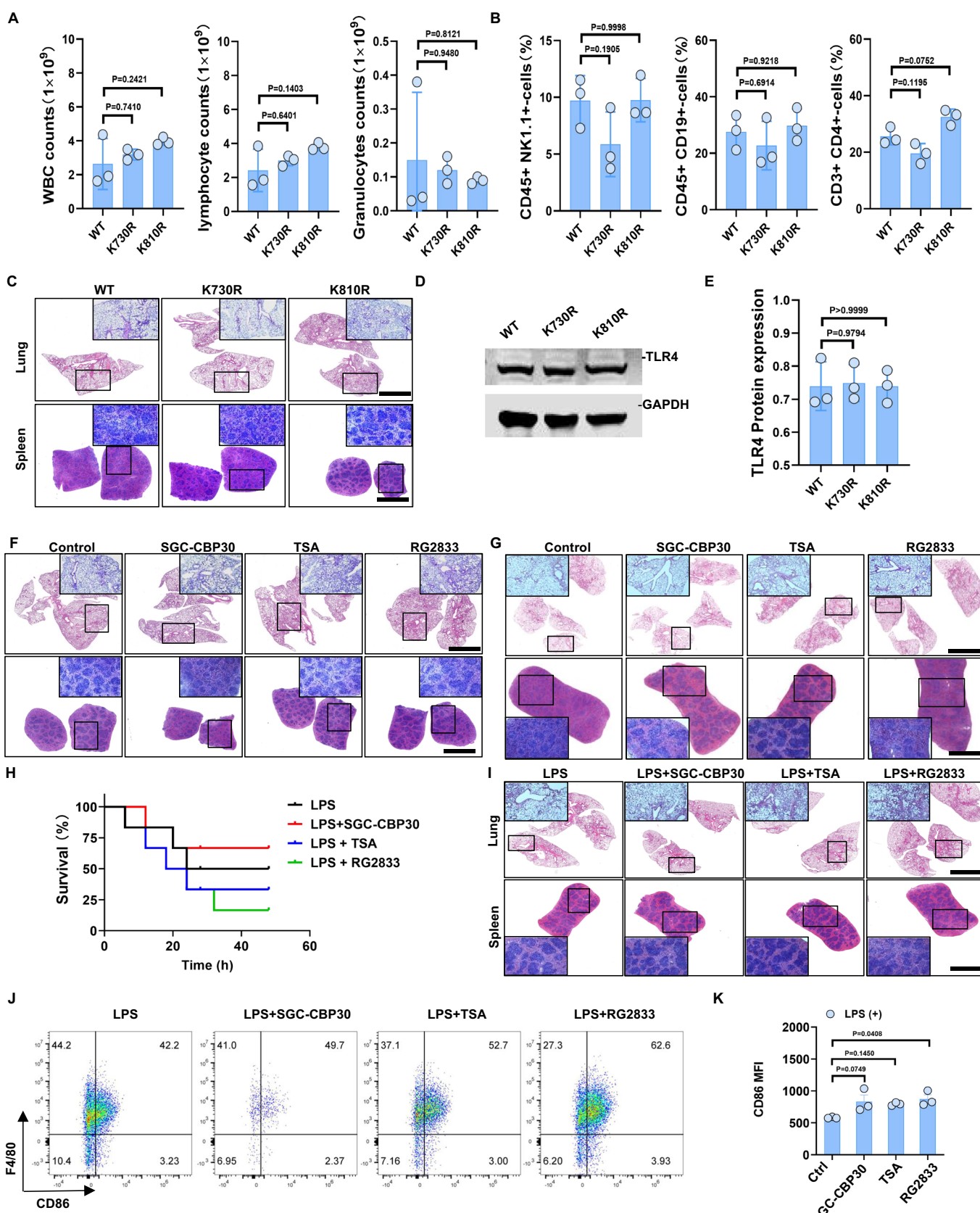

**Figure EV3.   Detection of WBC in TLR4-KR mutant mice and therapy in sepsis of TLR4-K810R mice.**

(A) Blood routine analysis of white blood cells, lymphocytes, and granulocytes in TLR4-WT or TLR4-KR mutant mice blood ($n = 3$). (B) Splenic cells obtained from TLR4-WT or TLR4-KR mutant mice. Cells were collected for flow cytometer analysis to detect the polarization of NK, B, CD4$^+$T cells ($n = 3$). (C) Representative H&E staining images of lung and spleen tissues isolated from TLR4-WT or TLR4-KR mutant mice. Scale bar $=$ 2 mm. (D, E) Macrophages derived from TLR4-WT or TLR4-KR mutant mice bone marrow and subjected to Western blot analysis with TLR4 antibody (D). Densitometry measurements were performed using ImageJ (E). Data shown are representative of three independent experiments. (F) Representative H&E staining images of lung and spleen tissues isolated from TLR4-WT mice subjected to various treatments. The scale bar $=$ 2 mm. (G) Representative H&E staining images of lung and spleen tissues isolated from TLR4-K810R mice subjected to various treatments. The scale bar $=$2 mm. (H) Survival curves of TLR4-K810R mice subjected to different treatments after intraperitoneal injection with 20 mg/kg LPS ($n = 6$). (I) Representative H&E staining images of lung and spleen tissues isolated from TLR4-K810R mice subjected to various treatments after intraperitoneal injection with 20 mg/kg LPS for 12 h. The scale bar $=$ 2 mm. (J, K) Peritoneal macrophages from TLR4- K810R mice were subjected to different treatments after intraperitoneal injection with 20 mg/kg LPS for 12 h. Cells were collected for flow cytometer analysis to detect the polarization of M1 macrophages using anti-CD45, anti-F4/80 and anti-CD86 fluorescence antibodies ($n = 3$). One-way ANOVA. Log-rank test. Error bars, s.e.m. Source data are available online for this figure.

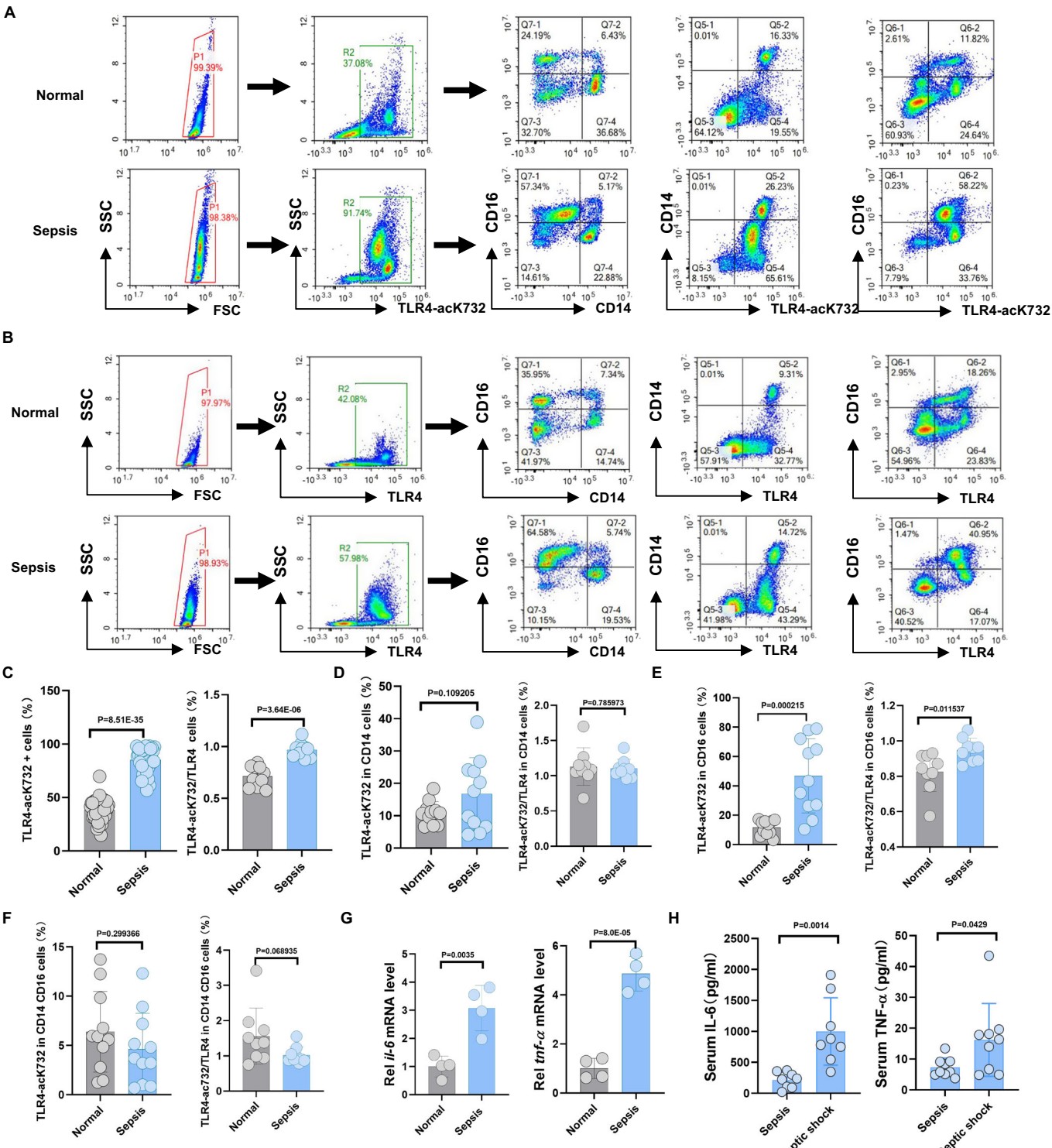

◀ **Figure EV4. TLR4-acK732 and TLR4 expression in monocytes from sepsis patients.**

(A, B) Monocytes were collected for flow cytometry analysis to detect the polarization of TLR4-acK732 (A) and TLR4 (B) in CD14 + , CD16 + , and CD14 + CD16+ cell types. (C) Monocytes were obtained for flow cytometry analysis to detect the ratio of TLR4-acK732 ($n = 43$) and TLR4-acK732/TLR4 ($n = 10$) in normal and sepsis patients. (D) Monocytes were obtained for flow cytometry analysis to detect the ratio of TLR4-acK732 ($n = 11$) and TLR4-acK732/TLR4 ($n = 9$) in normal and sepsis patients' CD14+ cell type. (E) Monocytes were obtained for flow cytometry analysis to detect the ratio of TLR4-acK732 ($n = 11$) and TLR4-acK732/TLR4 ($n = 9$) in normal and sepsis patients' CD16+ cell type. (F) Monocytes were obtained for flow cytometry analysis to detect the ratio of TLR4-acK732 ($n = 11$) and TLR4-acK732/TLR4 ($n = 9$) in normal and sepsis patients' CD14 + CD16+ cell type. (G) Mononuclear cells were obtained by sorting with TLR4-acK and CD16 antibodies from both normal individuals and sepsis patients. The expression of IL-6 and TNF-α in these cells was then detected using real-time PCR ($n = 4$). (H) Serum IL-6 and TNF-α levels in patients with sepsis and septic shock ($n = 8$). Student's *t* test. Error bars, s.e.m.

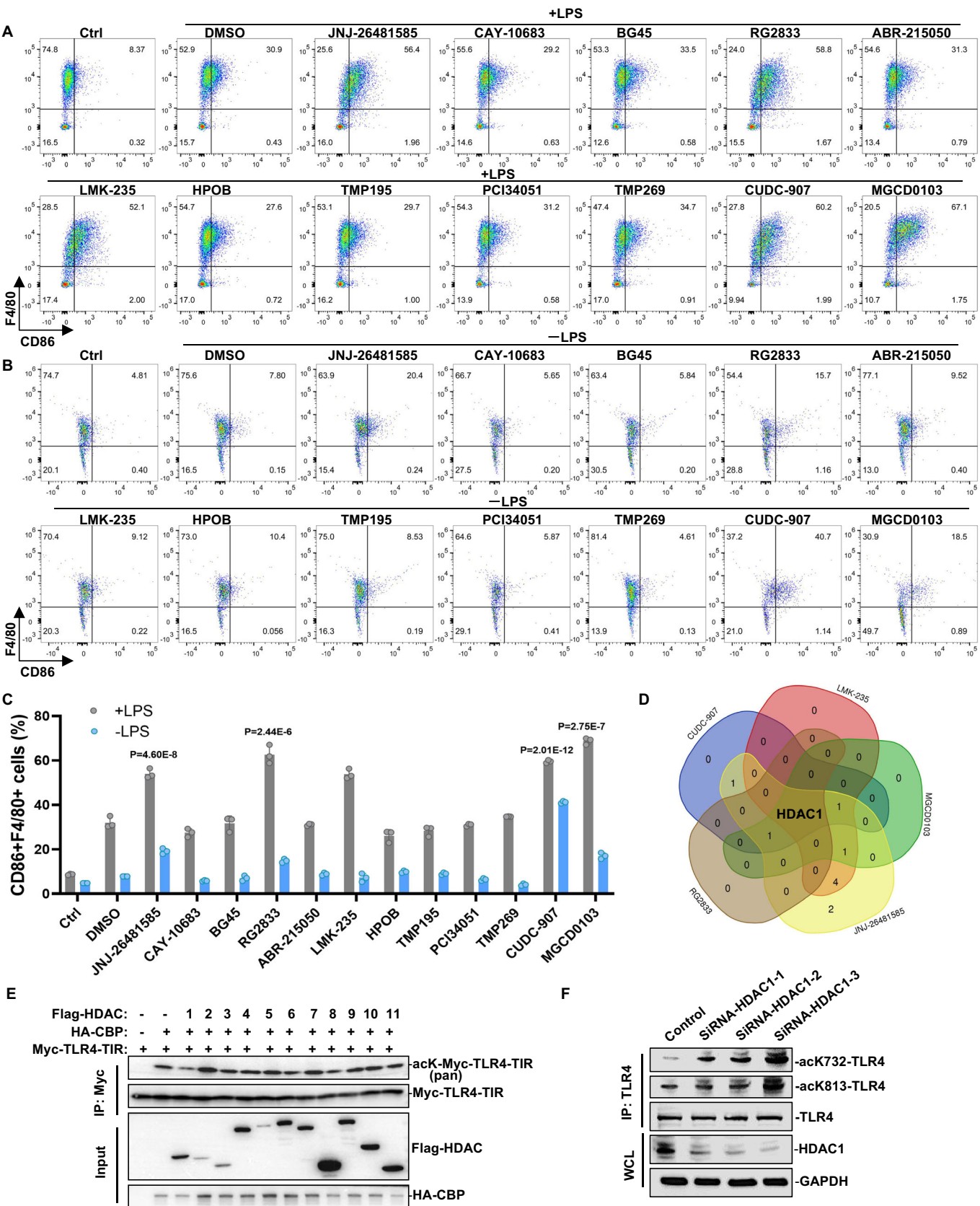

◀ **Figure EV5. HDAC1 mediated TLR4-TIR deacetylation.**

(A, B) Macrophages derived from TLR4-WT mice bone marrow, as described before, were pretreated with a series of HDAC family inhibitors for 6 h and then treated with (A) or without (B) 100 ng/mL LPS for 12 h. Cells were collected for flow cytometry analysis to detect the expression of M1 macrophages by staining with anti-F4/80 (macrophage marker) and anti-CD86 (M1 marker) fluorescence antibodies. (C) Statistical analysis in the expression of M1 macrophages treated with or without LPS (n = 3). (D) Venn diagram analysis illustrating HDAC family members targeted by inhibitors in the expression of M1 macrophages. (E) HEK293T cells were transfected with Myc-TLR4-TIR, HA-CBP, and Flag-HDAC family plasmids. Cells were subjected to immunoprecipitation with Myc antibody followed by western blot with pan-acetyl-K, Flag, HA, and Myc antibodies. The TLR4-TIR domain was efficiently deacetylated by HDAC1. (F) THP-1 cells were induced into macrophages and then were transfected with siRNA for 36 h, followed by 100 ng/ml LPS treatment for 30 min. Cells were subjected to immunoprecipitation with TLR4 antibody followed by western blot with acK732-TLR4、acK813-TLR4 antibodies. One-way ANOVA. Error bars, s.e.m.

