## [Peer Review File · The EMBO Journal]

Acetylation of TIR domains in the TLR4-Mal-MyD88 complex regulates immune responses in sepsis

Xue Li, Xiangrong Li, Pengpeng Huang, Facai Zhang, Juanjuan Du, Ying Kong, Ziqiang Shao, Xinxing Wu, Weijiao Fan, Houquan Tao, Chuanzan Zhou, Yan Shao, Yanling Jin, Meihua Ye, Yan Chen, Jong Deng, Jimin Shao, Jicheng Yue, Xia-ju Cheng, and Y. Eugene Chin

Corresponding author(s): Y. Eugene Chin (chinyue@suda.edu.cn) , Xia-ju Cheng (xjcheng@suda.edu.cn), Xue Li (snowlee@zju.edu.cn)

Review Timeline:

Submission Date:	17th Mar 24
Editorial Decision:	3rd May 24
Revision Received:	29th Jun 24
Editorial Decision:	31st Jul 24
Revision Received:	14th Aug 24
Accepted:	20th Aug 24

Editor: Ioannis Papaioannou

Transaction Report:

Dear Prof. Chin,

Thank you for submitting your manuscript EMBOJ-2024-117310 for consideration by The EMBO Journal, and for your patience during peer review. It has now been seen by three experts in the field, and we have received the full set of their comments, which you can find below.

As you will see, the referees -who have provided very detailed and well-informed reports- are largely supportive of the study. They acknowledge that it provides substantial advance over our previous knowledge of TLR4 regulation, which is also relevant to TLR4-related diseases including sepsis. They also raise a number of concerns and provide several constructive suggestions for further strengthening of the study and the manuscript.

Given the referees' comments and recommendations, I would like to invite you to submit a revised version of the manuscript along with a detailed point-by-point response addressing all referees' comments. I should add that it is EMBO Journal policy to allow only a single round of major revision, and acceptance of your manuscript will therefore depend on the completeness of your responses in this revised version. Please let me know if you have any questions or comments.

We generally allow three months as standard revision time (August 2, 2024). As a matter of policy, competing manuscripts published during this period will not negatively impact our assessment of the conceptual advance presented by your study. However, we request that you contact us as soon as possible upon publication of any related work, to discuss how to proceed. Should you foresee a problem in meeting this three-month deadline, please let us know in advance and we may be able to grant an extension.

Thank you for the opportunity to consider your work for publication in The EMBO Journal. I look forward to your revision.

Best regards,

Ioannis

Instructions for preparing your revised manuscript

1. When you are ready to submit the revision, please upload:

- A Word file of the manuscript text (including legends of main Figures, EV Figures and Tables). Please make sure that changes are highlighted (or "tracked") to be clearly visible.

- Individual production-quality figure files (one file per figure). When assembling your figures, please refer to our figure preparation guidelines in order to ensure proper formatting and readability in print as well as on screen:

If the data shown in a figure are obtained from n {less than or equal to} 2, please use scatter plots showing the individual data points.

- i. the name of the statistical test used to generate error bars and P values
- ii. the number (n) of independent experiments (please specify technical or biological replicates) underlying each data point (discussion of statistical methodology can be reported in the Materials and Methods section, but figure legends should contain a basic description of n , P , and the test applied)
- iii. the nature of the bars and error bars (s.d., s.e.m.).

- A point-by-point response to the referees' comments, with a detailed description of the changes made (as a word file). All referees' concerns must be fully addressed and their suggestions taken on board. When preparing your letter of response to the referees' comments, please bear in mind that this will form part of the Review Process File and will therefore be available online to the community. Please note that you have the possibility to opt out of the transparent process at any stage prior to publication by letting the editorial office know (contact@embojournal.org); if you do opt out, the Review Process File link will point to the

following statement: "No Review Process File is available with this article, as the authors have chosen not to make the review process public in this case.". For more details on our Transparent Editorial Process, please visit our website: <https://www.embopress.org/page/journal/14602075/authorguide#transparentprocess>

- Expanded View (EV) files (replacing Supplementary Information) that are collapsible/expandable online. A maximum of 5 EV Figures can be typeset. EV Figures should be cited as "Figure EV1, Figure EV2" etc. in the text, and their respective legends should be included in the manuscript file after the legends of regular figures. See detailed instructions regarding Expanded View files here:

- For the figures that you do NOT wish to display as Expanded View figures, they should be bundled together with their legends in a single PDF file called "Appendix", which should start with a short Table of Contents (including page numbers). Appendix figures should be referred to in the main text as: "Appendix Figure S1, Appendix Figure S2" etc. Please see detailed instructions here: <https://www.embopress.org/page/journal/14602075/authorguide#expandedview>

- A complete author checklist, which you can download from our author guidelines (<https://www.embopress.org/page/journal/14602075/authorguide>). Please note that the checklist will also be part of the Review Process File.

2. Please note that no statistics should be calculated and shown in Figures if $n=2$. Please also note that each p value should be reported as an exact value.

3. Before submitting your revision, primary datasets (and computer code, where appropriate) produced in this study need to be deposited in appropriate public databases (see <https://www.embopress.org/page/journal/14602075/authorguide#dataavailability>). In particular, you are kindly requested to deposit all RNA sequencing and mass spectrometry data produced in your study. The accession numbers and databases should be listed in a formal "Data availability" section (placed after Materials and Methods) that follows the model below (see also <https://www.embopress.org/page/journal/14602075/authorguide#dataavailability>):

Data availability

- RNA-seq data: Gene Expression Omnibus GSE46843 (<https://www.ncbi.nlm.nih.gov/geo/query/acc.cgi?acc=GSE46843>)
- [data type]: [name of the resource] [accession number/identifier/doi] ([URL or identifiers.org/DATABASE:ACCESSION])

*** All links should resolve to a page where the data can be accessed. ***

*** Please remember to provide in the Data availability section of your revised manuscript reviewer passwords if the datasets are not yet public. ***

*** The Data Availability Section is restricted to new primary data that are part of this study. In case you have no data that require deposition in a public database, please state so instead of referring to the database: "Our study includes no data deposited in public repositories." under the heading "Data availability". ***

4. Please check that the title and the abstract of the manuscript are brief, yet explicit, even to non-specialists. The length of the title should not exceed 100 characters, and the abstract should be a single paragraph not exceeding 175 words.

5. Please also note our reference format: <https://www.embopress.org/page/journal/14602075/authorguide#referencesformat>.

7. Please remember: digital image enhancement is acceptable practice, as long as it accurately represents the original data and conforms to community standards. If a figure has been subjected to significant electronic manipulation, this must be noted in the figure legend or in the "Materials and Methods" section. The editors reserve the right to request original versions of figures and the original images that were used to assemble the figure.

8. Our journal encourages inclusion of data citations in the reference list to directly cite datasets that were obtained from public databases. Data citations in the article text are distinct from normal bibliographical citations and should directly link to the database records from which the data can be accessed. In the main text, data citations are formatted as follows: "Data ref: Smith et al, 2001" or "Data ref: NCBI Sequence Read Archive PRJNA342805, 2017". In the Reference list, data citations must be labeled with "[DATASET]". A data reference must provide the database name, accession number/identifiers, and a resolvable link to the landing page from which the data can be accessed at the end of the reference. Further instructions are available at:

<https://www.embopress.org/page/journal/14602075/authorguide#referencesformat>.

9. We request authors to consider both actual and perceived competing interests. Please review our policy (<https://www.embopress.org/page/journal/14602075/authorguide#conflictsofinterest>) and update your competing interests statement if necessary. Please name this section 'Disclosure and competing interests statement' and place it after the Acknowledgements section.

10. Please note that all corresponding authors are required to provide an ORCID ID upon submission of a revised manuscript (<https://orcid.org/>). Please find instructions on how to link your ORCID ID to your account in our manuscript tracking system in our Author guidelines (<https://www.embopress.org/page/journal/14602075/authorguide#authorshipguidelines>).

11. We use CRediT to specify the contributions of each author in the journal submission system. CRediT replaces the author contribution section, which should be removed from the manuscript. Please use the free text box to provide more detailed descriptions. See also guide to authors: <https://www.embopress.org/page/journal/14602075/authorguide#authorshipguidelines>.

13. We would also welcome the submission of cover suggestions or motifs to be used by our Graphics Illustrator in designing a cover.

14. Please use the link below to submit your revision:
<https://emboj.msubmit.net/cgi-bin/main.plex>

Referee #1:

Li et al. showed that TLR4/MAL/MyD88 signaling pathway in macrophages is modulated by the acetylation of TIR domain complex by CREB-binding protein (CBP) in response to LPS. Specifically, the K732 and K813 residues of TLR4 are served as specific substrates of CBP, and the acetylation of these TLR4-TIR domain residues play essential roles in the classical TLR4-dependent signal transduction. The Authors also demonstrated that human CD16+ monocytes can differentiate into inflammatory macrophages. The CD16+ monocyte subset can act as key innate immune components of sepsis showing an increased acetylation level of TLR4-acK732 and TLR4-acK813. It was shown that targeting histone deacetylase 1 (HDAC1) is a potential deacetylase of TLR4-TIR domain in inflammatory type 1 (M1) macrophages, and the endotoxin-induced inflammatory response and sepsis progression can be modulated by targeting HDAC1 and CBP.

Major point of criticism:

- One limitation of the study is that targeting histone deacetylases in vivo can modify the activation of TLR4-mediated signalosome, but the epigenomic regulation of inflammatory and anti-inflammatory gene expression could be affected at the same time. The Authors showed that the serum levels of pro-inflammatory cytokines in mice during sepsis show a statistical difference between control and HDAC1-inhibitor injected mice. However, these differences show minor biological differences in the serum-derived parameters to measure disease development.
- It is not obvious how human CD16+ monocyte-derived inflammatory macrophage can be associated to the tested murine inflammatory macrophage (M1) polarization. The monocytes represent a heterogenous populations of innate immune cells in mice, and it is not clear which population of macrophages were analyzed in the murine sepsis model. Are these monocyte-derived macrophages isolated from the spleen or from the peritoneum?

Minor points of criticism:

- Peritoneal macrophages express cell surface CD86 molecule at homeostatic environment, but the authors show that there is a significant CD86- population of macrophages in Figure 1 and 4. It is clearly visible that there are three populations on the dot plots; one negative, a CD86low and a CD86high cell population. The median or mean expression level of CD86 expression intensity should be presented instead of the frequency. Using technical controls, the authors can demonstrate that the macrophages express CD86 which expression level depends on the cell activation state.
- The BMDM differentiation and collection of primary peritoneal macrophages is the section 'Cell lines' of Methods. The Authors have to separate the methods utilizing primary cells and cell lines.
- It is confusing how the Authors refer to macrophage subpopulations in figure legends and the results. The Authors should clearly define what is a mouse macrophage. a primary macrophage, macrophages obtained from bone marrow, M1 macrophage etc. These terms should be used consistently across the manuscript.

- On Figure 3D,E,F: please check if the gene Tlr3 is mistyped.
- The origo of y axis showing cytokine concentrations (Figure 7J) has to be zero.
- In the discussion, some sentences should be reedited, or merged: "However, we were not sure the TLR4-TIR acetylation associated adaptor preference resulted from acetylation site specificity or cell type specificity. Even it is reported that it is CD14 that transferred LPS to the TLR4-MD2 dimerization complex (Park and Lee, 2013; Ryu et al., 2017)." ".... And marker genes..."

Referee #2:

In this manuscript, Li et al, present evidence towards acetylation of TIR domain of TLR4 dictating activation of downstream NFkB pathway and associated proinflammatory cytokine gene expression. This acetylation is mediated by the histone acetyltransferase (HAT), CBP and antagonized by the histone deacetylase, HDAC1. The authors identify two lysine residues in the TIR domain of TLR4, positioned at 730 and 810 (in mouse), of which lysine residue at 810th position is critical to this response. In an in vivo model of sepsis induced by LPS, antagonizing CBP ameliorated the severity of the disease whilst inhibiting HDAC1 using the pan HDAC inhibitor TSA or the isoform specific inhibitor RG2833 worsened the outcome of the disease. The acetylation of lysine residues at 730 and 810 facilitated the interaction and recruitment of MAL and Myd88 to the TIR domain of TLR4, whilst the Myd88-independent pathway involving TRIF and TRAM were not affected by this acetylation. The inclusion of human PBMC data to validate their overall findings also makes this study relevant in the field of macrophage biology.

This body of work adds to our existing knowledge of TLR4 activation biology and is crucial to our understanding of TLR4-driven diseases including sepsis. The authors should be commended for identifying specific residues in TLR4 TIR domain that contributes to hyper activation of the receptor in addition to the identification of its key activator and repressor. Despite being most widely studied, TLR4 biology remains an enigma, with new mechanisms of activation and repression being reported every now and then. However, this does not undermine the novelty of this work. Using several in silico approaches, the authors also compliment their findings which further validates their hypothesis.

In my opinion, this body of work could be made even stronger if the authors also showed a hyperactive phenotype using an acetyl-mimic mutant of the TIR domain (K-Q mutation), at least for one lysine residue.

Can the authors confirm the specificity of HDAC1 using a genetic approach like siRNA mediated knockdown in macrophages or a Hdac1-tissue specific knockout model?

What is the immediate downstream consequence of TLR4-TIR acetylation? How do the signaling components that eventually drive the activation of NFkB get affected upon altered acetylation states of TLR4?

Minor Point: It seems that the authors while using multiple data sets always used student t-test for determining statistical significance. It would be better suited to use non-parametric ANOVA test in this respect for data sets involving more than 2 parameters.

Are these lysine residues conserved amongst other TLRs?

Referee #3:

The study by Li and Li et al. reports acetylation of lysine residues in the TLR4-TIR domain and contributions of these residues in mouse TLR4 to LPS-induced macrophage activation and host survival in a model of lethal endotoxemia. The authors implicate CBP as the TLR4-interacting acetyl transferase mediating TLR4-TIR-domain acetylation, whereas HDAC1 is suggested to be the corresponding deacetylase. The authors use TLR4-lysine mutant mice, as well as small molecule inhibitors to investigate contributions of TLR4-TIR acetylation to LPS-induced host responses in vivo. Acetylated lysine residues in the TLR4-TIR-domain in patients with sepsis is being described and relevance in monocyte polarization ascribed particularly to the CD16+CD14- monocyte subset.

Positioning TLR4 and TLR signaling adapters as direct targets of acetylation-mediated regulation adds an additional layer to the current understanding of how TLR-induced signaling and cellular functions are regulated by acetylation and HDACs. A strength of the manuscript is the application of complementary experimental approaches, indications from in silico, in vitro and vivo analyses, as well as analyses in humans and mice.

Major concerns.

1. The conclusions of the paper substantially rely on a set of newly generated antibodies that are suggested to specifically recognize singly acetylated lysine residues in the TLR4 TIR domain. Can the authors demonstrate specificity of these antibodies? The data presented in Fig S1B using peptide recognition are not sufficient to demonstrate that these antibodies recognize specifically acetylated K residues in TLR4. Fig S1B also only relates to the anti-mouse TLR4 antibodies.

2. The authors invested in the generation of mice with TLR4 mutations, K730R and K810R. These mice could become an asset to the field. This warrants a detailed description of these mice and confirmation that they are phenotypically unremarkable, e.g. leukocyte proportions and composition, base line histology, WT-equivalent responses to stimulation other than TLR4 agonists etc. This information will also aid interpretation of the observations related to survival and pathology in the in vivo endotoxemia model. It is also recommended to confirm that mutation of lysine residues in the TLR4 TIR domain does not affect total and cell surface expression of TLR4, to ensure that the impaired LPS responses in vivo and in vivo are attributable to impaired TLR4 function. Do the TLR4 K730R and K810R mutant macrophages exhibit defects in LPS-induced expression of TRIF-dependent genes (e.g. *Irfb1*)? This information would further support the proposal that lysine mutations specifically affect TLR4/Mal/MyD88 signaling.

3. If the authors wish to apply the "M1/M2" paradigm, it is worth noting that polarization in this framework is assigned based on relative changes of cellular expression of cell surface markers (in addition to functional attributes such as cytokine production), rather than proportions of cells expressing surface markers (that are not necessarily exclusive). Moreover, there are concerns over reliance on proportion of cells expressing CD86 as a single marker for M1-polarized macrophages.

4. The methodology section is missing important information for experiments (e.g. molecular modelling, surface plasmon resonance, in vivo small molecule inhibitor studies, CBP-KO mice...), or is incomplete (e.g. which TLR4 antibody used for which studies, type of LPS, ELISA, original report and background of TLR4-/- mice used here, human study cohorts inclusion criteria, clinical characteristics, and demographics...), or seem inconsistent across the document (e.g. antibodies against TLR4-TIR domain yet no indication of permeabilization methods used for the cells, LPS dose and time points and group sizes for in vivo studies...). Without detailed and accurate reflection of the experimental details, it is difficult to assess the quality and validity of study results presented.

Minor points:

- Figure 1F/G would benefit from controls (e.g. unstained cells and isotype, TLR4 signalling inhibitor).
- The analyses in Fig 3 would benefit from confirming that TLR4-lysine mutation does not significantly affect gene expression in unstimulated macrophages. How much overlap is there between the DEG upon LPS stimulation (compared to unstim cells) in each genotype? Could you provide this detail in a Venn diagram ?
- Without a comparator, the data in Figure 6 are difficult to interpret. What is the conclusion based on that the gene signatures are specific to CD16+CD14- TLR4-ack+ monocytes, as opposed to CD16+CD14- monocytes more generally?
- Histology analyses from mock-treated mice should be included in Fig 4 and Fig 7.
- Statistical analyses in Fig 7E should compare inhibitor treatments to vehicle-control (LPS)
- To link the effects of CBP and HDAC1 inhibitors in vivo to TLR4-TIR acetylation, rather than generalized effects on TLR4-driven host responses, the analyses in Fig 7 should be extended to the K810R mice.
- The data in S5E seem to show diminished TIR-domain acetylation with HDAC8 transfection, similar to HDAC1. Has a contribution by HDAC8 been investigated/excluded?
- Please include WCL / input controls for Fig S1C; Figure 2A/B: WCL blots for FLAG-MAL and HA-MyD88
- The experiments in Fig S3E require HA-CBP-only transfected cells as control. Were these cells co-transfected with MD-2 and CD14 to enable LPS-mediated cellular activation ?
- Please ensure statistical methods are fully described for all data and appropriate for the data type and experimental design (e.g. t-test is not suitable for >2 groups).
- Can the authors please clarify their approach to plasmid transfection of primary bone marrow-derived macrophages (Fig 2M, S3F)? These cells rapidly die upon liposome-mediated transfection of plasmid DNA, due to cytosolic surveillance pathways (AIM1 inflammasome).
- Please ensure that all figures are referred to in the text or remove if irrelevant.

Additional comments

- The WB data underpinning Fig 1A should be shown and quantification presented across independent experiments.
- Fig 7B,C - due to differential effects in unstimulated cells, normalization of data to relevant control is recommended.
- Fig S5A,B. Inhibitor effects on unstimulated cells should be confirmed and targets/specificities of all inhibitors noted.
- Page 3. Please confirm accuracy of reference made to Shi et al. 2015.

- Page 3. Please correct statement: "[...] MyD88, the critical adaptor molecule employed by all TLR family members, [...]" to reflect the fact that TLR3 does not engage MyD88.
- Page 3. "At first, in the myeloid differentiation factor 2 (MD2) knockout embryonic fibroblasts, TLR4 cannot reach the plasma membrane and do not respond to LPS (Nagai et al., 2002), implying that MD2 is required for the activation of TLR4 in the form of heterodimer complex."
MD-2 affecting TLR4 maturation cannot be interpreted as MD-2 requirement for TLR4 activation.
- It is recommended that the authors carefully distinguish their conclusions from biochemical studies that assess TLR4 ligand-induced events as opposed to events observed in overexpressing systems using the TLR4-TIR domain constructs.
- Please ensure it is always clear whether reference is made to human or mouse TLR4.

- "The analysis indicated that the proposed residues K732 and K813 were situated at the interaction surface of both TLR4-MAL and TLR4-MyD88 interaction models (Fig 2E)." The motivation for analysing a TLR4-MyD88 interaction model is curious if MyD88 is recruited to Mal rather than TLR4.

- It is recommended to reserve the term "significant" for such instances where an appropriate statistical analysis was undertaken and meets pre-defined criteria, e.g. $p < 0.05$.

- It will be helpful to clearly state in each case what cells were used, how many independent experiments have been performed with similar outcomes and what the data represent.

- "Overall, these findings indicated that CBP promoted TLR4-TIR acetylation, while HDAC1 served as the key deacetylase of TLR4-TIR in both mouse M1 macrophages and human CD16+ M1 macrophages."
This conclusion is an overstatement..

- Please provide references for the following statements:
"Sepsis is a severe systemic inflammatory response disease usually triggered by severe pathogenic infections, the transformation of M0 macrophages into M1 macrophages played a significant role in the progression of sepsis."

- "Our single-cell sequencing data demonstrated that sepsis marker genes [...]"

Y Eugene Chinn, M.D./Ph.D.
Director and Professor
Clinical Medicine Research Institute,
Zhejiang Provincial People's Hospital,
Hangzhou, Zhejiang 310014
Email: chinyue@suda.edu.cn

Jul 05, 2024

Dear Dr. Papaioannou,

We appreciate you for giving us this opportunity to revise our manuscript. We appreciate the constructive comments from the three reviewers. We have now carried out additional experiments and revised the manuscript according to the reviewers' suggestions, concerns, and criticisms.

Along with this letter, please find the revised manuscript entitled "LPS-induced TIR Domain Complex Acetylation Activates the TLR4/MAL/MyD88 Signaling Pathway in Sepsis (EMBOJ-2024-117310)". A copy with highlighted changes is also submitted for your reference, which includes a point-by-point response to address all the comments raised by the reviewers.

We thank you again for your kind consideration.

Sincerely yours,

Y Eugene Chinn

Point-by-point response

Reviewer #1: Li et al. showed that TLR4/MAL/MyD88 signaling pathway in macrophages is modulated by the acetylation of TIR domain complex by CREB-binding protein (CBP) in response to LPS. Specifically, the K732 and K813 residues of TLR4 are served as specific substrates of CBP, and the acetylation of these TLR4-TIR domain residues play essential roles in the classical TLR4-dependent signal transduction. The Authors also demonstrated that human CD16+ monocytes can differentiate into inflammatory macrophages. The CD16+ monocyte subset can act as key innate immune components of sepsis showing an increased acetylation level of TLR4-acK732 and TLR4-acK813. It was shown that targeting histone deacetylase 1 (HDAC1) is a potential deacetylase of TLR4-TIR domain in inflammatory type 1 (M1) macrophages, and the endotoxin-induced inflammatory response and sepsis progression can be modulated by targeting HDAC1 and CBP.

Major point of criticism:

1. One limitation of the study is that targeting histone deacetylases in vivo can modify the activation of TLR4-mediated signalosome, but the epigenomic regulation of inflammatory and anti-inflammatory gene expression could be affected at the same time. The Authors showed that the serum levels of pro-inflammatory cytokines in mice during sepsis show a statistical difference between control and HDAC1-inhibitor injected mice. However, these differences show minor biological differences in the serum-derived parameters to measure disease development.

Response: Indeed, that HDAC may affect both epigenetics in nuclei and TLR4 signaling in cytoplasm. Therefore, the histone deacetylase inhibitors may simultaneously affect the epigenomic gene expression regulation for inflammation or anti-inflammation in nuclei or the factor of upstream signaling pathway in cytoplasm. However, our results show that HDAC1 inhibitor significantly promoted the sepsis development of mice mainly manifested as a remarkable decreased survival rate, a significant increased M1 polarization of peritoneal macrophages, and a higher level of pro-inflammatory cytokines in serum than that of control (Figure 7). Therefore, sepsis development is closely related to the acetylation dependent TLR4 activation which is reversible by HDAC1.

Detection of these serum cytokines is one of the main indicators for evaluating hyper-inflammatory phase of sepsis patients in clinical setting. TNF, IL-1 β and IL-6 could be potentially useful cytokines as biomarkers of sepsis (Faix, 2013). Sepsis can be divided into three stages: sepsis, severe sepsis and septic shock (Andriolo et al., 2017). To better clarify it, we collected and analyzed the serum IL-6 and TNF- α levels of patients with sepsis and septic shock. We found that the serum IL-6 and TNF- α levels of patients with septic shock were significantly higher than that of sepsis patients. The related results were supplemented as Figure EV4H in the revised manuscript. Thus, we

believe that the serum levels of pro-inflammatory cytokines in mice during sepsis can show biological differences in the serum-derived parameters to measure disease development in some extent.

Figure EV4H. Serum IL-6 and TNF- α levels of patients with sepsis and septic shock. (n = 8 independent experiments).

2. It is not obvious how human CD16+ monocyte-derived inflammatory macrophage can be associated to the tested murine inflammatory macrophage (M1) polarization. The monocytes represent a heterogeneous populations of innate immune cells in mice, and it is not clear which population of macrophages were analyzed in the murine sepsis model. Are these monocyte-derived macrophages isolated from the spleen or from the peritoneum?

Response: In the mouse model of LPS-induced sepsis, the macrophages we analyzed were mainly from the peritoneum. Peritoneal macrophages may be newly generated macrophages derived from monocytes or tissue resident macrophages that are already present in the peritoneum (Wu et al., 2023). In human patients of sepsis, we unexpectedly observed a higher level of TLR4 acetylation in CD16+ monocytes than healthy individuals (Figure 5A-E).

Subsequently, upon sorting of TLR4-acetylated cells for single-cell analysis, we discovered that CD16+ monocytes exhibited a significantly higher expression level of M1 macrophage markers, such as CD86 (Figure 6E,F). Additionally, we also found that the expression of M1 macrophages-related activating factors IL-6 and TNF- α in TLR4-acetylated CD16+ monocytes were significantly increased in sepsis patients. The related results were supplemented as Figure EV4G in the revised manuscript. These new findings indicated the potential differentiation of TLR4-acetylated CD16+ monocytes to M1 macrophages due to the high expression of M1 macrophages markers in them.

Figure EV4G. Monocytes were obtained by sorting with TLR4-acK and CD16 antibodies from both normal individuals and sepsis patients. The expression of IL-6 and TNF- α in these cells were then detected using qRT-PCR. (n = 4 independent experiments).

Minor points of criticism:

1. Peritoneal macrophages express cell surface CD86 molecule at homeostatic environment, but the authors show that there is a significant CD86- population of macrophages in Figure 1 and 4. It is clearly visible that there are three populations on the dot plots; one negative, a CD86 low and a CD86 high cell population. The median or mean expression level of CD86 expression intensity should be presented instead of the frequency. Using technical controls, the authors can demonstrate that the macrophages express CD86 which expression level depends on the cell activation state.

Response: We appreciate the reviewer's suggestion. Now, we have replaced the calculation method of CD86+ macrophages with the median expression level of CD86 expression intensity in Figure 1N and Figure 4D of the revised manuscript. As shown in Figure 1N and Figure 4D, without LPS treatment, CD86 expression have no significant difference in TLR4-WT or non-acetylation TLR4-KR mutant mice. However, upon LPS treatment, CD86 expression in TLR4-WT mice was significantly increased and much higher than that of TLR4-KR mutant mice. These results were consistent with the previous results, suggesting that TLR4 lysine acetylation is very critical for M1 macrophages polarization.

Figure 1N. Macrophages from TLR4-WT or TLR4-KR mutant mice bone marrow were treated with or without 100 ng/mL LPS for 12 hours. Flow cytometry analysis of cells was conducted to detect the polarization of M1 macrophages by staining with anti-F4/80 (macrophage marker) and anti-CD86 (M1 marker) fluorescence antibodies. (n = 3 independent experiments).

Figure 4D. Mouse peritoneal macrophages obtained from TLR4-WT or TLR4-KR mutant mice after intraperitoneal injection with 25 mg/kg LPS for 4 h. Cells were collected for flow cytometry analysis to detect the polarization of M1 macrophages using anti-F4/80 (macrophage marker), anti-CD86 (M1 marker), and anti-CD206 (M2 marker) fluorescence antibodies. (n = 3 independent experiments).

2. The BMDM differentiation and collection of primary peritoneal macrophages is the section 'Cell lines' of Methods. The Authors have to separate the methods utilizing primary cells and cell lines.

Response: We fully agree with your suggestion. According to the suggestion, we have separated the methods utilizing primary cells and cell lines in the revised manuscript.

3. It is confusing how the Authors refer to macrophage subpopulations in figure legends and the results. The Authors should clearly define what is a mouse macrophage. A primary macrophage, macrophages obtained from bone marrow, M1 macrophage etc. These terms should be used consistently across the manuscript.

Response: We feel sorry for this confusion. To better define it, we have standardized the nomenclature of macrophages used in the revised manuscript.

4. On Figure 3D,E,F: please check if the gene Tlr3 is mistyped.

Response: We really appreciate the reviewer pointing out this mistake. The gene Tlr3 was incorrectly spelled, and we have corrected it to Tlr3 in the revised

manuscript.

5. The origo of y axis showing cytokine concentrations (Figure 7J) has to be zero.

Response: Thanks for the suggestion. We have changed the origo of y axis in Figure 7J to be zero in the revised manuscript.

Figure 7J. The levels of IL-6, TNF- α , and IL-1 β in the serum collected from TLR4-WT mice subjected to different treatments for 12 h were detected by ELISA assay. (n = 3 independent experiments).

6. In the discussion, some sentences should be reedited, or merged: "However, we were not sure the TLR4-TIR acetylation associated adaptor preference resulted from acetylation site specificity or cell type specificity. Even it is reported that it is CD14 that transferred LPS to the TLR4-MD2 dimerization complex (Park and Lee, 2013; Ryu et al., 2017)." ".... And marker genes..."

Response: Thank you for this suggestion. We have checked and re-edited some sentences in the revised manuscript.

Reviewer #2: In this manuscript, Li et al, present evidence towards acetylation of TIR domain of TLR4 dictating activation of downstream NF- κ B pathway and associated proinflammatory cytokine gene expression. This acetylation is mediated by the histone acetyltransferase(HAT), CBP and antagonized by the histone deacetylase, HDAC1. The authors identify two lysine residues in the TIR domain of TLR4, positioned at 730 and 810 (in mouse), of which lysine residue at 810th position is critical to this response. In an in vivo model of sepsis induced by LPS, antagonizing CBP ameliorated the severity of the disease whilst inhibiting HDAC1 using the pan HDAC inhibitor TSA or the isoform specific inhibitor RG2833 worsened the outcome of the disease. The acetylation of lysine residues at 730 and 810 facilitated the interaction and recruitment of MAL and Myd88 to the TIR domain of TLR4, whilst the Myd88-independent pathway involving TRIF and TRAM were not affected by this acetylation. The inclusion of human PBMC data to validate their overall findings

also makes this study relevant in the field of macrophage biology.

This body of work adds to our existing knowledge of TLR4 activation biology and is crucial to our understanding of TLR4-driven diseases including sepsis. The authors should be commended for identifying specific residues in TLR4 TIR domain that contributes to hyper activation of the receptor in addition to the identification of its key activator and repressor. Despite being most widely studied, TLR4 biology remains an enigma, with new mechanisms of activation and repression being reported every now and then. However, **this does not undermine the novelty of this work**. Using several in silico approaches, the authors also compliment their findings which further validates their hypothesis.

Response: Thanks for your affirmation and encouragement to us.

1. In my opinion, this body of work could be made even stronger if the authors also showed a hyperactive phenotype using an acetyl-mimic mutant of the TIR domain (K-Qmutation), at least for one lysine residue.

Response: Thanks for your great suggestion. Actually, In the preliminary experimental design, we have already constructed TLR4-K732Q and TLR4-K813Q mutants. Unfortunately, we did not see the activation effect of mimic K732 or K813 acetylation on the downstream pathway of TLR4. Meanwhile, either TLR4-K732Q or TLR4-K813Q did not show an enhance effect on the interaction between TLR4-TIR and MAL. Therefore, we did not conduct in-depth experiments in this article.

2. Can the authors confirm the specificity of HDAC1 using a genetic approach like siRNA mediated knockdown in macrophages or a Hdac1-tissue specific knockout model?

Response: Thanks for this suggestion. In response to it, we have added the HDAC1 siRNA to confirm the specificity of HDAC1 in regulation of TLR4 acetylation. After treated with siRNA-HDAC1, we found that the acetylation at the K732 and K813 residues were significantly increased, confirming that HDAC1 acts as the deacetylase for the two acetylation sites of TLR4. The related results were supplemented as Figure EV5F in the revised manuscript.

Figure EV5F. THP1 cells were induced into macrophages and transfected with siRNA for 36 hours, followed by 100ng/ml LPS treatment for 30 minutes. Cells were subjected to immunoprecipitation with TLR4 antibody followed by western blot with acK732-TLR4, acK813-TLR4 antibodies.

3. What is the immediate downstream consequence of TLR4-TIR acetylation? How do the signaling components that eventually drive the activation of NF κ B get affected upon altered acetylation states of TLR4?

Response: Thanks for this comment. In the literatures, It has been reported that the TIR domain of TLR4 can recruit two different sets of adaptors: one set including MyD88 and MAL, while the other set comprising TRIF and TRAM, which couple NF- κ B and IRF-3 signaling pathways, respectively. In our experiments, we found that LPS treatment can lead to the acetylation of TIR domains of TLR4 (Figure 1C,D), MyD88 (Figure EV1H) and MAL (Figure EV1I), resulting in the formation of the acetylated TIR domain complex. Interestingly, the acetylation of TLR4-TIR promotes the interaction between TLR4, MyD88, and MAL (Figure 2A-G), while not affecting the interaction with TRIF and TRAM (Figure EV2I-L). These TIR-TIR interactions and associated post-translational modifications within the complex ultimately lead to NF- κ B activation. Furthermore, by mutating the acetylation site of the TLR4-TIR domain, we observed a decrease in the nuclear translocation and activity of NF- κ B (Figure 2L,M).

Minor Point:

1. It seems that the authors while using multiple data sets always used student-t-test for determining statistical significance. It would be better suited to use non-parametric ANOVA test in this respect for data sets involving more than 2 parameters.

Response: We completely agree with your suggestion and have thoroughly reviewed it in the entire text. The comparison between two groups was assessed using Student's t-test. For more than two groups, we utilized one-way ANOVA.

2. Are these lysine residues conserved amongst other TLRs?

Response: Thanks for this comment. The comparison of TLR family members

in sequence alignment uncovered that the K732 of TLR4 is conserved among all TLR family members, whereas K813 was quite specific for TLR4. We have now included the related results as Figure EV1G in the revised manuscript.

Figure EV1G. While K732 is exclusively conserved among TLR family members, K813 is relatively specific for TLR4.

Reviewer #3: The study by Li and Li et al. reports acetylation of lysine residues in the TLR4-TIR domain and contributions of these residues in mouse TLR4 to LPS-induced macrophage activation and host survival in a model of lethal endotoxemia. The authors implicate CBP as the TLR4-interacting acetyl transferase mediating TLR4-TIR-domain acetylation, whereas HDAC1 is suggested to be the corresponding deacetylase. The authors use TLR4-lysine mutant mice, as well as small molecule inhibitors to investigate contributions of TLR4-TIR acetylation to LPS-induced host responses in vivo. Acetylated lysine residues in the TLR4-TIR-domain in patients with sepsis is being described and relevance in monocyte polarization ascribed particularly to the CD16+CD14-monocyte subset.

Positioning TLR4 and TLR signaling adapters as direct targets of acetylation-mediated regulation **adds an additional layer to the current understanding** of how TLR-induced signaling and cellular functions are regulated by acetylation and HDACs. **A strength of the manuscript is the application of complementary experimental approaches**, indications from in silico, in vitro and vivo analyses, as well as analyses in humans and mice.

Response: Thanks for your affirmation and encouragement to us.

Major concerns.

1. The conclusions of the paper substantially rely on a set of newly generated antibodies that are suggested to specifically recognize singly acetylated lysine residues in the TLR4 TIR domain. Can the authors demonstrate specificity of these antibodies? The data presented in Fig S1B using peptide recognition are not sufficient to demonstrate that these antibodies recognize specifically acetylated K residues in TLR4. Fig S1B also only relates to the anti-mouse TLR4 antibodies.

Response: We really appreciate the reviewer's insightful comments. In response, we have expanded our validation of the specificity of acK732 and acK813 antibodies. In addition to testing the acetylated and non-acetylated TLR4-K732/813 peptides, we also assessed the specificity of these antibodies using the acetylated TLR4-K780 peptide (acK780). The results showed that only the acetylated TLR4-K732/813 peptide exhibited specificity with their respective antibodies (Figure EV1B). Furthermore, we transfected HEK293T cells with TLR4-WT, TLR4-K732R, TLR4-K813R, and CBP plasmids. It was found that the TLR4-K732R group exhibited miniMAL detection with TLR4-acK732 antibodies, and the TLR4-K813R group showed miniMAL detection with TLR4-acK813 antibodies (Figure EV1C). Based on these findings, we are confident in the specificity of TLR4 acK732 and acK813 antibodies. We have added the related results as Figure EV1B and Figure EV1C in the revised manuscript.

Figure EV1B. The specificity of polyclonal antibodies targeting residues of TLR4-acK732 and TLR4-acK813 acetylation, using synthesized peptides as antigens, was assessed through dot blot analysis.

Figure EV1C. HEK293T cells were transfected with the indicated plasmids. Immunoprecipitation with Flag antibody was followed by western blot analysis with acK732 and acK813-TLR4 and Flag antibodies.

2. The authors invested in the generation of mice with TLR4 mutations, K730R and K810R. These mice could become an asset to the field. This warrants a detailed description of these mice and confirmation that they are phenotypically unremarkable, e.g. leukocyte proportions and composition, base line histology, WT-equivalent responses to stimulation other than TLR4 agonists etc. This information will also aid interpretation of the observations related to survival and pathology in the in vivo endotoxemia model. It is also recommended to confirm that mutation of lysine residues in the TLR4 TIR domain does not affect total and cell surface expression of TLR4, to ensure that the impaired LPS

responses in vivo and in vivo are attributable to impaired TLR4 function. Do the TLR4 K730R and K810R mutant macrophages exhibit defects in LPS-induced expression of TRIF-dependent genes (e.g. *Ifnb1*)? This information would further support the proposal that lysine mutations specifically affect TLR4/MAL/MyD88 signaling.

Response: Thanks for your great suggestion. We conducted blood routine tests on TLR4-WT, TLR4-K730R, and TLR4-K810R mice, which revealed no significant differences in white blood cells, granulocytes, and lymphocytes among the three groups (Figure EV3A). Additionally, we used flow cytometry to assess NK, B, and CD4⁺T cells in the spleens from these mice (Figure EV3B). The results indicated no significant differences among the three groups of mice. We have added the related results as Figure EV3A,B in the revised manuscript.

Figure EV3A. Blood routine analysis of white blood cells, lymphocytes, and granulocytes in the TLR4-WT or TLR4-KR mutant mice blood. (n = 3 independent experiments).

Figure EV3B. Splenic cells obtained from TLR4-WT or TLR4-KR mutant mice were collected for flow cytometry analysis to detect the population of NK, B, CD4⁺T cells. (n = 3 independent experiments).

We performed hematoxylin and eosin (H&E) staining on the lung and spleen tissues of TLR4-WT, TLR4-K730R and TLR4-K810R mice. The results indicated clear and transparent alveolar cavities in the mice, with the spleen structure remaining intact (Figure EV3C). After LPS treatment, western blot analysis was conducted to detect TLR4 expression of macrophages derived from three types of mice bone marrow, and no significant difference were observed among the groups (Figure EV3D,E). We have added the related results as Figure EV3C-E in the revised manuscript.

Figure EV3C. Representative H&E staining images of lung and spleen tissues

isolated from TLR4-WT or TLR4-KR mutant mice. Scale bar = 2 mm.

Figure EV3D,E. Macrophages derived from TLR4-WT or TLR4-KR mutant mice bone marrow were subjected to western blot analysis with TLR4 antibody (D). Quantitative analysis of the TLR4 expression were performed using ImageJ (E). (n = 3 independent experiments).

Additionally, qRT-PCR was used to assess the gene expression of *ifn-β1* in macrophages derived from bone marrow of TLR4-WT, TLR4-K730R, and TLR4-K810R mice. The results demonstrated a consistent pattern of *ifn-β1* gene expression with or without LPS treatment across all three types of mice (Figure EV4I). This observation aligns with our earlier findings indicating that the acetylation of TLR4-TIR domain does not enhance the recruitment of TRIF and TRAM (Figure EV2I-L). Meanwhile, we have also verified that TLR4-K730R or K810R mutants did not abolish LPS-mediated IRF3 activation (Figure EV2N). These findings suggest that TLR4-K730R and K810R mutations specifically affect TLR4/MAL/MyD88 signaling. We have added the related results as Figure EV4I in the revised manuscript.

Figure EV4I. Quantitative analysis of *ifn-β1* in macrophages derived from bone marrow of TLR4-WT or TLR4-KR mutant mice treated with or without LPS. (n = 3 independent experiments).

3. If the authors wish to apply the "M1/M2" paradigm, it is worth noting that polarization in this framework is assigned based on relative changes of cellular expression of cell surface markers (in addition to functional attributes such as cytokine production), rather than proportions of cells expressing surface markers (that are not necessarily exclusive). Moreover, there are concerns over reliance on proportion of cells expressing CD86 as a single marker for M1-polarized macrophages.

Response: We completely agree with your suggestion and have updated the charts by changing the proportion of CD86+ macrophages to relative expression of CD86. In addition to using flow cytometry to identify F4/80 and CD86 double positive cells as M1 cells, we also evaluated the expression of M1 macrophage activating factor TNF-α and IL-6.

4. Without detailed and accurate reflection of the experimental details, it is

difficult to assess the quality and validity of study results presented. The methodology section is missing important information for experiments (e.g. molecular modelling, surface plasmon resonance, in vivo small molecule inhibitor studies, CBP-KO mice...), or is incomplete (e.g. which TLR4 antibody used for which studies, type of LPS, ELISA, original report and background of TLR4-/- mice used here, human study cohorts inclusion criteria, clinical characteristics, and demographics...), or seem inconsistent across the document (e.g. antibodies against TLR4-TIR domain yet no indication of permeabilization methods used for the cells, LPS dose and time points and group sizes for in vivo studies...).

Response: Thanks for your great suggestion. We have supplemented these experimental details in the revised manuscript.

Minor points

1. Figure 1F/G would benefit from controls (e.g. unstained cells and isotype, TLR4 signalling inhibitor).

Response: Thanks for this suggestion. We have repeated the experiment in Figure 1F/G, and added the isotype control as you suggested. The new results were consistent with our previous conclusion, which LPS enhanced the acetylation of TLR4-K732 and K813 sites. We have replaced it as Figure 1F,G in the revised manuscript.

Figure 1F,G. Macrophages derived from mouse bone marrow were treated with 100 ng/mL LPS for 30 minutes followed by flow cytometry analysis to assess the acetylation of TLR4-K813 and TLR4-K732. (n = 4 independent experiments).

2. The analyses in Fig 3 would benefit from confirming that TLR4-lysine mutation does not significantly affect gene expression in unstimulated macrophages. How much overlap is there between the DEG upon LPS stimulation (compared to unstim cells) in each genotype? Could you provide this detail in a Venn diagram ?

Response: Thanks for this suggestion. Venn analysis of gene differences in TLR4-WT vs TLR4-K730R and TLR4-WT vs TLR4-K810R with or without LPS treatment were supplemented as Fig 3G,H in the revised manuscript.

Figure 3G,H. Venn analysis of gene differences in peritoneal macrophages from TLR4-WT and TLR4-K730R mutant mice treated with or without LPS treatment(G). Venn analysis of gene differences in peritoneal macrophages from TLR4-WT and TLR4- K810R treated with or without LPS treatment (H).

3. Without a comparator, the data in Figure 6 are difficult to interpret. What is the conclusion based on that the gene signatures are specific to CD16+CD14- TLR4-acK+ monocytes, as opposed to CD16+CD14- monocytes more generally?

Response: Thanks for this comment. Actually, we firstly observed that TLR4 acetylation predominantly occurs in the PBMCs of sepsis patients, particularly CD16+CD14- cells, while there was no significant difference in CD16-CD14+ or CD16+CD14+ cells (Figure 5). Therefore, we aimed to investigate that whether CD16+CD14- TLR4-acK+ cells are related to the development of sepsis in these patients. That is why we isolated TLR4-acK+ cells for single-cell sequencing analysis in figure 6. We found that CD16+CD14- cells in sepsis patients exhibited high expression of genes closely associated with the development of sepsis. Additionally, CD16+CD14- TLR4-acK+ cells displayed the specific characteristics of the M1 macrophage subtype in figure 6.

4. Histology analyses from mock-treated mice should be included in Fig 4 and Fig 7.

Response: Thanks for your suggestion. We have performed histology analyses on various types of mice that were mock-treated. It was found that the alveolar cavities of mice not treated with LPS were clean, transparent, and exhibited an intact spleen structure. We have added the related results as Figure EV3C,F in the revised manuscript.

Figure EV3C. Representative H&E staining images of lung and spleen tissues isolated from TLR4-WT or TLR4-KR mutant mice. Scale bar = 100 μ m.

Figure EV3F. Representative H&E staining images of lung and spleen tissues isolated from TLR4-WT mice subjected to various treatments. The scale bar = 100 μ m.

5. Statistical analyses in Fig 7E should compare inhibitor treatments to vehicle-control (LPS).

Response: Thank you for pointing out our mistake. We have corrected it in the revised manuscript.

6. To link the effects of CBP and HDAC1 inhibitors in vivo to TLR4-TIR

acetylation, rather than generalized effects on TLR4-driven host responses, the analyses in Fig 7 should be extended to the K810R mice.

Response: Thank you for your suggestion. In response, we have performed an evaluation of the effects of CBP and HDAC1 inhibitors on the TLR4-K810R mutant mice in a LPS-induced sepsis model. Initially, in the absence of LPS, SGC-CBP30, TSA or RG2833 did not cause damage to the structure of lung and spleen tissues of TLR4-K810R mice (Fig EV3G). Subsequently, in a LPS-induced sepsis model, the results showed that TSA and RG2833 treatment resulted in a lower survival rate of mice (Fig EV3H), more severe damage to lung and spleen tissues (Fig EV3I), compared to control group. We suspect that this phenomenon may be associated with the acetylation of TLR4-K732 site in TLR4-K810R mutant mice. But the CD86 expression of peritoneal macrophage (Fig EV3J-K) and tissues injury (Fig EV3I) have no significant difference between the control and SGC-CBP30 groups. It may be due to that TLR4-K810R mice naturally exhibit the reduced sepsis symptoms and M1 macrophage activity. Therefore, the therapeutic effect of SGC-CBP30 on TLR4-K810R mice is unobvious. We have added Figure EV3G-K in the revised manuscript.

Figure EV3G. Representative H&E staining images of lung and spleen tissues isolated from TLR4-K810R mice subjected to various treatments. The scale bar = 100 μ m.

Figure EV3H. Survival curves of TLR4-K810R mice subjected to different treatments after intraperitoneal injection with 20 mg/kg LPS. (n = 6 independent

experiments).

Figure EV3I. Representative H&E staining images of lung and spleen tissues isolated from TLR4-K810R mice subjected to various treatments after intraperitoneal injection with 20 mg/kg LPS for 12 h. The scale bar = 100 μ m.

Figure EV3J,K. Peritoneal macrophages were obtained from the TLR4- K810R mice that have been injected intraperitoneally with 20 mg/kg LPS for 12 h followed by exposure to different treatments. Then, Cells were collected for flow cytometer analysis to assess the polarization of M1 macrophages. (n = 3 independent experiments).

7. The data in S5E seem to show diminished TIR-domain acetylation with HDAC8 transfection, similar to HDAC1. Has a contribution by HDAC8 been investigated/excluded?

Response: Thanks for this comment. As shown in Figure S5E, we think the expression of HDAC8 in the input panel was much higher than that of HDAC1, while the deacetylates effect of HDAC1 was greater than HDAC8. Furthermore, in Figure S5A, the HDAC8 inhibitor PCI34051 cannot increase the polarization of M1 macrophages. Therefore, we did not further investigate the contribution of HDAC8.

8. Please include WCL / input controls for Fig S1C; Figure 2A/B: WCL blots for FLAG-MAL and HA-MyD88.

Response: Thanks for this suggestion. We have added the WCL or input controls in these assays. To increase the coherence of the article, original Figure S1C was set as Figure EV1D in the revised manuscript.

9. The experiments in Fig S3E require HA-CBP-only transfected cells as control. Were these cells co-transfected with MD-2 and CD14 to enable LPS-mediated cellular activation ?

Response: Thanks for your suggestion. Actually, HEK293T cells were co-transfected with TLR4, CBP, MD-2, and CD14 plasmids in this experiment. We found that these cells co-transfected with MD-2 and CD14 alone did not activate NF- κ B in response to LPS, while them co-transfected with TLR4, MD-2 and CD14 can a slightly enhancement of LPS-mediated NF- κ B activation. Notably, under CBP transfection, the LPS-mediated NF- κ B activation was significantly higher compared to other conditions. To increase the clarity of the article, the original Figure S3E was replaced with Figure EV2M in the revised manuscript.

10. Please ensure statistical methods are fully described for all data and appropriate for the data type and experimental design (e.g. t-test is not suitable for >2 groups).

Response: We completely agree with your suggestion and have thoroughly reviewed it in the entire text. The comparison between two groups was assessed using Student's t-test. For more than two groups, we utilized one-way ANOVA.

11. Can the authors please clarify their approach to plasmid transfection of primary bone marrow-derived macrophages (Fig 2M, S3F)? These cells rapidly die upon liposome-mediated transfection of plasmid DNA, due to cytosolic surveillance pathways (AIM1 inflammasome).

Response: We apologize for the oversight. After careful verification, we have confirmed that these two assays in Fig 2M and S3F were performed on HeLa cells. We did not do the experiment involved in TLR4^{-/-} mouse model.

12. Please ensure that all figures are referred to in the text or remove if irrelevant.

Response: Thanks for this suggestion. We have removed irrelevant data in the revised manuscript.

Additional comments

1. The WB data underpinning Fig 1A should be shown and quantification presented across independent experiments.

Response: Thanks for this suggestion. We have repeated it for three independent experiments, and the updated results were given in Figure 1A of the revised manuscript. The WB data underpinning Figure 1A was shown in Figure EV1.

Figure 1A. Monocyte derived from mouse bone marrow were induced into macrophages by administering 10 ng/ml m-CSF for 7 days, followed by stimulation with 100 ng/mL LPS for 0, 5, 10, 15, 30, and 60 minutes. Immunoprecipitation with a TLR4 antibody was followed by western blotting

with pan-acetyl-K and TLR4 antibodies. Whole cell lysate (WCL) was probed with I κ B α antibody. (n = 3 independent experiments).

2. Fig 7B,C - due to differential effects in unstimulated cells, normalization of data to relevant control is recommended.

Response: Thanks for this suggestion. To eliminate the effect of differential effects of inhibitors in unstimulated cells, we normalized data to a value of 1 in control group that was not stimulated with LPS.

3. Fig S5A,B. Inhibitor effects on unstimulated cells should be confirmed and targets/specificities of all inhibitors noted.

Response: Thanks for your valuable suggestion. We performed flow cytometry analysis to investigate the impact of the inhibitors mentioned in Figure S5A on M1 macrophage polarization in the absence of LPS stimulation. Detailed information regarding the dosage and targets of the HDAC family inhibitors can be found in supplementary Table 3. We have added the related results as Figure EV5B,C in the revised manuscript.

Figure EV5B. Macrophages derived from bone marrow of TLR4-WT mice were treated with a series of HDAC family inhibitors for 18 h. Cells were collected for flow cytometry analysis to assess the polarization of M1 macrophages.

Figure EV5C. Quantitatively analysis of the polarization of macrophages treated with or without LPS. (n = 3 independent experiments).

4. Page 3. Please confirm accuracy of reference made to Shi et al. 2015. Page 3. Please correct statement: "[...] MyD88, the critical adaptor molecule employed by all TLR family members, [...]" to reflect the fact that TLR3 does not

engage MyD88. Page 3. "At first, in the myeloid differentiation factor 2 (MD2) knockout embryonic fibroblasts, TLR4 cannot reach the plasma membrane and do not respond to LPS (Nagai et al., 2002), implying that MD2 is required for the activation of TLR4 in the form of heterodimer complex." MD-2 affecting TLR4 maturation cannot be interpreted as MD-2 requirement for TLR4 activation.

Response: Thanks for your great suggestion. We have re-edited these sentences to make it accurate in the revised manuscript.

5. It is recommended that the authors carefully distinguish their conclusions from biochemical studies that assess TLR4 ligand-induced events as opposed to events observed in overexpressing systems using the TLR4-TIR domain constructs.

Response: Thanks for your suggestion. We have made relevant modifications in the revised manuscript.

6. Please ensure it is always clear whether reference is made to human or mouse TLR4.

Response: Thanks for your suggestion. In order to avoid misunderstanding your question, we have made two modifications in our study. One is that we have supplemented the information in the references about whether TLR4 is made in human or mouse model. Another is that the TLR4-related plasmids used in WB experiments are all human TLR4 sequences, in which the acetylation of TLR4-TIR region sites are K732 and K813. These two sites are highly conserved with K730 and K810 in mouse TLR4, respectively. We have added these detailed information in the revised manuscript.

E

Hu HKSRR₇₃₂VIVVV...RRLR₈₁₃XALL
Mo HKSRR₇₃₀VIVVV...RRLX₈₁₀NALL

7. The analysis indicated that the proposed residues K732 and K813 were situated at the interaction surface of both TLR4-MAL and TLR4-MyD88 interaction models (Fig 2E)." The motivation for analysing a TLR4-MyD88 interaction model is curious if MyD88 is recruited to MAL rather than TLR4.

Response: TLR4 was originally discovered to recruit MyD88 for signaling (Wesche et al., 1997). Subsequently, MAL was reported to be recruited by MyD88 for the so called MyD88-dependent signaling (Bernard and O'Neill, 2013). Published evidence indicates that MAL also interacts directly with TLR4 and perhaps such association between TLR4 and MAL is stronger than TLR4 and MyD88 (Ohnishi et al., 2009). TLR4-MAL/MyD88 forms triple TIR domain-some is established obviously due to these three proteins all carry TIR domains (Thomas et al., 2017). Therefore, MAL and MyD88 can form complex, which may continue to form triple-TIR domain complex with TLR4. Given this signaling pathway is LPS stimulation dependent, TLR4's TIR domain

associates with MAL and/or MyD88 followed by MAL-MyD88 interaction is expected. But it does not rule out a basal level of MyD88-MAL complex formation in the cells independent of TLR4 activation. Such a TIR-domain triangle signalsome formation during TLR4 signaling may be quite stable for the downstream signaling during LPS stimulation.

8. It is recommended to reserve the term "significant" for such instances where an appropriate statistical analysis was undertaken and meets pre-defined criteria, e.g. $p < 0.05$.

Response: Thanks for your suggestion. When the statistical analysis was $P < 0.05$, we reserve the term "significant" in the revised manuscript.

9. It will be helpful to clearly state in each case what cells were used, how many independent experiments have been performed with similar outcomes and what the data represent.

Response: Thanks for your suggestion. We have added these detailed informations in the revised manuscript.

10. "Overall, these findings indicated that CBP promoted TLR4-TIR acetylation, while HDAC1 served as the key deacetylase of TLR4-TIR in both mouse M1 macrophages and human CD16+M1 macrophages." This conclusion is an overstatement..

Response: Thanks for this comment. We have changed it to "Overall, these findings indicated that CBP can promote TLR4-TIR acetylation, while HDAC1 may also serve as the deacetylase of TLR4-TIR in human CD16+ monocyte during sepsis." in the revised manuscript.

11. Please provide references for the following statements: "Sepsis is a severe systemic inflammatory response disease usually triggered by severe pathogenic infections, the transformation of M0 macrophages into M1 macrophages played a significant role in the progression of sepsis." "Our single-cell sequencing data demonstrated that sepsis marker genes [...]"

Response: Thanks for your suggestion. We have added the related references to these sentences in the revised manuscript.

References

- Andriolo, B.N.G., R.B. Andriolo, R. Salomão, and Á.N. Atallah. 2017. Effectiveness and safety of procalcitonin evaluation for reducing mortality in adults with sepsis, severe sepsis or septic shock. *Cochrane Database of Systematic Reviews* 2019:
- Bernard, N.J., and L.A. O'Neill. 2013. MAL, more than a bridge to MyD88. *IUBMB Life*
- Faix, J.D. 2013. Biomarkers of sepsis. *Critical Reviews in Clinical Laboratory*

Sciences 50:23-36.

- Ohnishi, H., H. Tochio, Z. Kato, K.E. Orii, A. Li, T. Kimura, H. Hiroaki, N. Kondo, and M. Shirakawa. 2009. Structural basis for the multiple interactions of the MyD88 TIR domain in TLR4 signaling. *Proceedings of the National Academy of Sciences* 106:10260-10265.
- Thomas, PariMALa, Vajjhala, Andrew, Hedger, Tristan, Croll, and Frank. 2017. Structural basis of TIR-domain-assembly formation in MAL- and MyD88-dependent TLR4 signaling. *Nature Structural & Molecular Biology*
- Wesche, H., W.J. Henzel, W. Shillinglaw, S. Li, and Z. Cao. 1997. MyD88: an adapter that recruits IRAK to the IL-1 receptor complex. *Immunity* 7:837-847.
- Wu, L., M. Kohno, J. Murakami, A. Zia, J. Allen, H. Yun, M. Chan, C. Baci, M. Liu, and V. Serrebeinier. 2023. Defining and targeting tumor-associated macrophages in MALignant mesothelioma. *Proceedings of the National Academy of Sciences of the United States of America*. 120.

Dear Prof. Chin,

Thank you for the submission of your revised manuscript to The EMBO Journal and your patience during its peer review. It has now been seen by the three original referees who previously assessed the earlier version of your manuscript, and we have received the full set of their comments (included below). As you will see, the referees recognize that all major experimental concerns have been appropriately addressed, and they acknowledge that the manuscript has been strengthened. Referee #1 has no other comments, while referees #2 and #3 list a number of suggestions for improvement and clarification of the text. We kindly ask you to address these remaining points in the text of a final revised version of your manuscript. Please also include in your resubmission a detailed point-by-point reply to the referee comments, explaining how you addressed their points and describing the respective changes to the manuscript.

From the editorial side, there are also a few changes and corrections that we need from you before we can proceed with acceptance of your manuscript for publication in The EMBO Journal:

- Please provide a list of up to 5 keywords after the Abstract of your revised manuscript.
- Please make sure that all deposited datasets mentioned in your Data availability statement are made publicly available. The reviewer access tokens can now be removed from this statement, while the active and permanent URLs giving public access to the datasets must be included.
- The author contributions statement should be removed from the manuscript file. Instead, we now use CRediT to specify the contributions of each author in the journal submission system. Please feel free to use the free text box to provide more detailed descriptions during submission. See also our guide to authors for more information:
<https://www.embopress.org/page/journal/14602075/authorguide#authorshipguidelines>.
- Please make sure that the callouts for Fig. 5F-H are closer in the text to the other callouts of Fig. 5 panels.
- Please note that EMBO press papers are accompanied online by:
 - A) a short (2 sentences) summary of the findings and their significance,
 - B) 2-5 short bullet points highlighting the key results, and
 - C) a synopsis image in .jpg or .png format that is exactly 550 pixels wide and 300-600 pixels high (the height is variable). Please note that the text needs to be legible at the final size. Please upload this information along with your revised manuscript (the text for A and B should be provided in a separate Word file).
- Please rename the Figure legends for Figure S5 to "Figure EV5" in the manuscript file.
- The Materials and Methods need to be described in the manuscript using our "Structured Methods" format, which is now required for all research articles. According to this format, the Materials and Methods section includes a single "Reagents and Tools Table" -listing key reagents, experimental models, software and relevant equipment and including their sources and relevant identifiers- followed by a "Methods and Protocols" section describing the methods using a step-by-step protocol format. The aim is to facilitate adoption of the methodologies across labs. More information on this format as well as detailed instructions, examples, and a template (.docx) for the "Reagents and Tools Table" can be found in our author guide:
<https://www.embopress.org/page/journal/14602075/authorguide#structuredmethods>.
- Please incorporate the information previously found in your "Supplementary table 1" and "Supplementary Table 3" in the new "Reagents and Tools Table" (see above for more information). Please update all callouts throughout the manuscript accordingly.
- The remaining "Supplementary Table 2" should be renamed to "Table 1". Please update all callouts throughout the manuscript accordingly.
- Please note that the exact p values should be provided in the legends of Figures 2m; 3i-j; 4e-f; 5b-e, h; 7b; EV 4c, g; EV 5c.
- Please indicate the statistical test used for data analysis in the legends of Figures 6c, g.
- Please note that information related to "n" is missing in the legend of Figure 6f.
- Please note that for Figures 1i; 2d, l; 5f; the scale bar unit should be defined in the Figure legends rather than in the Figures themselves (please remove the units from the figures). Please also note that the scale bar units should be corrected from μM to μm .

Please also note that as part of the EMBO publications' Transparent Editorial Process, The EMBO Journal publishes online a

Peer Review File along with each accepted manuscript. This File will be published in conjunction with your paper and will include the referee reports, your point-by-point response and all pertinent correspondence relating to the manuscript. You can opt out of this by letting the editorial office know (contact@embojournal.org). If you do opt out, the Peer Review File link will point to the following statement: "No Peer Review File is available with this article, as the authors have chosen not to make the review process public in this case."

We look forward to seeing a final version of your manuscript as soon as possible. Please use this link to submit your revision: <https://emboj.msubmit.net/cgi-bin/main.plex>

Yours sincerely,

Referee #1:

The authors addressed all the issues raised.

Referee #2:

The authors have addressed all the concerns I had raised. However, it would be best if the authors could clarify certain things.

1. Reviewer 3 Minor Point 2 - The Venn Diagram description/explanation is confusing. For panel G, do the authors mean that there are 379 genes that are differentially regulated between WT and K730R (-LPS)? This would raise a different set of curiosity if mutation of one lysine residue at a basal level induces so many differential expression of genes. Please clarify in the description section of the manuscript. Same logic goes for Panel H as well.
2. Figure 2D - TSA treatment promoted interaction between TLR4-TIR and Myd88 and TLR4-TIR and Mal. But the microscopy images don't convincingly state similar facts. Could the authors shed some light as to how they came to this conclusion from these images?
3. In all of the RMSD plots, the authors are claiming 'significant' differences in interaction between the signalosome complexes with the TLR4-TIR K-R mutants. However, these plots again are very unconvincing (except EV2A). The manuscript would definitely benefit from adjacent Area under the curve plots to these RMSD plots to confirm the difference observed as being stated in the manuscript.

Referee #3:

The additional data and corrections have improved the manuscript. Nevertheless, it is recommended that the authors consider the following aspects for clarity and accuracy:

1. Abstract - reconsider accuracy and context of the first sentence
2. Page 4 - "Inhibition of HDAC1 exacerbated sepsis-associated syndromes, [...]" Please reconsider use of the word syndrome and perhaps more specifically detail the observations made.
3. Description of Fig 1L (p5) - it appears that CBP KD did not "abolish" but rather reduce LPS-induced TLR4 acetylation
4. Description of Fig 1M,N (p6) - the comparison between WT and K730R does not meet statistical significance.
5. While it is appreciated that the heterologous expression data indicate that TLR4 and MyD88 interact directly, most current data and models of TLR4 activation place MAL as an intermediary between TLR4 and MyD88. This warrants acknowledgement, contextualisation, and discussion.
6. Page 7 - "The formation of the TLR4/MAL/MyD88 complex [...]" This statement is somewhat misleading as it seems to suggest that TLR4/MAL/MyD88 (rather than TLR4/TRAM/TRIF) signaling activates IRF3?

7. Page 8 - description of Fig 3J - the data show that LPS induces cytokine release in TLR4-WT but also in TLR4-KR macrophages. It is just less in the mutant cells and this is what the statistical analysis addresses. Similarly, please re-consider interpretation of data in Fig 4E,F, described on p 9.
8. P 9 - "fluid accumulation [...] in the alveolar cavity." It is not clear what is meant by this.
9. P9, 1st paragraph - positioning of data presented in Fig EV4H does not align well with the animal model focused on here.
10. P9 - Fig 5A, EV4A, EV4B - the authors state that "levels" TLR4 and acetylates TLR4 are elevated in sepsis. But the data indicate that there are more cells expressing TLR4 or ac-TLR4. The levels of expression do not seem to be different. Please clarify in results and discussion.
11. P9 / Fig 6F - expression of CXCL6 and IL15 seems to be elevated in both CD16+ as well as in the CD16/CD14+ subsets when compared to the CD14+ only subsets?
12. P10 - "and served as the main pro-inflammatory cells in the progression of sepsis." The data presented by the authors do not support this statement.
13. P10 - EV5A-C - these analyses are restricted to CD86 expression and in the absence of further phenotypic characterisation, it is advisable to just state the actual analysis performed, rather than naming it "polarization of M1 macrophage analysis".
14. P11 - subheading - specify that these analyses were done in mice
15. Inhibitor treatment of mice described on p11 and related methodology is not clear on inhibitor doses and time of administration in the context of LPS challenge. Was a solvent control administered with LPS alone? Fig 7C - the effect of SGC-CBP30 does not meet criteria for statistical significance (contrary to the statement on p11). Similarly, SGC-CBP30 effects in Fig 7I,J are subtle and only meet criteria for statistical significance in one instance.
16. The discussion would benefit from some expansion on the potential molecular mechanisms underpinning CBP and HDAC1 regulation of TLR4 activation. Also, sustained impact of CBP and HDAC inhibition on survival and pathology in the K810 mice might indicate CBF and HDAC1 effects outside of TLR4 acetylation. There would be value in exploring this in more detail in the discussion.
17. P15 - antibodies against TLR4- K732 and TLR4-K813: please detail which animals were immunized (including ethics clearance) and details of antibody purification and validation. The extended data in Fig EV1B-D are very helpful and could be extended using cells from the TLR4 mutant mice to establish specificity in both species.
18. P15 Histology - reference is made to "fresh hearts"?
19. P16 - Human samples and Table 2 - requires details on the clinical parameters of sepsis and septic shock that were applied here as inclusion criteria. It is relevant and meaningful to describe the patient cohort and report sepsis-relevant parameters including (but not limited to) SOFA score, ionotropes, vasopressors, requirement for ventilation, etc.
20. P18 - Surface plasmon resonance - what is the source and purity of recombinant human TLR4? According to Fig 1K, recombinant CBP was also used but no details are provided.
21. P18/19 - flow cytometry - please clarify whether/when/how cells were permeabilised for analysis of ac-TLR4.
22. What was the statistical analysis method for mouse survival?
23. Reporting of "independent experiments" - Please distinguish independent experiments (independent experiments using groups of animals or cells from animals) from individuals (patients or mice). Can the authors confirm that the in vivo mouse experiments reflect observations more than one independent experiment?
24. Fig 1J - Please provide input and WCL controls.
25. Fig 7B,C, EV2M - these experiments show differences unstimulated cells. Comparing the impact of inhibitors or transfection upon LPS stimulation needs to consider these differences in baseline responses. One approach is to compare the effect of LPS compared to untreated cells within each independent experiment and for each condition. It is then possible to more accurately compare the effects of LPS in the various treatment conditions.
26. Fig EV3H - please provide statistical analyses
27. Fig EV 3C,F,G,I would benefit from scale bars for insets.
28. Fig EV4I seems misplaced.

Y Eugene Chinn, M.D./Ph.D.
Director and Professor
Clinical Medicine Research Institute,
Zhejiang Provincial People's Hospital,
Hangzhou, Zhejiang 310014
Email: chinyue@suda.edu.cn
August 13, 2024

Dear Dr. Papaioannou,

Thank you for your email dated August 1, 2024. We greatly appreciate the constructive feedback from the reviewers and are grateful for the opportunity to revise our manuscript. In response to your request and the reviewers' comments, we have conducted additional experiments and made further revisions to the manuscript.

Along with this letter, please find the revised manuscript entitled "LPS-induced TIR Domain Complex Acetylation Activates the TLR4/MAL/MyD88 Signaling Pathway in Sepsis (EMBOJ-2024-117310)". We have also submitted a copy with highlighted changes for your reference, which includes a point-by-point response addressing all the reviewers' comments.

We thank you again for your kind consideration.

Sincerely yours,

Y Eugene Chinn

Point-by-point response

From the Editor:

All editorial and formatting issues were resolved by the authors.

Referee #1:

The authors addressed all the issues raised.

Response: Thanks for your affirmation and encouragement to us.

Referee #2:

The authors have addressed all the concerns I had raised. However, it would be best if the authors could clarify certain things.

1. Reviewer 3 Minor Point 2 - The Venn Diagram description/explanation is confusing. For panel G, do the authors mean that there are 379 genes that are differentially regulated between WT and K730R (-LPS)? This would raise a different set of curiosity if mutation of one lysine residue at a basal level induces so many differential expression of genes. Please clarify in the description section of the manuscript. Same logic goes for Panel H as well.

Response: When comparing macrophages from TLR4-WT and TLR4-K730R mutant strains, 390 genes (379 + 11) are differentially regulated even in the absence of LPS treatment. This data reflects the basal level of gene regulation mediated by the K730 site. However, upon LPS treatment, 27 genes (16 + 11) are differentially regulated. These genes represent those influenced by LPS through the K730 site. Specifically, 16 genes are dependent on both the K730 site and LPS stimulation, while 11 genes are differentially regulated independently of LPS stimulation. Similarly, for the K810 site, 1339 genes (1235 + 104) are differentially regulated. LPS treatment results in differential expression of 403 genes (299 + 104). Of these, 299 genes are dependent on both the K810 site and LPS stimulation for expression, while 104 genes are differentially regulated independently of LPS stimulation.

2. Figure 2D - TSA treatment promoted interaction between TLR4-TIR and Myd88 and TLR4-TIR and Mal. But the microscopy images don't convincingly state similar facts. Could the authors shed some light as to how they came to this conclusion from these images?

Response: Thank you for suggestion. Based on the immunostaining results, we found that without TSA treatment, the overlapping color exhibited a yellow dominance, indicating the co-localization of MAL (green) and MyD88 (red). In contrast, after TSA treatment, the overlapping color displayed a fuchsia dominance, signifying the co-localization of TLR4 (purple) and MyD88 (red). We have provided a detailed description of these findings in the revised manuscript.

3. In all of the RMSD plots, the authors are claiming 'significant' differences in interaction between the signalosome complexes with the TLR4-TIR K-R mutants. However, these plots again are very unconvincing (except EV2A). The manuscript would definitely benefit from adjacent Area under the curve plots to these RMSD plots to confirm the difference observed as being stated in the manuscript.

Response: Thank you for your suggestion. As we know, RMSD represents the total all atomic deviations between a conformation and the target conformation at a certain moment, serving as a key indicator of system stability. As shown in Fig EV2A, both systems reached a stable state after 60 ns, with average values of 0.230 ± 0.010 nm and 0.260 ± 0.016 nm, respectively, indicating that both systems achieved relative stability during the simulation. In Fig EV2B, we observed that the MyD88 protein in the TLR4-MyD88 and TLR4acetyl-MyD88 systems exhibited considerable fluctuations throughout the simulation, with average RMSD values of 0.270 ± 0.035 nm and 0.203 ± 0.026 nm after 60 ns, respectively. The greater fluctuation of the MyD88 protein in the TLR4-MyD88 system suggests that the acetylation of TLR4 may enhance the stability of its binding to MyD88.

Referee #3:

The additional data and corrections have improved the manuscript. Nevertheless, it is recommended that the authors consider the following aspects for clarity and accuracy:

1. Abstract - reconsider accuracy and context of the first sentence.

Response: Thank you for your suggestion. We have revised it to: "TLR4 activation by bacterial endotoxin in macrophages plays a crucial role in the pathogenesis of sepsis. However, the precise mechanism underlying TLR4 activation in macrophages is still not fully understood." This change has been incorporated into the revised manuscript.

2. Page 4 - "Inhibition of HDAC1 exacerbated sepsis-associated syndromes, [...]." Please reconsider use of the word syndrome and perhaps more specifically detail the observations made.

Response: We have revised it to: "Inhibition of HDAC1 exacerbated the M1 macrophage polarization and pro-inflammatory cytokine production leading to the progression of LPS-induced sepsis, whereas CBP inhibition alleviated these symptoms." This change has been incorporated into the revised manuscript.

3. Description of Fig 1L (p5) - it appears that CBP KD did not "abolish" but rather reduce LPS-induced TLR4 acetylation.

Response: We have corrected it to: "Conversely, depletion of CBP in macrophages dramatically reduced the induction of TLR4 acetylation by LPS (Fig 1L) ..." in the revised manuscript.

4. Description of Fig 1M,N (p6) - the comparison between WT and K730R does not meet statistical significance.

Response: We have corrected it to: "the results revealed a significant reduction in CD86 expression in TLR4-K810R mutant mice." in the revised manuscript.

5. While it is appreciated that the heterologous expression data indicate that TLR4 and MyD88 interact directly, most current data and models of TLR4 activation place MAL as an intermediary between TLR4 and MyD88. This warrants acknowledgement, contextualisation, and discussion.

Response: Thank you for your advice. We have added the related discussion in the revised manuscript.

6. Page 7 - "The formation of the TLR4/MAL/MyD88 complex [...]." This statement is somewhat misleading as it seems to suggest that TLR4/MAL/MyD88 (rather than TLR4/TRAM/TRIF) signaling activates IRF3?

Response: Thank you for your advice. We have corrected it to: "The formation of the TLR4 signalosome complex led to the activation of several serine/threonine kinases, resulting in nuclear localization of NF- κ B and IRF3." in the revised manuscript.

7. Page 8 - description of Fig 3J - the data show that LPS induces cytokine release in TLR4-WT but also in TLR4-KR macrophages. It is just less in the mutant cells and this is what the statistical analysis addresses. Similarly, please re-consider interpretation of data in Fig 4E,F, described on p 9.

Response: Thank you for your suggestion. We have corrected the description of Fig 3J to: "These cytokines were significantly upregulated by LPS in TLR4-WT macrophages, showing higher levels compared to those in TLR4-KR macrophages (Fig 3J)." in the revised manuscript.. And the description of Fig 4E,F were also corrected to: "Furthermore, analysis of pro-inflammatory cytokine mRNA expression in abdominal macrophages revealed significant upregulation of IL-6 and TNF- α mRNA levels by LPS in WT macrophages. However, a weaker mRNA upregulation trend of them was observed by LPS in TLR4-KR macrophages compared to WT macrophages (Fig 4E), which was further confirmed by their serum concentrations (Fig 4F)." in the revised manuscript.

8. P9 - "fluid accumulation [...] in the alveolar cavity." It is not clear what is meant by this.

Response: The fluid accumulation refers to effusion. When sepsis causes lung injury, it damages the alveolar-capillary membrane, increasing its permeability. This allows protein-rich edema fluid to enter the alveolar cavity.

9. P9, 1st paragraph - positioning of data presented in Fig EV4H does not align well with the animal model focused on here.

Response: Thank you for your suggestion. We have placed this data into the description of sepsis patient section in the revised manuscript.

10. P9 - Fig 5A, EV4A, EV4B - the authors state that "levels" TLR4 and acetylates TLR4 are elevated in sepsis. But the data indicate that there are more cells expressing TLR4 or ac-TLR4. The levels of expression do not seem to be different. Please clarify in results and discussion.

Response: Thank you for your suggestion. To clarify the contribution of TLR4-K732 and K813 acetylation to the progression of sepsis, we examined the expression of TLR4 and TLR4-acK in sepsis patients and calculated the ratios of TLR4-acK732/TLR4 and TLR4-acK813/TLR4. While elevated levels of TLR4 and TLR4-acK were detected in all monocyte populations of sepsis patients, the relative acetylation level (TLR4-acK/TLR4) was significantly elevated only in CD16+ monocytes of sepsis patients compared to normal individuals (Fig 6B-E and EV4C-F).

11. P9 / Fig 6F - expression of CXCL6 and IL15 seems to be elevated in both CD16+ as well as in the CD16/CD14+ subsets when compared to the CD14+ only subsets?

Response: Indeed, the expression of CXCL6 and IL15 are elevated in both CD16+ and the CD16/CD14+ subsets. Circulating monocytes can be divided into three subpopulations: classical (CD14+), intermediate (CD14+CD16+), and nonclassical (CD16+) (Williams et al., 2023). It is reasonable that the intermediate subset (CD14+CD16+) can be considered as the subset of CD16+ cells.

12. P10 - "and served as the main pro-inflammatory cells in the progression of sepsis." The data presented by the authors do not support this statement.

Response: Thank you for your comment. Indeed, we found that genes associated with the progression and prognosis of sepsis were highly elevated in CD16+ monocytes, including NAP1L1, CFD, BID, BCL2A1, MALAT1, RNH1, and LILRA5 (Fig. 5C,D). And IL16, CXCL6, IL15, and IL21R genes were found to be significantly elevated in CD16+ monocytes (Fig. 5F). An increase of classical M1 pro-inflammatory macrophage markers, CD68 and CD86, was also detected in CD16+ monocytes (Fig. 5E). Additionally, markers of M1 macrophage activation, TNF- α and IL-6, were significantly elevated in CD16+ cells from sepsis patients (Fig. EV4G). Therefore, we think that CD16+ monocytes may have the potential to differentiate into the classical M1 macrophage phenotype and serve as one of the important pro-inflammatory cells in the progression of human sepsis.

13. P10 - EV5A-C - these analyses are restricted to CD86 expression and in the absence of further phenotypic characterisation, it is advisable to just state the actual analysis performed, rather than naming it "polarization of M1 macrophage analysis".

Response: Thank you for your suggestion. We have made the related changes in the revised manuscript.

14. P11 - subheading - specify that these analyses were done in mice.

Response: Thank you for your suggestion. We have made appropriate changes to the subheading.

15. Inhibitor treatment of mice described on p11 and related methodology is not clear on inhibitor doses and time of administration in the context of LPS challenge. Was a solvent control administered with LPS alone? Fig 7C - the effect of SGC-CBP30 does not meet criteria for statistical significance (contrary to the statement on p11). Similarly, SGC-CBP30 effects in Fig 7I,J are subtle and only meet criteria for statistical significance in one instance.

Response: Thank you for your suggestion. The doses and time of HDAC inhibitors were provided in "Small molecule inhibitor" method of the revised manuscript. We have set LPS alone as solvent control in Fig EV5A-C. Additionally, we have noticed that some data in Fig 7C, I, and J indeed showed no statistically significant difference between the SGC-CBP30 group and the control group, while these data have a decreasing trend in the SGC-CBP30 group. Moreover, the expression of CD86 (Fig 7B), TNF- α (Fig 7I) and IL1 β (Fig 7J) have the significantly difference ($p < 0.05$) between the SGC-CBP30 group and the control group. We have re-edited these description more precise in the revised manuscript.

16. The discussion would benefit from some expansion on the potential molecular mechanisms underpinning CBP and HDAC1 regulation of TLR4 activation. Also, sustained impact of CBP and HDAC inhibition on survival and pathology in the K810 mice might indicate CBF and HDAC1 effects outside of TLR4 acetylation. There would be value in exploring this in more detail in the discussion.

Response: Thank you for your suggestion. We have added some discussion about these issues in the revised manuscript.

17. P15 - antibodies against TLR4- K732 and TLR4-K813: please detail which animals were immunized (including ethics clearance) and details of antibody purification and validation. The extended data in Fig EV1B-D are very helpful and could be extended using cells from the TLR4 mutant mice to establish specificity in both species.

Response: Thank you for your suggestion. We have included the details of these experiments in the Materials and Methods section of the revised manuscript. All animal experiments were conducted in accordance with the approved protocol by Zhejiang Provincial People's Hospital (Number: A20230401001). Additionally, we evaluated the specificity of the TLR4-acK antibody using TLR4 mutant mice. As shown in Fig EV1D, the acetylation antibody could not detect the corresponding site when the lysine residue of TLR4 was mutated to arginine, confirming the high specificity of these antibodies.

Fig EV1D. Macrophages derived from the bone marrow of TLR4-WT and TLR4-KR mice were treated with 100 ng/mL LPS for 30 minutes. Subsequently, the cells were immunoprecipitated with a TLR4 antibody, followed by western blot analysis using TLR4-acK732 and TLR4-acK813 antibodies.

18. P15 Histology - reference is made to "fresh hearts"?

Response: We really appreciate for pointing out our mistake. We have corrected it in the revised manuscript.

19. P16 - Human samples and Table 2 - requires details on the clinical parameters of sepsis and septic shock that were applied here as inclusion criteria. It is relevant and meaningful to describe the patient cohort and report sepsis-relevant parameters including (but not limited to) SOFA score, ionotropes, vasopressors, requirement for ventilation, etc.

Response: Thank you for your suggestion. The inclusion criteria for patients with sepsis and septic shock are based on "The Third International Consensus Definitions for Sepsis and Septic Shock (Sepsis-3.0)"(Singer et al., 2016). Additionally, we have included this information in the "Human Sample" method section of the revised manuscript. We have also added details about the patient cohort and sepsis parameters, such as SOFA score, vasoactive drug therapy, mechanical ventilation, C-reactive protein, and procalcitonin, to Table 1 in the revised manuscript.

20. P18 - Surface plasmon resonance - what is the source and purity of recombinant human TLR4? According to Fig 1K, recombinant CBP was also used but no details are provided.

Response: We have included these detailed information in the revised manuscript.

21. P18/19 - flow cytometry - please clarify whether/when/how cells were permeabilised for analysis of ac-TLR4.

Response: Thank you for your suggestion. We have incorporated these detailed information in the revised manuscript.

22. What was the statistical analysis method for mouse survival?

Response: The statistical analysis method used for mouse survival was the log-rank test. We have included these detailed information in the revised manuscript.

23. Reporting of "independent experiments" - Please distinguish independent experiments (independent experiments using groups of animals or cells from animals) from individuals (patients or mice). Can the authors confirm that the in vivo mouse experiments reflect observations more than one independent experiment?

Response: Thank you for your suggestion. We apologize for any confusion caused. We have corrected the related content to ensure greater precision in the revised manuscript.

24. Fig 1J - Please provide input and WCL controls.

Response: Thank you for your suggestion. We have supplemented the input of Fig 1J in the revised manuscript.

25. Fig 7B,C, EV2M - these experiments show differences unstimulated cells. Comparing the impact of inhibitors or transfection upon LPS stimulation needs to consider these differences in baseline responses. One approach is to compare the effect of LPS compared to untreated cells within each independent experiment and for each condition. It is then possible to more accurately compare the effects of LPS in the various treatment conditions.

Response: Thank you for this comment. Indeed, we did observe a difference between the inhibitor group and control group in the absence of LPS. However, we believe that the baseline response is justifiable. As shown in Fig 7D, TLR4 acetylation in macrophages was detected even without LPS treatment. The impact of the inhibitors or transfection on the expression of CD86, iNOS and NF- κ B activation (Fig 7B,C, EV2M) may contribute to the baseline levels of TLR4 acetylation. Following LPS treatment, the effects of the inhibitors or transfection on these markers were amplified, likely due to the synergistic regulation of TLR4 acetylation by both LPS and the inhibitors.

26. Fig EV3H - please provide statistical analyses

Response: Thank you for your suggestion. We have added these detailed information in the revised manuscript.

27. Fig EV 3C,F,G,I would benefit from scale bars for insets.

Response: Thank you for your suggestion. We have added the scale bars to these figures in the revised manuscript.

28. Fig EV4I seems misplaced.

Response: Thank you for pointing out this mistake. We have corrected it in the revised manuscript.

Singer, M., C.S. Deutschman, C.W. Seymour, M. Shankar-Hari, D. Annane, M. Bauer, R. Bellomo, G.R. Bernard, J.-D. Chiche, C.M. Coopersmith, R.S. Hotchkiss, M.M. Levy, J.C. Marshall, G.S. Martin, S.M. Opal, G.D. Rubenfeld, T. van der Poll, J.-L. Vincent, and D.C. Angus. 2016. The Third International Consensus Definitions for Sepsis and Septic Shock (Sepsis-3). *Jama* 315:

Williams, H., C. Mack, R. Baraz, R. Marimuthu, S. Naralashetty, S. Li, and H. Medbury. 2023. Monocyte Differentiation and Heterogeneity: Inter-Subset and Interindividual Differences. *International Journal of Molecular Sciences* 24:

Dear Prof. Chin,

Congratulations on an excellent manuscript, I am very pleased to inform you that it has been accepted for publication in The EMBO Journal. Thank you very much for your comprehensive responses to the referee concerns and for addressing all editorial requests.

Your manuscript will now be processed for publication by EMBO Press. It will be copy edited and you will receive page proofs prior to publication. Please note that you will be contacted by Springer Nature Author Services to complete licensing and payment information.

If you have any questions, please do not hesitate to contact the Editorial Office. Thank you for your contribution to The EMBO Journal. Working with you has been a pleasure!

Best wishes,

Ioannis
